EMBO
Molecular Medicine

# Loss of EHMT2 enhances NK cell-driven anti-tumor immunity through TGF-β1 suppression

Suresh Chava [ID][1], Suresh Bugide [ID][1], Parmanand Malvi [ID][1], Kelly D DeMarco[2,3,4], Boyang Ma[2,3,4], Chaitanya N Parikh[2,3,4], Marcus Ruscetti [ID][2,3,4], Allan Zajac [ID][5,6], Guoping Cai[7], Romi Gupta [ID][1,6✉] & Narendra Wajapeyee [ID][1,6✉]

## Abstract

**Natural Killer (NK) cells play a critical role in regulating tumor growth, but our understanding of the mechanisms underlying their anti-tumor activity remains limited. We identified the histone methyltransferase EHMT2 as a key suppressor of NK cell-mediated cytotoxicity. EHMT2 inhibition in cancer cells enhanced NK cell-mediated elimination of diverse cancers, including uveal melanoma, breast cancer, and pancreatic cancer. EHMT2 loss increased AZGP1 and decreased TGF-β1 levels, resulting in the autocrine elevation of NKG2D ligands MICB and ULBP3, chemokines in cancer cells, and the paracrine stimulation of NK cell function. In a syngeneic pancreatic cancer model, EHMT2 inhibition suppressed tumors in an NK cell-dependent manner, as NK cell depletion restored tumor growth. This effect persisted and remained dependent on NK cells in *Rag2* knockout mice (lacking T and B cells), but not in NSG mice (lacking T-, B- and NK-cells). Furthermore, EHMT2 and TGF-β1 inhibitors suppressed tumors in immunocompetent, but not in immunodeficient mice. These findings establish EHMT2 as a suppressor of NK cell-mediated anti-tumor immunity and a promising therapeutic target.**

**Keywords** Anti-tumor immunity; NK Cells; EHMT2; TGF-β1; AZGP1
**Subject Categories** Cancer; Chromatin, Transcription & Genomics; Immunology

## Introduction

The immune system plays a central role in preventing cancer initiation and progression (Ghorani et al, 2023; Hiam-Galvez et al, 2021; Mellman et al, 2023). However, tumors can evade immune detection via several distinct mechanisms, resulting in an immunosuppressive microenvironment that facilitates tumor progression by allowing cancer cells to grow and metastasize to distant organs largely unchecked (Ghorani et al, 2023; Seliger and Koehl, 2022; Tauriello et al, 2022; Togashi et al, 2019; Veglia et al, 2021; Zheng et al, 2023). Therefore, understanding these interactions between cancer cells and the immune system is essential for developing immunotherapies that can effectively target and eliminate tumors.

Natural killer (NK) cells are a component of the innate immune system and can suppress tumor growth through their ability to recognize and destroy malignant cells without prior sensitization (Bugide et al, 2018b; Seliger and Koehl, 2022; Vivier et al, 2024). NK cells exert their cytotoxic effects by releasing cytolytic granules containing perforin and granzymes, which induce apoptosis in target cells (Bugide et al, 2018b; Vivier et al, 2024; Zheng et al, 2023). They also secrete cytokines, such as IFN-γ, which enhance the anti-tumor activity of other immune cells (Vivier et al, 2024). Tumors can evade NK cell-mediated cytotoxicity by downregulating NKG2D ligands, or by secreting immunosuppressive cytokines that inhibit NK cell function (Bugide et al, 2018b; Seliger and Koehl, 2022; Vivier et al, 2024). Therapeutic strategies aimed at enhancing NK cell activity are being tested in the clinic to overcome these evasion mechanisms and improve anti-tumor immunity (Bugide et al, 2018b; Demaria et al, 2019). However, we still do not fully understand how cancer cells escape NK cell-mediated anti-tumor immunity. A better understanding of this aspect will enable more effective immune control of cancer and significantly enhance the outcomes of NK cell-based therapies.

Euchromatic histone lysine methyltransferase 2 (EHMT2) is a histone methyltransferase that primarily catalyzes the mono- and dimethylation of histone H3 at lysine 9 (H3K9me1/2), leading to chromatin condensation and transcriptional repression (Tachibana et al, 2001; Tachibana et al, 2002). EHMT2 plays an important role in the regulation of gene expression, maintenance of genomic stability, and cellular differentiation (Pribluda et al, 2022; Tachibana et al, 2001; Tachibana et al, 2002). EHMT2 is frequently overexpressed, and, in some cases, mutated, and contributes to various malignancies by enhancing cell proliferation, survival, and metastasis through its ability to silence tumor suppressor genes and facilitate oncogenic pathways (Pribluda et al, 2022; Sun et al, 2024; Yang et al, 2024). EHMT2 also plays an important role in immune

[1]Department of Biochemistry and Molecular Genetics, University of Alabama at Birmingham, Birmingham, AL 35233, USA. [2]Department of Molecular, Cell, and Cancer Biology, University of Massachusetts Chan Medical School, Worcester, MA 01605, USA. [3]Immunology and Microbiology Program, University of Massachusetts Chan Medical School, Worcester, MA 01605, USA. [4]Cancer Center, University of Massachusetts Chan Medical School, Worcester, MA 01605, USA. [5]Department of Microbiology, University of Alabama at Birmingham, Birmingham, AL 35233, USA. [6]O'Neal Comprehensive Cancer Center, University of Alabama at Birmingham, Birmingham, AL 35233, USA. [7]Department of Pathology, Yale University School of Medicine, New Haven, CT 06510, USA. ✉E-mail: romigup@uab.edu; nwajapey@uab.edu

regulation by inducing changes in both cancer cells and immune cells (Kato et al, 2020; Li et al, 2022; Scheer and Zaph, 2017; Sun et al, 2024). For instance, gain-of-function genetic alterations of EHMT2 in cutaneous melanoma cells can drive tumorigenesis and immune suppression in a T cell-dependent manner (Kato et al, 2020). EHMT2 has also been shown to stimulate M1 macrophage polarization and regulate immune cell differentiation and function (Li et al, 2022; Scheer and Zaph, 2017). Furthermore, inhibition of EHMT2 in microsatellite-stable colorectal cancer can enhance T cell-mediated immune responses via upregulation of galectin-7 (Sun et al, 2024). However, whether EHMT2 is important for NK cell-mediated tumor suppression is not known.

In this study, we identified EHMT2 as a key suppressor of NK cell-mediated anti-tumor immunity across multiple cancer types. EHMT2 inhibition enhanced NK cell-mediated cytotoxicity by upregulating AZGP1 and NKG2D ligands, reducing TGF-β1 levels in cancer cells, and relieving TGF-β1-mediated suppression of NK cell function and migration. Tumor suppression by EHMT2 inhibition was NK cell-dependent, highlighting its potential as a therapeutic target.

# Results

## A chemical epigenetic inhibitor screen identifies EHMT2 as a regulator of NK cell-mediated cytotoxicity

The ability of cancer cells to evade the host immune system, such as escaping NK cell-mediated anti-tumor immunity, is one of the key necessities for their unchecked growth. Although several regulators and mechanisms of NK cell-mediated cytotoxicity against cancer cells have been identified (Bugide et al, 2018b; Vivier et al, 2024; Wolf et al, 2023), our understanding of the regulators and mechanisms of NK cell-mediated anti-tumor immunity still remains incomplete. Thus, with the goal of identifying such regulators, we performed a chemical genetic screen using uveal melanoma (UM) cells as a model system. UM is an aggressive and rare form of melanoma that arises in the eye and is typically resistant to a wide variety of therapeutic approaches (Jager et al, 2020; Johansson et al, 2020; Leonard-Murali et al, 2024). To perform the chemical genetic screen in UM cells, we used the structural genomics consortium (SGC) epigenetic chemical probes library, which comprises 32 small-molecule inhibitors targeting 36 different epigenetic regulators (Table EV1). We treated the UM cell line MP46 with these small-molecule inhibitors of epigenetic regulators and co-cultured it with the human NK cell line, NK92MI. We then measured the NK cell-mediated cytotoxicity against UM cells using a lactate dehydrogenase (LDH)-based cytotoxicity assay (Fig. 1A) (Chava et al, 2020). The screen was based on the rationale that if an epigenetic regulator drives escape from NK cell-mediated cytotoxicity, its inhibition will promote NK cell-mediated tumor cell eradication. Through this screen, we identified several small-molecule inhibitors that increased NK cell-mediated cytotoxic clearance of the UM cell line MP46. These included inhibitors of BAZ2A/2B; BAZ2-ICR, EZH2; GSK343, BRPF1/2/3; OF1, L3MBTL3; UNC1215, PADI4; GSK484, JMJD3/UTX; GSK-J4, LSD1; KDM1A; GSK-LSD1, mutant IDH1; GSK864, and EHMT2; UNC0642 and A366 (Figs. 1B,C and EV1A,B).

The results of the primary chemical genetic screen were further validated in three additional UM cell lines (Mel92.1, MP41, and OMM2.5 cells). These studies showed that two different EHMT2 inhibitors, UNC0642 and A366, consistently enhanced the NK cell-mediated cytotoxicity in all tested UM cell lines (Figs. 1B,C and EV1C–E). UNC0642 and A366 can also inhibit EHMT1. EHMT1 forms a complex with EHMT2 to promote H3K9 methylation, and they often co-localize in chromatin silencing regions (Tachibana et al, 2008; Tachibana et al, 2005). We next investigated whether, similar to EHMT2, EHMT1 also regulates NK cell-mediated cytotoxicity. However, in our studies, we did not identify a direct interaction of EHMT1 with EHMT2 in UM cells (Appendix Fig. S1A). Additionally, we did not observe a consistent pattern of impact on NK cell-mediated cytotoxicity following EHMT1 knockdown across the tested UM cell lines (Appendix Fig. S1B,C).

To further establish that the observed impact of EHMT2 loss led to increased NK cell-mediated cytotoxicity in UM cell lines, we used a genetic approach. To do so, we knocked down EHMT2 expression using short-hairpin RNAs (shRNAs) in UM cell lines (Mel92.1 and MP41). In complete agreement with the results obtained with EHMT2 inhibitors, EHMT2 knockdown also enhanced NK cell-mediated cytotoxicity against UM cells compared to nonspecific (NS) shRNA-expressing UM cells (Fig. 1D–G; Appendix Fig. S2A).

Similar increases in NK cell-mediated cytotoxicity were observed across various NK cell-to-cancer cell ratios in both EHMT2 inhibitor-treated UM cells and UM cells expressing EHMT2 shRNAs (Appendix Fig. S2B,C). To further validate the role of EHMT2 loss in enhancing NK cell-mediated cytotoxicity, we employed a fluorescence-based quantitative Calcein-AM assay. This assay confirmed that both the treatment of cancer cells with EHMT2 inhibitors (UNC0642 and A366) and the knockdown of EHMT2 in cancer cells significantly increased NK cell-mediated cytotoxicity in UM cell lines (Appendix Fig. S2D,E).

Finally, we investigated whether the effects observed with EHMT2 inhibitors and EHMT2 shRNAs could be recapitulated through CRISPR-Cas9-based genetic knockout of EHMT2. Consistent with our findings, EHMT2 knockout led to increased NK cell-mediated cytotoxicity in UM cell lines Mel92.1 and MP41 (Fig. EV2A,B). Collectively, these results demonstrate that both pharmacological and genetic inhibition of EHMT2 enhance NK cell-mediated eradication of UM cells. Based on these collective findings, we focused on EHMT2 for further studies.

## EHMT2 inhibition enhances NK cell-mediated cytotoxicity via upregulation of its transcriptional target AZGP1

EHMT2 functions by causing transcriptional repression (Tachibana et al, 2001; Tachibana et al, 2002). Therefore, to identify the potential downstream mediators of EHMT2 function, we performed an RNA-sequencing (RNA-seq) analysis. The UM cell lines Mel92.1 and MP41 were treated with EHMT2 inhibitors UNC0642 or A366, followed by RNA-seq. The analysis of RNA-seq data identified 8 commonly upregulated genes (AZGP1, BANCR, BST2, KRT23, LINC00426, NOX5, SERPINA1, and SLAMF7) in both A366- and UNC0642-treated UM cell lines (Mel92.1 and MP41)

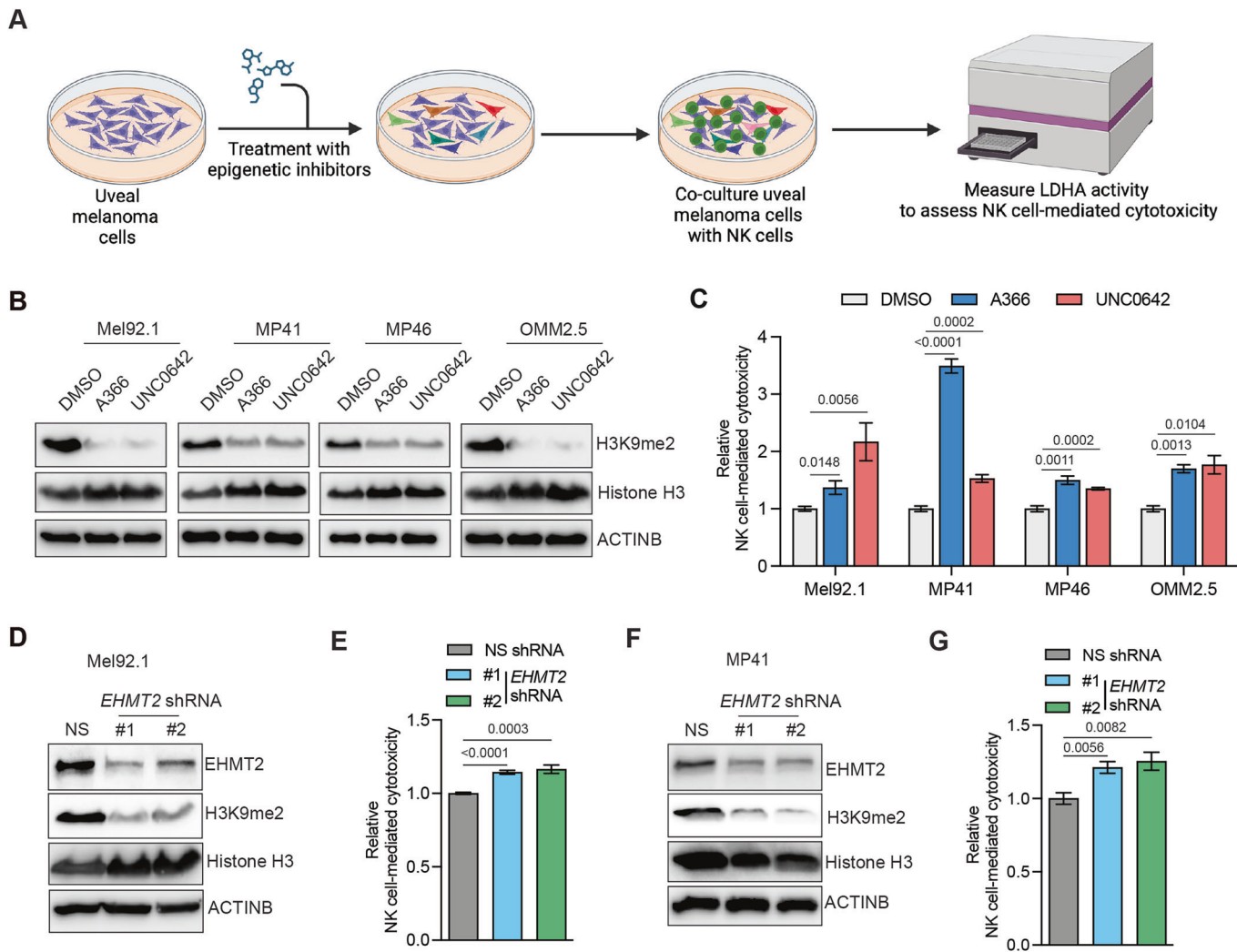

**Figure 1. An epigenetic regulator inhibitor screen identifies EHMT2 as a suppressor of NK cell-mediated cytotoxicity.**

(A) Schematic overview of the chemical epigenetic inhibitor screen used to identify suppressors of NK cell-mediated cytotoxicity. (B) Indicated uveal melanoma (UM) cells were treated with EHMT2 inhibitors A366 (1 µM) or UNC0642 (1 µM) or DMSO control for 48 h, and immunoblotting was performed to measure histone H3 lysine 9 dimethylation (H3K9me2). Histone H3 and ACTINB were used as loading controls. (C) Indicated UM cell lines were treated with A366 (1 µM) or UNC0642 (1 µM) or with DMSO as a negative control for 48 h. NK cell-mediated cytotoxicity was measured using LDH-based cytotoxicity assay. Relative NK cell-mediated cytotoxicity under indicated conditions for indicated UM cell lines is plotted. [($n = 6$ for Mel92.1), ($n = 5$ for MP41), ($n = 4$ for DMSO and $n = 5$ for UNC0642 and A366 for MP46), ($n = 3$ for OMM2.5)]. *P* values were calculated using an unpaired two-tailed Student's *t*-test. (D) Immunoblotting to measure expression of EHMT2 and H3K9me2 mark in Mel92.1 cells expressing either nonspecific (NS) shRNA or *EHMT2* shRNAs. Histone H3 and ACTINB were used as loading controls. (E) Mel92.1 cells expressing either NS shRNA or *EHMT2* shRNAs were analyzed for NK cell-mediated cytotoxicity using LDH-based cytotoxicity assay. Relative NK cell-mediated cytotoxicity under the indicated conditions are plotted. ($n = 6$). *P* values were calculated using an unpaired two-tailed Student's *t*-test. (F) Immunoblotting to measure the expression of EHMT2 and H3K9me2 mark in MP41 cells expressing either NS shRNA or *EHMT2* shRNAs. Histone H3 and ACTINB were used as loading controls. (G) MP41 cells expressing either NS shRNA or *EHMT2* shRNAs were analyzed for NK cell-mediated cytotoxicity using LDH-based cytotoxicity assay. Relative NK cell-mediated cytotoxicity under the indicated conditions are plotted. ($n = 5$). *P* values were calculated using an unpaired two-tailed Student's *t*-test. All quantitative data were presented as mean ± SEM. Source data are available online for this figure.

compared to the DMSO-treated cells (Mel92.1 and MP41) (Fig. 2A,B; Dataset EV1). We then performed validation of the candidates identified from RNA-seq using RT-qPCR in both UM cell lines (Mel92.1 and MP41) and determined that Alpha-2-glycoprotein 1, zinc-binding (AZGP1), also known as Zinc-α2-glycoprotein (ZAG), was the most consistently upregulated gene in both A366- and UNC0642-treated UM cell lines (Mel92.1 and MP41) at both the mRNA and protein levels (Fig. 2C–E). Finally, to establish that AZGP1 is a direct target of EHMT2, we performed a

CUT&RUN assay to measure the recruitment of EHMT2 and the associated H3K9me2 mark on the *AZGP1* promoter. The *ACTINB* promoter was used as a negative control. We found that EHMT2 was significantly enriched on the *AZGP1* promoter but not on the negative control *ACTINB* gene promoter (Fig. 2F). Consistent with this, the H3K9me2 mark was also enriched on the *AZGP1* promoter but not on the *ACTINB* gene promoter (Fig. 2G). Notably, patient sample data analysis revealed that increased *AZGP1* expression was also associated with better overall patient survival in UM patients

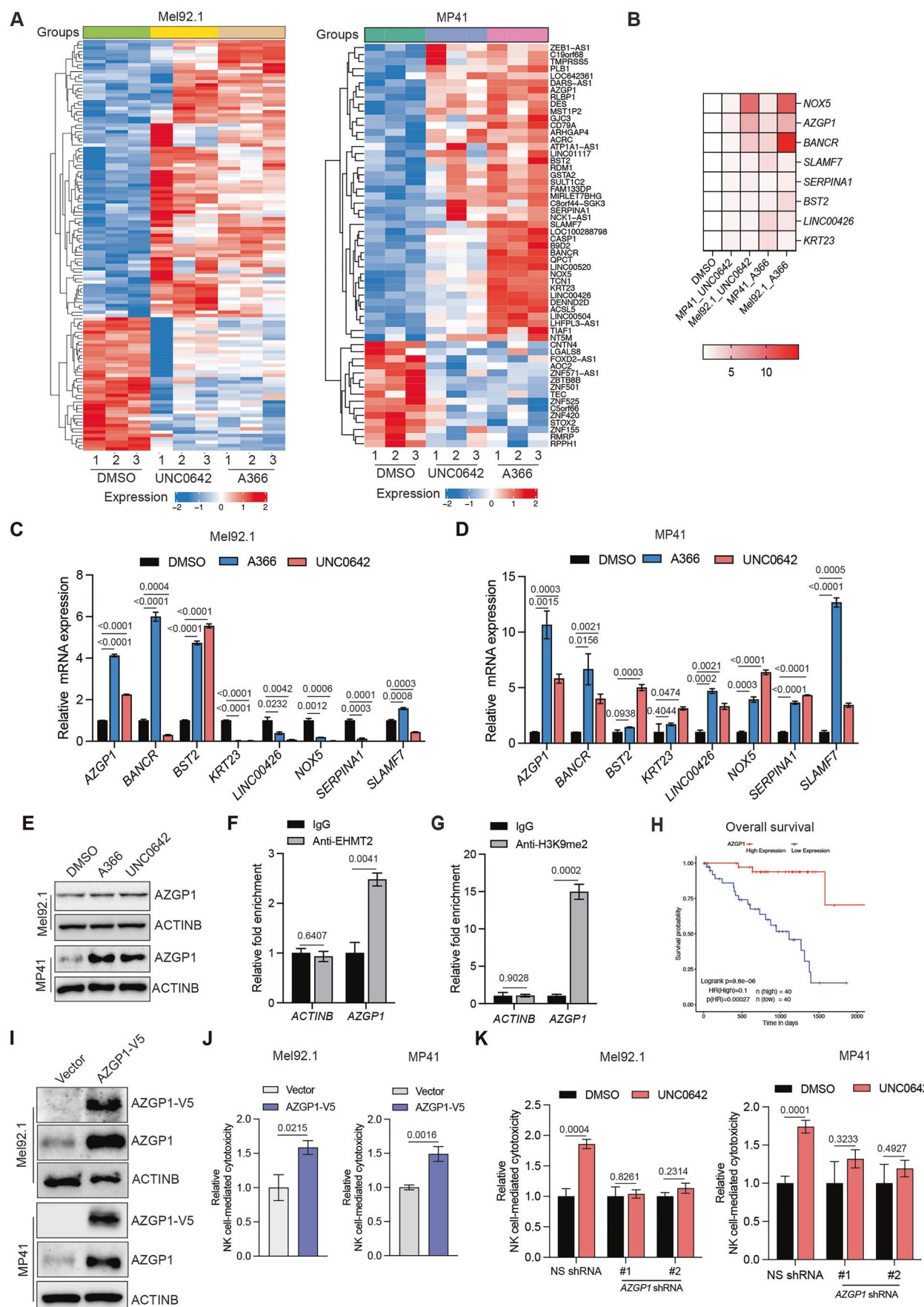

Figure 2. **EHMT2 inhibition-induced increase in NK cell-mediated cytotoxicity is dependent upon EHMT2 target gene AZGP1.**

(A) Heatmap showing differentially expressed mRNAs in Mel92.1 and MP41 cells following 48 h treatment with UNC0642 (1 µM) or A366 (1 µM) compared to DMSO-treated cells. (B) Heatmap showing commonly upregulated mRNAs in Mel92.1 and MP41 cells following 48 h treatment with UNC0642 (1 µM) and A366 (1 µM) compared to DMSO-treated cells. (C) Mel92.1 cells treated with DMSO or EHMT2 inhibitors UNC0642 (1 µM) or A366 (1 µM) for 48 h were analyzed for the indicated mRNAs using RT-qPCR. mRNA expression relative to DMSO-treated cells is plotted. *ACTINB* was used for normalization. ($n = 3$). *P* values were calculated using an unpaired two-tailed Student's *t*-test. (D) MP41 cells treated with DMSO or EHMT2 inhibitors UNC0642 (1 µM) or A366 (1 µM) for 48 h were analyzed for the indicated mRNA using RT-qPCR. mRNA expression relative to DMSO-treated cells is plotted. *ACTINB* was used for normalization. ($n = 3$). *P* values were calculated using an unpaired two-tailed Student's *t*-test. (E) Immunoblotting for AZGP1 in Mel92.1 and MP41 cells treated with DMSO or EHMT2 inhibitors UNC0642 (1 µM) or A366 (1 µM) for 48 h. ACTINB was used as a loading control. (F) EHMT2 enrichment on the *AZGP1* or ACTINB promoters was measured in Mel92.1 cells using the CUT&RUN assay. Relative fold enrichment of EHMT2 on the *AZGP1* and *ACTINB* promoters compared to the IgG control is plotted. ($n = 3$). *P* values were calculated using an unpaired two-tailed Student's *t*-test. (G) H3K9me2 mark enrichment on the *AZGP1* or *ACTINB* promoters was measured in Mel92.1 cells using CUT&RUN assays. Relative fold enrichment of the H3K9me2 mark on the *AZGP1* and *ACTINB* promoters compared to IgG control is plotted. ($n = 3$). *P* values were calculated using an unpaired two-tailed Student's *t*-test. (H) Overall survival for UM patients with higher ($n = 40$) or lower ($n = 40$) levels of *AZGP1* mRNA expression in the UM TCGA dataset. The log-rank test was used to calculate the *p* value of comparing Kaplan–Meier curves, and the Wald test was used to calculate the *p* value of Cox proportional hazards regression analysis. (I) Mel92.1 and MP41 cells expressing either an empty vector or V5-tagged *AZGP1* ORF were analyzed for the expression of AZGP1 by immunoblotting using V5-tag antibody or AZGP1 antibody. ACTINB was used as a loading control. (J). Mel92.1 and MP41 cells expressing either an empty vector or V5-tagged *AZGP1* ORF were analyzed for NK cell-mediated cytotoxicity using an LDH-based cytotoxicity assay. Relative NK cell-mediated cytotoxicity under the indicated conditions are plotted. ($n = 6$). *P* values were calculated using an unpaired two-tailed Student's *t*-test. (K) Mel92.1 and MP41 cells expressing either nonspecific (NS) shRNA or *AZGP1* shRNAs were treated with DMSO or EHMT2 inhibitors UNC0642 (1 µM) for 48 h and analyzed for NK cell-mediated cytotoxicity using LDH-based cytotoxicity assay. Relative NK cell-mediated cytotoxicity under the indicated conditions are plotted. ($n = 5$ for Mel92.1 and $n = 6$ for MP41). *P* values were calculated using unpaired two-tailed Student's *t*-test. All quantitative data were presented as mean ± SEM. Source data are available online for this figure.

(Fig. 2H). AZGP1 is a soluble protein encoded by the AZGP1 gene located on chromosome 7. This glycoprotein is primarily known for its role in lipid metabolism, specifically in the mobilization of fat stores (Faulconnier et al, 2019; Kong et al, 2010). Furthermore, in a series of cancer types, AZGP1 has been shown to function as a tumor suppressor (Hanamura et al, 2024; Huang et al, 2013; Kong et al, 2010; Tang et al, 2017). Based on this collective literature, we focused on testing AZGP1 as a downstream mediator of the NK cell regulatory function of EHMT2.

We first evaluated the role of AZGP1 in NK cell-mediated cytotoxicity against UM cells. To test this, we ectopically expressed AZGP1 in UM cells and determined whether AZGP1-expressing cells are more sensitive to NK cell-mediated cytotoxicity compared to vector-expressing cells. We found that ectopic expression of AZGP1 promoted NK cell-mediated eradication of UM cells (Fig. 2I,J). Next, we asked whether AZGP1 upregulation is necessary for the EHMT2 inhibition-induced increase in NK cell-mediated cytotoxicity. To test this likelihood, we knocked down *AZGP1* expression using shRNAs in UM cell lines (Mel92.1 and MP41) (Fig. EV2C,D). The *AZGP1* knockdown UM cells were then treated with the EHMT2 inhibitor UNC0642, and then NK cell-mediated cytotoxicity was measured. Control nonspecific (NS) shRNA-expressing and DMSO-treated UM cells were used as negative controls. We found that knocking down *AZGP1* significantly inhibited NK cell-mediated cytotoxicity following UNC0642 treatment in UM cells (Fig. 2K). Collectively, these results demonstrate that the upregulation of AZGP1 following EHMT2 inhibition is both necessary and sufficient for enhanced NK cell-mediated cytotoxicity.

## Loss of EHMT2 enhances NK cell-mediated cytotoxicity via the AZGP1-TGF-β1 pathway

The TGF-β pathway plays a crucial role in immune suppression against cancer cells by several mechanisms, including directly inhibiting immune cell function (Batlle and Massague, 2019; Massague and Sheppard, 2023; Tauriello et al, 2022). AZGP1 has been shown to inhibit the TGF-β pathway in cancer (Kong et al,

2010). Based on these previous reports, we asked if AZGP1 upregulation after EHMT2 suppression results in simultaneous attenuation of the TGF-β pathway in UM cells. Therefore, we first re-analyzed our RNA-seq data and identified that the TGF-β1 was downregulated in EHMT2 inhibitor-treated UM cells (Dataset EV1). Consistent with the results of RNA-seq, we were able to validate the downregulation of TGF-β1 in UM cells following EHMT2 inhibitor treatment (UNC0642 or A366), following EHMT2 knockdown, or following ectopic expression of AZGP1 (Fig. 3A–C).

Next, we asked if TGF-β1 downstream of EHMT2 drives suppression of NK cell-mediated cytotoxicity. To test this, we knocked down the expression of TGF-β1 in UM cell lines (Mel92.1 and MP41) (Fig. 3D) and measured NK cell-mediated cytotoxicity. We found that *TGF-β1* knockdown in UM cell lines (Mel92.1 and MP41) resulted in increased NK cell-mediated UM cell eradication (Fig. 3E).

Next, we asked whether TGF-β1, acting downstream of EHMT2, is required for the increase in NK cell-mediated cytotoxicity driven by EHMT2 loss. To determine this, we ectopically expressed TGF-β1 in UM cells and treated these cells with EHMT2 inhibitors (UNC0642 and A366) and measured NK cell-mediated cytotoxicity. We found that the ectopic expression of TGF-β1 in UM cells (Mel92.1 and MP41) rendered them more resistant to NK cell-mediated cytotoxicity even after EHMT2 inhibitor treatment, compared to control UM cells expressing vector alone (Fig. 3F–I). Furthermore, double knockdown of *TGF-β1* and *AZGP1* established that *AZGP1* knockdown-induced resistance to NK cell-mediated cytotoxicity was reversed following simultaneous *TGF-β1* knockdown in UM cells (Fig.EV2E,F). Taken together, these results demonstrate that TGF-β1 expression downstream of EHMT2 is sufficient to inhibit NK cell-mediated cytotoxicity.

Since TGF-β1 can promote immune evasion by acting on both cancer cells and NK cells (Batlle and Massague, 2019), we asked if the conditioned media collected from EHMT2 inhibitor-treated cells or AZGP1- and TGF-β1-expressing cells could influence NK cell-mediated cytotoxicity. We treated NK cells with conditioned media collected from EHMT2 inhibitor-treated or AZGP1- or

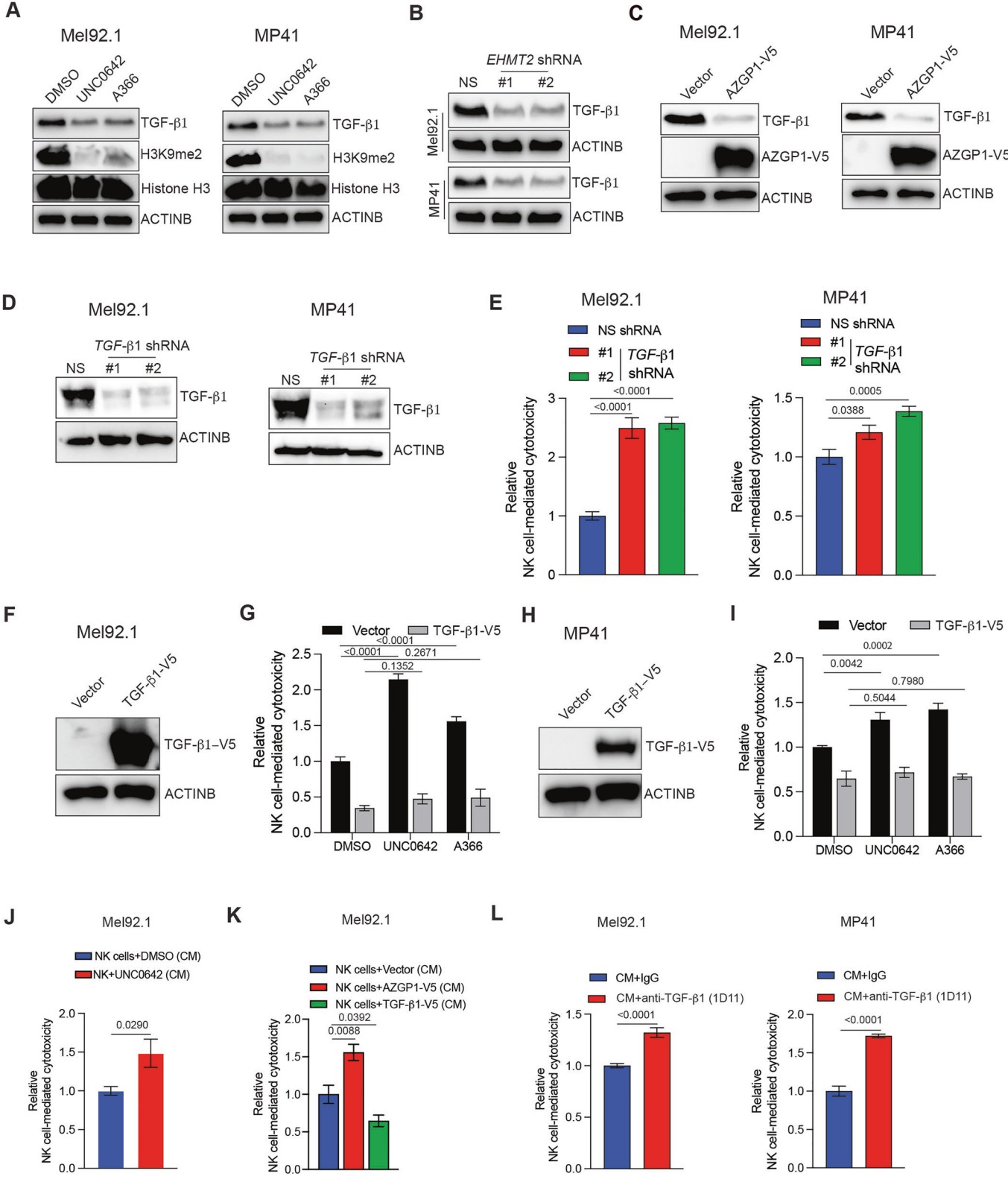

Figure 3. AZGP1 overexpression induces NK cell-mediated cytotoxicity by inhibiting the TGF-β1 signaling following EHMT2 inhibition.

(A) Immunoblotting for measuring expression of the indicated proteins in Mel92.1 and MP41 cells treated with either DMSO or EHMT2 inhibitors UNC0642 (1 μM) or A366 (1 μM) for 48 h. Histone H3 and ACTINB were used as loading controls. (B) Immunoblotting for measuring the expression of TGF-β1 in Mel92.1 and MP41 cells expressing either NS shRNA or EHMT2 shRNAs. ACTINB was used as a loading control. (C) Immunoblotting for measuring the expression of indicated proteins in Mel92.1 and MP41 cells expressing either an empty vector control or V5-tagged AZGP1 ORF. ACTINB was used as a loading control. (D) Immunoblotting for measuring expression of TGF-β1 in Mel92.1 and MP41 cells expressing either NS shRNA or TGF-β1 shRNAs. ACTINB was used as a loading control. (E) Mel92.1 and MP41 cells expressing either NS shRNA or TGF-β1 shRNAs were analyzed for NK cell-mediated cytotoxicity using an LDH-based cytotoxicity assay. Relative NK cell-mediated cytotoxicity under the indicated conditions are plotted. (n = 6). P values were calculated using unpaired two-tailed Student's t-test. (F) Mel92.1 cells expressing either an empty vector or V5-tagged TGF-β1 ORF were analyzed for the expression of TGF-β1 by immunoblotting using V5-tag antibody. ACTINB was used as a loading control. (G) Mel92.1 cells expressing either an empty vector or V5-tagged TGF-β1 ORF were treated with DMSO or EHMT2 inhibitor UNC0642 (1 μM) or A366 for 48 h and analyzed for NK cell-mediated cytotoxicity using LDH-based cytotoxicity assay. Relative NK cell-mediated cytotoxicity at the indicated conditions is plotted. (n = 6). P values were calculated using unpaired two-tailed Student's t-test. (H) MP41 cells expressing either an empty vector or TGF-β1 ORF were analyzed for the expression of TGF-β1 by immunoblotting using V5-tag antibody. ACTINB was used as a loading control. (I) MP41 cells expressing either an empty vector or V5-tagged TGF-β1 ORF were treated with DMSO or EHMT2 inhibitors UNC0642 (1 μM) or A366 (1 μM) for 48 h and analyzed for NK cell-mediated cytotoxicity using LDH-based cytotoxicity assay. Relative NK cell-mediated cytotoxicity at the indicated conditions is plotted. (n = 6). P values were calculated using unpaired two-tailed Student's t-test. (J) Mel92.1 cells treated with DMSO or EHMT2 inhibitor (UNC0642, 1 μM) in Opti-MEM for 48 h. Following this, the conditioned media was collected, concentrated and used to treat NK cells for 24 h. NK cells were analyzed for cytotoxicity against Mel92.1 cells using an LDH-based cytotoxicity assay. Relative NK cell-mediated cytotoxicity at the indicated conditions are plotted. (n = 6). P values were calculated using unpaired two-tailed Student's t-test. (K) Mel92.1 cells expressing either an empty vector, V5-tagged AZGP1 ORF or V5-tagged TGF-β1 ORF were grown in Opti-MEM for 48 h. Following this, the conditioned media was collected, concentrated and used to treat NK cells for 24 h. NK cells were analyzed for cytotoxicity against Mel92.1 cells using an LDH-based cytotoxicity assay. Relative NK cell-mediated cytotoxicity at the indicated conditions is plotted. (n = 5). P values were calculated using unpaired two-tailed Student's t-test. (L) Mel92.1 or MP41 cells were grown in Opti-MEM for 48 h. Following this, the conditioned media was either incubated with control IgG antibody or TGF-β1-neutralizing antibody and used to treat NK cells for 24 h. Following this, NK cell-mediated cytotoxicity was measured against Mel92.1 or MP41 cells using an LDH-based cytotoxicity assay. Relative NK cell-mediated cytotoxicity under the indicated conditions are shown. (n = 6). P values were calculated using unpaired two-tailed Student's t-test. All quantitative data are shown as the mean ± SEM. Source data are available online for this figure.

TGF-β1-expressing UM cell line Mel92.1 and measured NK cell-mediated cytotoxicity. As expected, conditioned media collected from EHMT2 inhibitor-treated or AZGP1-expressing UM cells were less inhibitory to NK cell-mediated cytotoxicity compared to those collected from the DMSO-treated or empty vector-expressing UM cells (Fig. 3J,K). Conversely, conditioned media collected from TGF-β1-expressing UM cells suppressed NK cell-mediated cytotoxicity more potently compared to conditioned media from empty vector-expressing UM cells (Fig. 3K). To more conclusively show the role of TGF-β1 in suppressing NK cell-mediated cytotoxicity in a paracrine manner, we neutralized TGF-β1 using the monoclonal TGF-β1-neutralizing antibody 1D11 (Dasch et al, 1989) in the conditioned media collected from UM cells. We found that the conditioned media treated with TGF-β1-neutralizing antibody was no longer as effective in suppressing NK cell-mediated cytotoxicity as the conditioned media collected from UM cells treated with control IgG (Fig. 3L). We also examined the effects of conditioned media collected from UM cells treated with the EHMT2 inhibitor UNC0642 and concurrently neutralized for TGF-β1 using the 1D11 antibody. Notably, simultaneous suppression of EHMT2 and neutralization of TGF-β1 was more effective in enhancing NK cell-mediated cytotoxicity against UM cells compared to either condition alone (Fig. EV3A,B), likely due to more efficient suppression of TGF-β1-driven immunosuppressive signaling. These results demonstrate that TGF-β1 suppresses NK cell-mediated UM cell eradication.

## The EHMT2-AZGP1-TGF-β1 pathway suppresses NK cell-mediated cytotoxicity by downregulating NKG2D ligands on cancer cells and by modulating the regulators of cytotoxicity in NK cells

We next aimed to determine how TGF-β1, downstream of EHMT2, operates to suppress NK cell-mediated cancer cell eradication. Numerous studies have documented the expression of NKG2D ligands in different cancer types to be important in NK cell-mediated tumor eradication (Bugide et al, 2018a; Cho et al, 2014; Friese et al, 2003; Salih et al, 2003; Vetter et al, 2002; Watson et al, 2006). Furthermore, it has been established that the quantitative level of NKG2D ligand expression defines the "threat" level of target cells, and that it correlates with NK cell-dependent tumor elimination (Diefenbach et al, 2001; Raulet et al, 2013). Based on these previous reports, to understand the mechanism behind suppression of NK cell-mediated cytotoxicity, we tested the effect of EHMT2 inhibition or TGF-β1 knockdown on NKG2D ligands. We found that the expression of NKG2D ligands ULBP3 and MICB were consistently upregulated following treatment with EHMT2 inhibitors in UM cells (Figs. 4A and EV3C–F; Appendix Fig. S3A). A similar upregulation of ULBP3 and MICB was observed following TGF-β1 knockdown in UM cells (Fig. 4B; Appendix Fig. S3B). Furthermore, ectopic expression of TGF-β1 prevented the re-expression of ULBP3 and MICB in UM cells following EHMT2 inhibitor treatment (Fig. 4C; Appendix Fig. S4). These results demonstrated that EHMT2, in a TGF-β1-dependent manner, suppresses the expression of the NKG2D ligands ULBP3 and MICB.

We next asked if the increased expression of NKG2D ligands following EHMT2 inhibition was necessary for enhanced NK cell-mediated cytotoxicity. To test this, we knocked down the expression of ULBP3 and MICB using shRNAs and asked if the loss of either of them resulted in reduced NK-mediated cytotoxicity against UM cells following EHMT2 inhibitor treatment. We found that the knockdown of MICB prevented the increase in NK cell-mediated cytotoxicity when EHMT2 was inhibited (Fig. 4D,E). While knockdown of ULBP3 effectively prevented NK-mediated cytotoxicity against the UM cell line MP41, only one out to the two shRNAs against ULBP3 inhibited NK-mediated cytotoxicity against Mel92.1 cells and the other shRNA was partly effective in blocking NK cell-mediated cytotoxicity (Fig. 4D; Appendix Fig. S5). These results demonstrate that the re-expression of MICB and, in some cases re-expression of ULBP3 following EHMT2

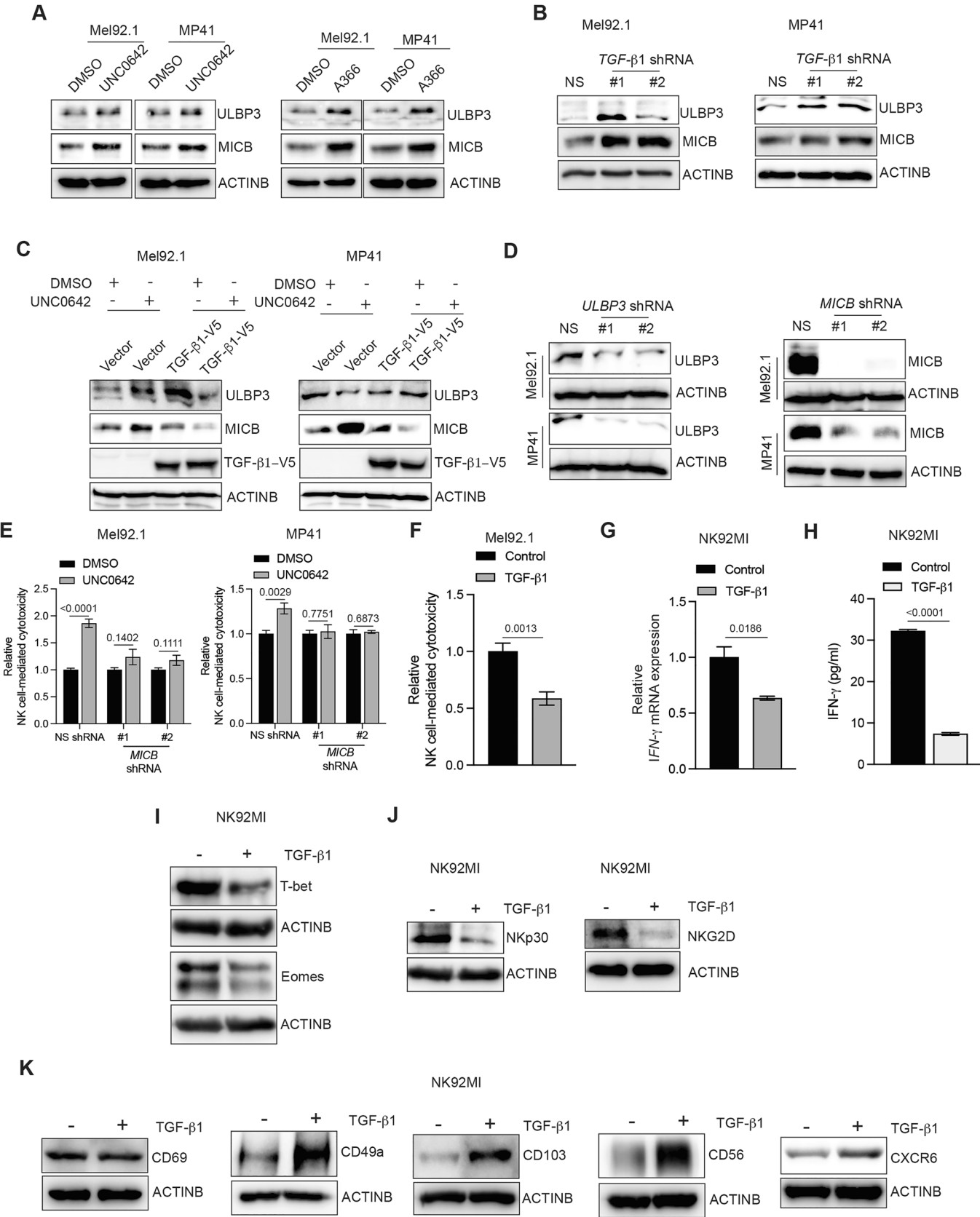

**Figure 4.   The EHMT2-AZGP1-TGF-β1 pathway suppresses NK cell-mediated cytotoxicity by downregulating NKG2D ligands on cancer cells, and by modulating regulators of cytotoxicity in NK cells.**

(A) Mel92.1 and MP41 cells were treated with DMSO, UNC0642 (1 μM), or A366 (1 μM) for 48 h and analyzed for the indicated proteins using immunoblotting. ACTINB was used as a loading control. (B) Mel92.1 and MP41 cells expressing either nonspecific (NS) shRNA or *TGF-β1* shRNAs were analyzed for the indicated proteins using immunoblotting. ACTINB was used as a loading control. (C) Mel92.1 and MP41 cells expressing either an empty vector or V5-tagged *TGF-β1* ORF were treated with DMSO or UNC0642 (1 μM) for 48 h and analyzed for the indicated protein using immunoblotting. ACTINB was used as a loading control. (D) Mel92.1 and MP41 cells expressing either NS shRNA, *ULBP3* shRNAs, or *MICB* shRNAs were analyzed for the indicated proteins using immunoblotting. ACTINB was used as a loading control. (E) Mel92.1 and MP41 cells expressing either NS shRNA or *MICB* shRNAs were treated with DMSO or EHMT2 inhibitors UNC0642 (1 μM) for 48 h and were analyzed for NK cell-mediated cytotoxicity using an LDH-based cytotoxicity assay. Relative NK cell-mediated cytotoxicity under the indicated conditions are plotted. (n = 6). P values were calculated using unpaired two-tailed Student's t-test. (F) NK92MI cells were treated with TGF-β1 (10 ng/ml) for 24 h followed by treated NK cells were used for NK cell-mediated cytotoxicity assay against Mel92.1 Cells. Relative NK cell-mediated cytotoxicity under indicated conditions are plotted. (n = 6). P values were calculated using unpaired two-tailed Student's t-test. (G) NK92MI cells were treated with TGF-β1 (10 ng/ml) for 24 h and were analyzed for *IFN-γ* mRNA levels using RT-qPCR. Relative *IFN-γ* mRNA levels are presented under the indicated conditions. *ACTINB* was used for normalization. (n = 3). P values were calculated using unpaired two-tailed Student's t-test. (H) NK92MI cells were treated with TGF-β1 (10 ng/ml) for 24 h and analyzed for IFN-γ protein levels using ELISA. IFN-γ protein levels are presented under the indicated conditions. (n = 3). P values were calculated using unpaired two-tailed Student's t-test. (I) NK92MI cells were treated with TGF-β1 (10 ng/ml) for 24 h and analyzed for EOMES and T-Bet protein levels using immunoblotting. ACTINB was used as a loading control. (J) NK92MI cells were treated with TGF-β1 (10 ng/ml) for 24 h and analyzed for NCR3 and NKG2D protein using immunoblotting. ACTINB was used as a loading control. (K) NK92MI cells were treated with TGF-β1 (10 ng/ml) for 24 h and analyzed for CD69, CD49a, CD103, CD56, and CXCR6 proteins using immunoblotting. ACTINB was used as a loading control. All quantitative data are shown as the mean ± SEM. Source data are available online for this figure.

inhibition is necessary for enhanced NK cell-mediated cytotoxicity against UM cells.

Next, we asked how TGF-β1 suppresses the cytotoxic function of NK cells. To investigate this, we analyzed the downstream targets of TGF-β1 in NK cells that have been shown to regulate NK cell function (Castriconi et al, 2003; Laouar et al, 2005; Viel et al, 2016; Yu et al, 2006). We treated NK cells with recombinant TGF-β1 and then analyzed the expression of IFN-γ, NCR3 and transcription factors T-bet and Eomes. IFN-γ induces cell death in cancer cells, enhances the cytotoxic function of other immune cells, such as cytotoxic T cells, and promotes the recruitment of other immune effectors to the tumor microenvironment (Bhat et al, 2017). We found that recombinant TGF-β1 treatment resulted in reduced NK cell-mediated cytotoxicity (Fig. 4F) and reduced IFN-γ levels in NK cells (Fig. 4G,H). The T-box family of transcription factors such as T-box expressed in T cells (T-bet) and Eomesodermin (Eomes) are important players in NK cell functions (Daussy et al, 2014; Gordon et al, 2012). NK cells lacking T-bet and Eomes show reduced cytotoxic activity (Gordon et al, 2012). Consistent with this, we also found reduced expression of T-bet and Eomes in NK cells treated with recombinant TGF-β1 (Fig. 4I; Appendix Fig. S6A). We also analyzed the expression of NKp30 and NKG2D following recombinant TGF-β1 treatment. NKp30 protein (encoded by the *NCR3* gene) is one of the three major natural cytotoxicity receptors in NK cells, along with NKp44 and NKp46 (Bryceson and Long, 2008; Moretta et al, 2001). NKp30 is a key mediator of NK cell activity and plays a non-redundant role in the killing of cancer cells (Brandt et al, 2009; Delahaye et al, 2011). Furthermore, NKG2D is an activating receptor expressed on NK cells, playing a crucial role in immune surveillance by recognizing stress-induced ligands on infected or transformed cells (Bauer et al, 1999). NKG2D triggers cytotoxic responses and cytokine production upon ligand engagement, contributing to anti-tumor immunity (Gilfillan et al, 2002). NKG2D ligands such as ULBP3 and MICB bind to NKG2D, which results in NK cell activation and cytotoxicity (Mou et al, 2014; Zhang et al, 2023). The expression of *NCR3* mRNA and NKp30 protein was also reduced in NK cells following recombinant TGF-β1 treatment (Fig. 4J; Appendix Fig. S6B). We found that, similar to NKp30, NKG2D expression was suppressed following treatment with TGF-β1 (Fig. 4J; Appendix Fig. S6B). Previous studies have

also shown that conversion of effector NK cells into type 1 innate lymphoid cells (ILC1) alters their cytotoxic potential, as effector NK cells are highly cytotoxic and release perforin and granzymes to lyse target cells, whereas ILC1 are typically non-cytotoxic (Lopes et al, 2023). Furthermore, NK cells and ILC1 cells differ in their expression of various cell surface markers (Lopes et al, 2022), and TGF-β1 promotes the conversion of NK cells to ILC1 cells (Cortez et al, 2017; Gao et al, 2017). Therefore, we treated NK92MI cells with TGF-β1 and measured the expression of these markers. We found that treatment of NK92MI cells with TGF-β1 resulted in increased expression of CD49a and CD103, while CD69 expression remained unchanged (Fig. 4K; Appendix Fig. S6C). A similar increase in CD56 and CXCR6 was observed in NK92MI cells (Fig. 4K; Appendix Fig. S6C). These results showcase that TGF-β1 upregulates factors that inhibit the cytotoxic activity of NK cells.

Immune cell function, such as that of NK cells, can also be regulated by excluding them from the tumor microenvironment by preventing their migration (Cozar et al, 2021). This can occur via several different mechanisms, including by modulating the expression of chemokines that, in turn, can influence NK cell migration. Therefore, we first performed a human chemokine array. The UM cell line Mel92.1 was first treated with EHMT2 inhibitor UNC0642 or, as a control, with DMSO for 72 h. Conditioned media were then collected and hybridized to human chemokine antibody array membranes, which can detect changes in the expression of 38 human chemokines (Fig. 5A). We identified several chemokines that were increased in response to pharmacological inhibition of EHMT2 by UNC0642 (Fig. 5B,C). To further validate and generalize these results, we treated UM cells with EHMT2 inhibitors UNC0642 and A366, and measured mRNA expression levels of the chemokines identified by the human chemokine array using RT-qPCR in Mel92.1 and MP41 cell lines. We found that *CXCL5, CCL15, CXCL10, CCL27,* and *CXCL8* was consistently upregulated in both UM cell lines in response to treatment with both EHMT2 inhibitors (Figs. 5D,E and EV4A,B), whereas the TGF-β1 inhibitor vactosertib resulted in the upregulation of only *CCL27* and *CXCL10* in Mel92.1 and MP41 cell lines (Figs. 5F,G and EV4C,D). Furthermore, conditioned media collected from UM cell lines (Mel92.1 and MP41) treated with UNC0642 (Fig. 5H) or vactosertib showed improved NK cell

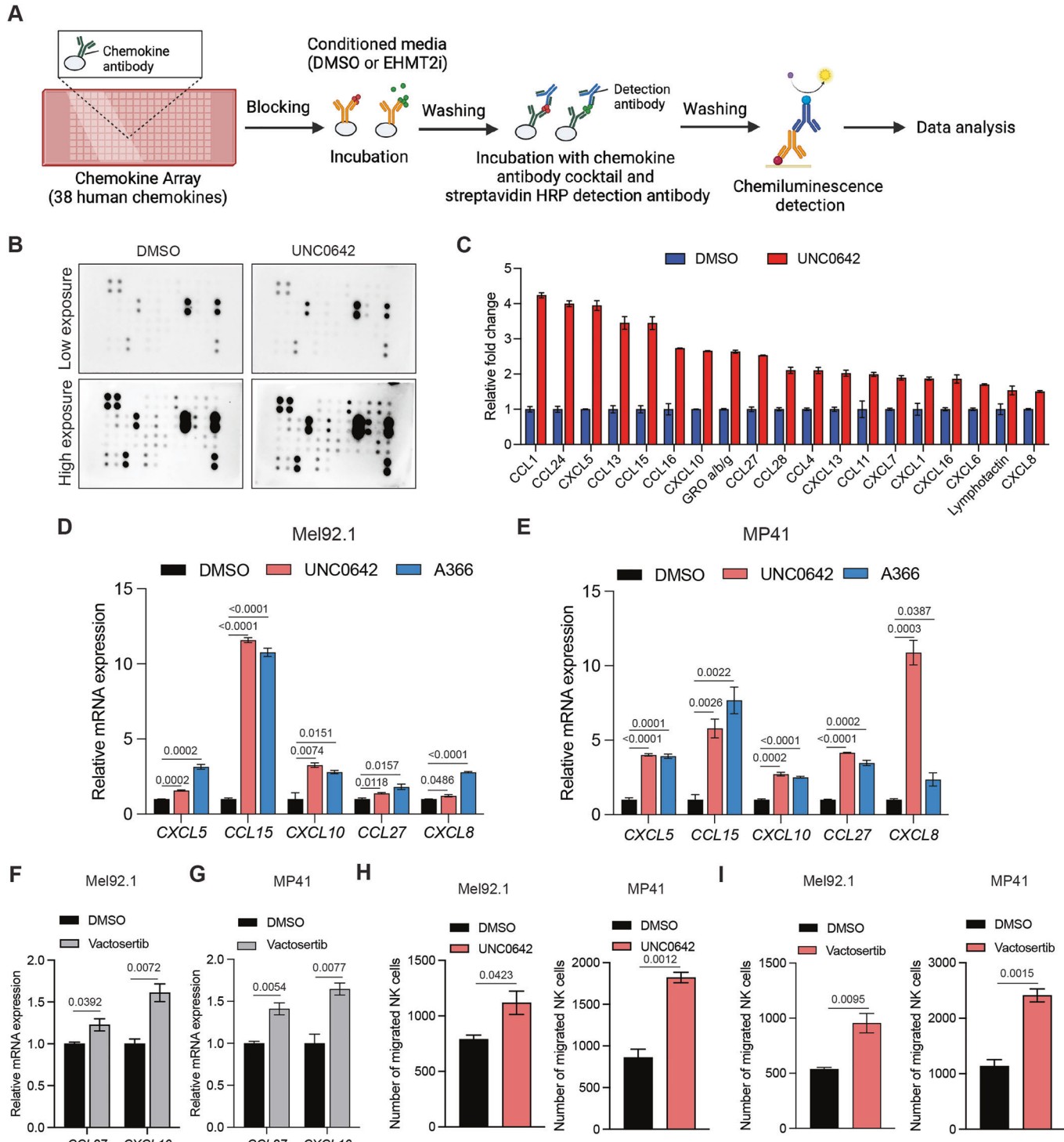

migration (Fig. 5I), indicating a direct role of TGF-β1 in suppressing NK cell migration. Collectively, these results demonstrate that EHMT2-mediated TGF-β1 upregulation in an autocrine manner, inhibits NKG2D ligands (MICB and ULBP3) on cancer cells and, in a paracrine manner, suppresses the cytotoxic ability of NK cells and their migration. These functions together contribute to the downstream suppressive effects of EHMT2 on NK cell antitumor immunity.

## EHMT2 loss also enhances NK cell-mediated eradication of other cancer types

We next asked if, similar to UM cells, the knockdown of EHMT2 can enhance NK cell-mediated eradication of other cancer cell types. We tested the impact of EHMT2 knockdown in triple-negative breast cancer (TNBC) cells, a cancer type that is not very responsive to adaptive immune checkpoint blockade therapies

**Figure 5. Inhibition of EHMT2 and TGF-β1 enhances the expression of chemokines that promote NK cell migration.**

(A) Schematic for identifying human chemokines altered as a result of EHMT2 inhibition. (B) Mel92.1 cells were treated with DMSO or UNC0642 (1 μM) for 72 h. Following this, conditioned media (CM) was collected and analyzed for the expression of 38 different human chemokines using human chemokine antibody arrays. (C) Average fold change in expression of indicated chemokines, quantified from the image shown in panel (B), using ImageJ software. (D, E) Mel92.1 and MP41 cells were treated with DMSO or UNC0642 (1 μM) or A366 (1 μM) for 72 h. mRNA expression for the indicated mRNAs were analyzed using RT-qPCR analysis. Relative mRNA expression for the indicated genes are plotted. *ACTINB* was used for normalization. ($n = 3$). *P* values were calculated using unpaired two-tailed Student's *t*-test. (F, G) Mel92.1 and MP41 cells were treated with DMSO or vactosertib (1 μM) for 24 h. mRNA expression for the indicated chemokines were analyzed using RT-qPCR analysis. Relative mRNA expression for the indicated chemokines are plotted. *ACTINB* was used for normalization. ($n = 3$). *P* values were calculated using unpaired two-tailed Student's *t*-test. (H) Mel92.1 and MP41 cells were treated with DMSO or UNC0642 (3 μM) for 72 h and NK cell migration was evaluated using FACS analysis. The number of migrated NK cells under the indicated conditions are plotted. ($n = 3$). *P* values were calculated using unpaired two-tailed Student's *t*-test. (I) Mel92.1 and MP41 cells were treated with DMSO or vactosertib (1 μM) for 72 h and NK cell migration was evaluated using FACS analysis. The number of migrated NK cells under the indicated conditions are plotted. ($n = 3$). *P* values were calculated using unpaired two-tailed Student's *t*-test. All quantitative data are shown as the mean ± SEM. Source data are available online for this figure.

(anti-PD-1, anti-CTLA4, etc.) (Adams et al, 2019). We found that TNBC cells treated with EHMT2 inhibitors (A366 and UNC0642) showed increased sensitivity to NK cell-mediated cytotoxicity (Fig. 6A). Furthermore, consistent with the results in UM cells, TNBC cells treated with EHMT2 inhibitors (A366 and UNC0642) showed AZGP1 upregulation and TGF-β1 downregulation (Fig. 6B). Additionally, we tested the impact of EHMT2 inhibition on NK cell-mediated pancreatic cancer eradication, which is another cancer type for which adaptive immune checkpoint blockade therapies have not worked as desired (Royal et al, 2010). Human pancreatic cancer cell lines were treated with EHMT2 inhibitors, and NK cell-mediated eradication was analyzed. We found that, in the case of pancreatic cancer also, treatment with EHMT2 inhibitors (A366 and UNC0642) resulted in increased NK cell-mediated cytotoxicity (Fig. 6C), AZGP1 upregulation, and TGF-β1 downregulation (Fig. 6D). These results demonstrate that EHMT2 inhibition results in increased NK cell-mediated cytotoxicity and AZGP1 upregulation in both TNBC and pancreatic cancer cells.

## Loss of EHMT2 drives strong tumor suppression in immunocompetent but not in immunodeficient mouse models

Immunocompetent, syngeneic mouse models to study host immune response to UM are not available to evaluate the in vivo impact of EHMT2 inhibition. Therefore, we used the TNBC and pancreatic cancer-based syngeneic models; two cancer types in which, similar to UM, we found that EHMT2 inhibition resulted in increased NK cell-mediated cytotoxicity. We tested the EMT6 syngeneic model of TNBC, which has been used by several other groups to identify regulators of the host immune response (Kirchhammer et al, 2022; Piranlioglu et al, 2019). The expression of *Ehmt2* was knocked down in EMT6 cells (Fig. EV5A,B), and NS shRNA-expressing EMT6 cells were used as a negative control. EMT6 cells were orthotopically injected into the mammary fat pads of either immunocompetent syngeneic BALB/c mice or immunodeficient NSG mice. We found that *Ehmt2* knockdown enhanced EMT6-driven tumor clearance in syngeneic BALB/c mice (Fig. 6E,F). We then analyzed immune cell infiltration in EMT6 tumors expressing *Ehmt2* shRNA compared to those expressing NS shRNA. NK cell and T-cell infiltration was assessed, revealing that *Ehmt2* shRNA-expressing tumors exhibited significantly higher levels of infiltrated NK cells (Fig. 6G), with no significant difference in CD8[+] effector T-cells and CD4[+] T-cells, compared to tumors with NS shRNA

(Fig. EV5C,D). Notably, although some reduction in EMT6 tumor burden was observed in immunodeficient NSG mice that lack T, B, and NK cells, this was less effective as compared to what was observed in syngeneic mice (Fig. 6H,I). Furthermore, consistent with strong tumor suppression, we noted reduced Ki-67 staining in EMT6 tumors expressing *Ehmt2* compared to those expressing NS shRNA from immunocompetent syngeneic BALB/c mice (Fig. EV5E,F).

Finally, we also analyzed ILC1 and effector NK cell populations in EMT6 tumors expressing *Ehmt2* shRNA or NS shRNA. We found that ILC1 (Lin[−]CD49a[+]CD49b[−]) were significantly reduced in the *Ehmt2* shRNA group compared to the NS shRNA group. Conversely, effector NK cells (Lin[−]CD49a[−]CD49b[+]) were significantly increased in the *Ehmt2* shRNA group relative to the NS shRNA group (Fig. EV5G).

We then tested the impact of *Ehmt2* knockdown using the pancreatic cancer cell line Panc02-based syngeneic mouse model. The Panc02 model has been used previously by various groups to study immune regulation (Bazhin et al, 2022; Kinkead et al, 2018). We knocked down the expression of *Ehmt2* using shRNA in Panc02 cells and used NS shRNA-expressing Panc02 cells as a negative control (Fig. EV6A,B). *Ehmt2* shRNA or NS shRNA-expressing cells were injected subcutaneously into the syngeneic C57BL/6 mice or into immunodeficient NSG mice. We found that *Ehmt2* knockdown in Panc02 resulted in reduced tumor growth in C57BL/6 mice (Fig. 6J,K). We also observed increased NK cells infiltration in tumors (Fig. 6L) but no significant change in CD8[+] effector T-cells and CD4[+] T-cells, in *Ehmt2* knockdown expressing Panc02 tumors compared to tumors with NS shRNA (Fig. EV6C,D). Furthermore, no tumor suppression was observed in immunodeficient NSG mice (Fig. 6M,N). Moreover, consistent with strong tumor suppression, we noted reduced Ki-67 staining in Panc02 tumors expressing *Ehmt2* compared to those expressing NS shRNA from immunocompetent syngeneic C57BL/6 mice (Fig. EV6E,F).

Furthermore, we also analyzed ILC1 and effector NK cell populations in EMT6 tumors expressing *Ehmt2* shRNA or NS shRNA. We found that ILC1 (Lin[−]CD49a[+]CD49b[−]) were significantly reduced in the *Ehmt2* shRNA group compared to the NS shRNA group. Conversely, effector NK cells (Lin[−]CD49a[−]CD49b[+]) were significantly increased in the *Ehmt2* shRNA group relative to the NS shRNA group (Fig. EV6G).

Since *Ehmt2* knockdown inhibited tumor growth, we also asked whether the differences in NK cell infiltration were due to the

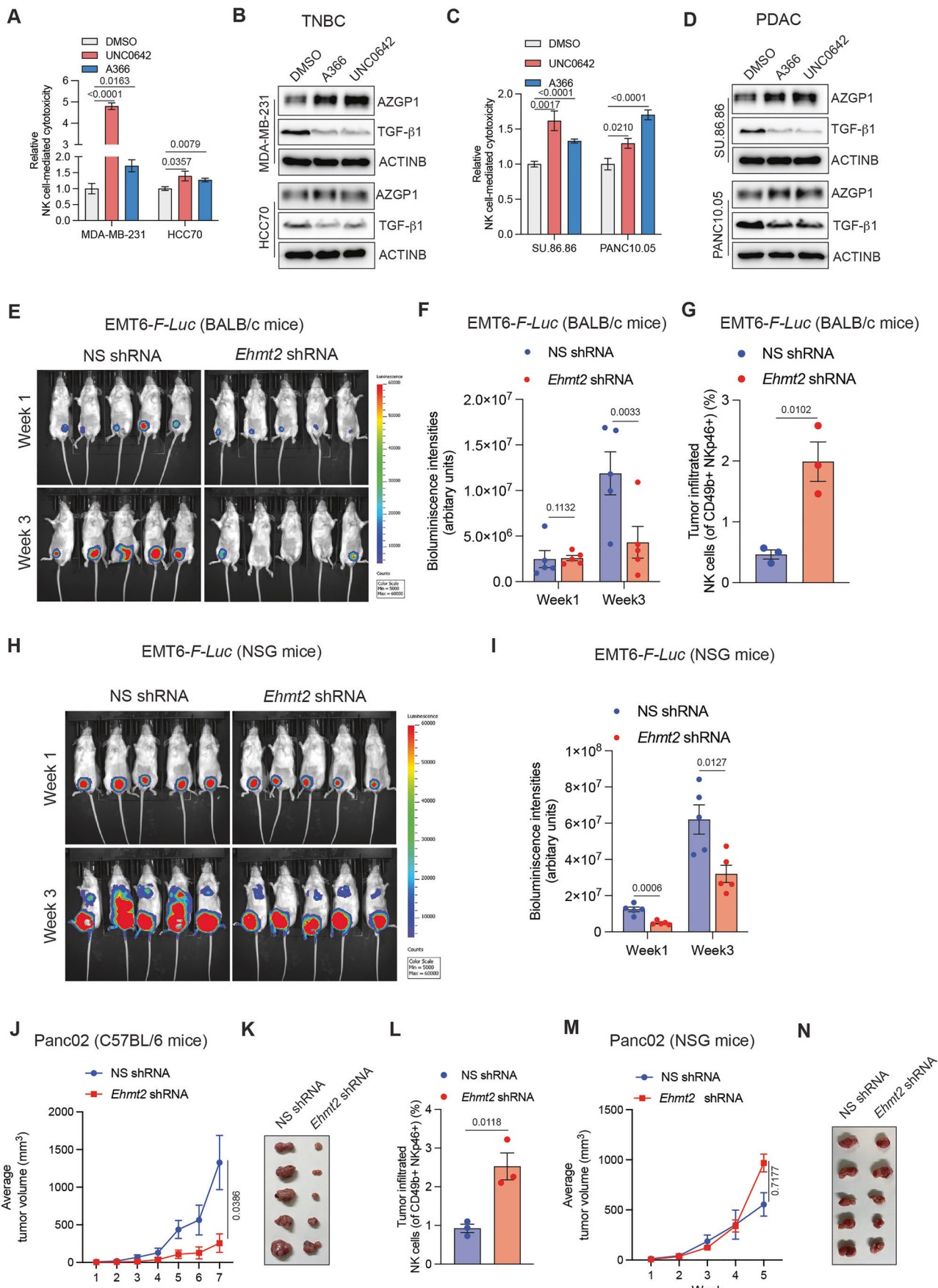

**Figure 6.** *Ehmt2* knockdown-mediated tumor suppression in immunocompetent syngeneic mouse models of breast cancer and pancreatic cancer.

(A) Indicated TNBC cells were treated with DMSO, UNC0642 (1 µM), and A366 (1 µM) for 48 h and were analyzed for NK cell-mediated cytotoxicity using LDH-based cytotoxicity. Relative NK cell-mediated cytotoxicity under indicated conditions. ($n = 6$). $P$ values were calculated using unpaired two-tailed Student's $t$-test. (B) Indicated TNBC cells were treated with DMSO, A366 (1 µM), or UNC0642 (1 µM) for 48 h and analyzed for the indicated proteins by immunoblotting. ACTINB was used as a loading control. (C) Indicated PDAC cells were treated with DMSO, UNC0642 (1 µM), A366 (1 µM) for 48 h and analyzed for NK cell-mediated cytotoxicity using LDH-based cytotoxicity. Relative NK cell-mediated cytotoxicity under indicated conditions. ($n = 6$). $P$ values were calculated using unpaired two-tailed Student's $t$-test. (D) Indicated PDAC cells were treated with DMSO, A366 (1 µM) or UNC0642 (1 µM) for 48 h and analyzed for the indicated proteins by immunoblotting. ACTINB was used as a loading control. (E) Firefly luciferase (*F-Luc*)-labeled EMT6 cells expressing either nonspecific (NS) shRNA or *Ehmt2* shRNA were orthotopically injected into the mammary fat pad of female BALB/c mice ($n = 5$). Bioluminescence images of mice at the indicated weeks after injection are shown. (F) Bioluminescence intensities of the mice at the indicated weeks from panel (E). ($n = 5$). $P$ values were calculated using unpaired two-tailed Student's $t$-test. (G) Measurement of tumor-infiltrated NK cells (%) in the EMT6-*F-Luc* tumors under the indicated conditions using FACS analysis. ($n = 3$). $P$ values were calculated using unpaired two-tailed Student's $t$-test. (H) *F-Luc*-labeled EMT6 cells expressing either NS shRNA or *Ehmt2* shRNA were orthotopically injected into the mammary fat pads of female NSG mice ($n = 5$). Bioluminescence images of mice at the indicated weeks after injections are shown. (I) Bioluminescence intensities of the mice at the indicated weeks from panel (H). ($n = 5$). $P$ values were calculated using unpaired two-tailed Student's $t$-test. (J) Panc02 cells expressing either NS shRNA or *Ehmt2* shRNA were injected subcutaneously into C57BL/6 mice. Tumor volumes at the indicated times. ($n = 5$). For the analysis of tumor progression in mice, the statistical assessment was performed using the area under the curve method, followed by unpaired two-tailed Student's $t$-tests. (K) Tumor images for the experiment presented in panel (J) under the indicated conditions. (L) Measurement of tumor-infiltrated NK cells (%) in the Panc02 tumors under the indicated conditions using FACS analysis. ($n = 3$). $P$ values were calculated using unpaired two-tailed Student's $t$-test. (M) Panc02 cells expressing either NS shRNA or *Ehmt2* shRNA were injected subcutaneously into the flank of NSG mice. Average tumor volumes at the indicated times. ($n = 5$). For the analysis of tumor progression in mice, the statistical assessment was performed using the area under the curve method, followed by unpaired two-tailed Student's $t$-tests. (N) Tumor images for the experiment presented in panel (M) under the indicated conditions. All quantitative data were shown as the mean ± SEM. Source data are available online for this figure.

differences in tumor volumes or were driven by the loss of Ehmt2 expression. To test this, we injected Panc02 cells expressing either *Ehmt2* shRNA or, as a control, NS shRNA and collected tumors when volume differences were not significant (Fig. EV6H). We found that even before tumor volumes became significantly different, the *Ehmt2* shRNA tumor group showed a higher number of infiltrating NK cells compared to the NS shRNA tumor group (Fig. EV6I). Collectively, these results demonstrate that *Ehmt2* loss promotes tumor suppression in the presence of the host immune system.

## NK cells are essential for tumor suppression induced by EHMT2 knockdown in immunocompetent mouse models

Next, to determine the role of NK cells in mediating the Ehmt2 loss-induced tumor suppression in the EMT6 model, we depleted the NK cells in BALB/c mice using anti-Asialo-GM1 antibody. An isotype control IgG antibody was used as a control. These mice were then injected with EMT6 cells expressing *Ehmt2* shRNA or NS shRNA orthotopically into the mammary fat pad. We found that *Ehmt2* knockdown enhanced EMT6-driven tumor clearance in syngeneic BALB/c mice, which was rescued following NK cell depletion (Fig. EV7A,B). As expected, Ehmt2 loss promoted NK cell infiltration as observed by an increased number of NK cells in the *Ehmt2* shRNA tumor group compared to the NS shRNA-expressing tumor group (Fig. EV7C). Furthermore, as expected, anti-Asialo-GM1 antibody efficiently depleted NK cells in mice (Fig. EV7D).

To more fully establish the role of NK cells in *Ehmt2*-loss-mediated tumor suppression, we used a Panc02-based pancreatic cancer model propagated in C57BL/6 mice. In C57BL/6 mice, depletion using an anti-NK1.1 antibody typically results in minimal nonspecific immune modulation, allowing for more accurate interpretation of the role of NK cells in experimental models. C57BL/6 mice were injected subcutaneously with Panc02 cells, and an anti-NK1.1 antibody was used to immunodeplete the NK cells. An IgG isotype-matched antibody was injected into the control group of C57BL/6 mice. NK cell depletion restored the growth of

Panc02 tumors expressing *Ehmt2* shRNAs (Fig. 7A,B). As expected, anti-NK1.1 antibody treatment resulted in NK cell depletion in mice (Fig. 7C). *Ehmt2* shRNA-expressing tumors exhibited significantly higher levels of infiltrated NK cells compared to tumors expressing NS shRNA (Fig. 7D). These results collectively demonstrate that NK cells play an essential role in Ehmt2 loss-driven tumor suppression.

Our data described above demonstrate that *Ehmt2* knockdown-mediated tumor suppression is dependent on NK cell activity. To further determine the importance of NK cells in *Ehmt2* knockdown-mediated tumor growth inhibition in vivo, we injected *Ehmt2* knockdown Panc02 cells into *Rag2* knockout (*Rag2* KO) mice, which lack T and B cells but have NK cells. *Rag2* KO mice were then treated with anti-NK1.1 or an isotype control IgG antibody as a control. We found that *Ehmt2* knockdown resulted in significantly reduced tumor growth inhibition in *Rag2* KO mice (Fig. 7E,F). Furthermore, NK cell depletion in these mice restored tumor growth of Panc02 tumors expressing *Ehmt2* shRNAs (Fig. 7E–G). These results demonstrate that NK cells, even in the absence of T and B cells, can promote *Ehmt2* knockdown-driven tumor suppression.

## Pharmacological inhibition of EHMT2 or TGF-β1 promotes tumor eradication in syngeneic tumor models

Finally, we tested pharmacological inhibition of EHMT2 or TGF-β1 as an approach to enhance anti-tumor immunity against tumor cells. We found that both UNC0642 and A366 were less suitable for in vivo studies, resulting in significant toxicity in mice during the course of treatment. Therefore, we used the newly developed BRD4770 that showed no mortality in our pilot studies and demonstrated in vivo activity in a previously published study (Yuan et al, 2012). Similar to UNC0642 and A366, BRD4770 treatment of cancer cells enhanced their NK cell-mediated cytotoxicity (Fig. EV8A), stimulated the expression of AZGP1 (Fig. EV8B,C), and suppressed the expression of TGF-β1 (Fig. EV8D,E). Furthermore, the expression of NKG2D ligand MICB (Fig. EV8F,G) and chemokines CXCL10 and CCL27 (Fig. EV8H) was increased

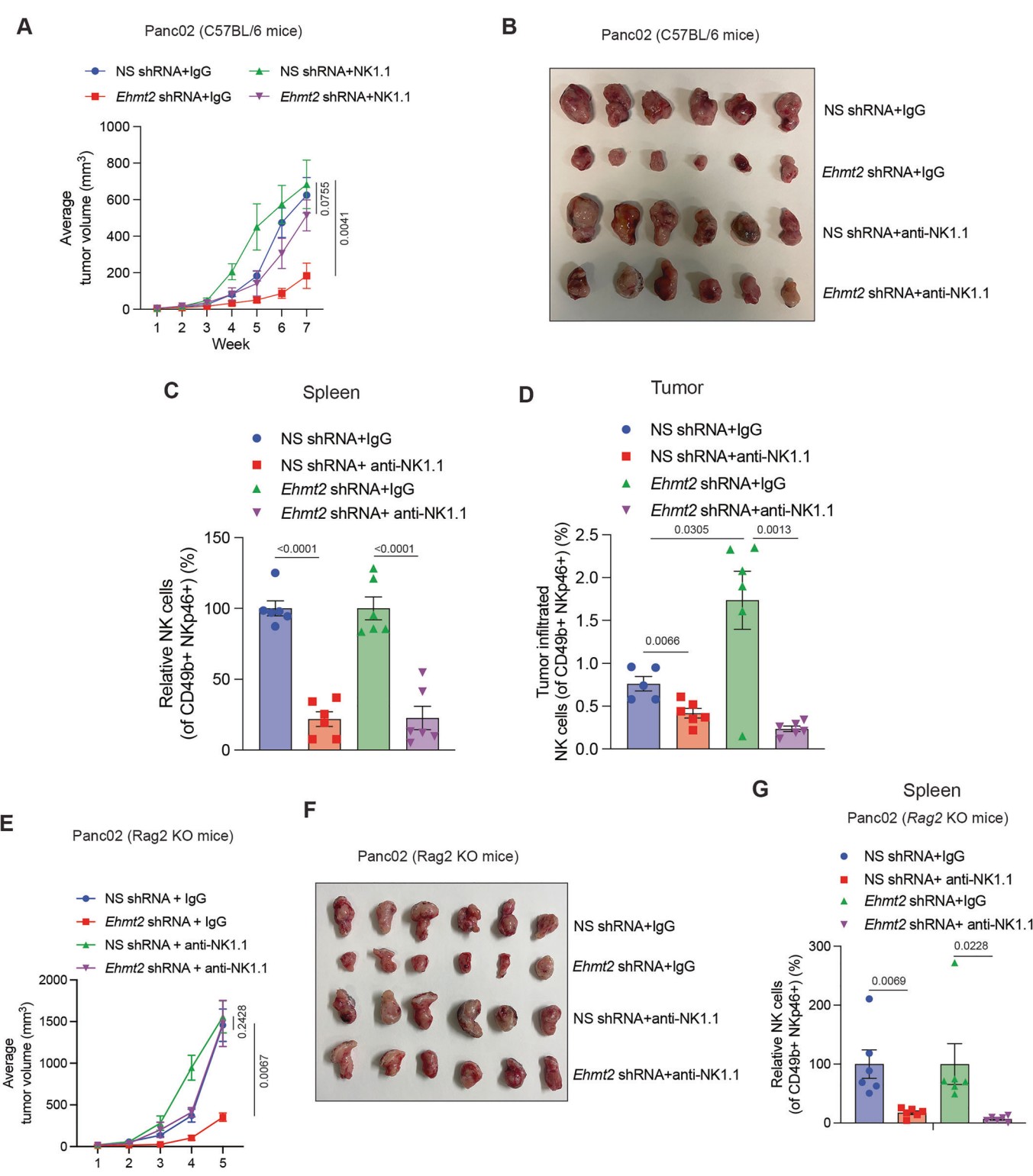

following treatment of cancer cells with BRD4770. Furthermore, consistent with our results from *Ehmt2* knockdown-based studies, we found that BRD4770 blocked the growth of Panc02 tumors in C57BL/6 mice (Fig. 8A,B). Furthermore, BRD4770-treated tumors also showed increased NK cell infiltration (Fig. 8C) and, as

expected, a decrease in the level of H3K9me2 (Fig. 8D). To further establish the suitability of BRD4770, we performed a detailed toxicity analysis of BRD4770-treated mice and found no significant impact on the body weight of mice (Appendix Fig. S7A), serum levels of alanine aminotransferase (ALT), aspartate

**Figure 7. NK cells are necessary for EHMT2 inhibition-mediated tumor suppression in immunocompetent syngeneic mouse models.**

(A) Panc02 cells expressing either Non-specific (NS) shRNA or *Ehmt2* shRNA were injected subcutaneously into C57BL/6 mice and treated with either IgG isotype control or anti-NK1.1 antibodies. Average tumor volumes at the indicated time points. ($n = 6$). For the analysis of tumor progression in mice, the statistical assessment was performed using the area under the curve method, followed by unpaired two-tailed Student's *t*-tests. (B) Tumor images for the experiment presented in panel (A) is shown under the indicated conditions. (C) Measurement of NK cells (%) in spleens from mice ($n = 6$) treated with IgG isotype control or anti-NK1.1 antibodies expressing either NS shRNA or *Ehmt2* shRNA using FACS analysis. ($n = 6$). P values were calculated using unpaired two-tailed Student's *t*-test. (D) Measurement of tumor-infiltrated NK cells (%) in Panc02 tumors treated with IgG isotype control or anti-NK1.1 antibodies expressing either NS shRNA or *Ehmt2* shRNA using FACS analysis. [($n = 5$ for NS shRNA + IgG and $n = 6$ for NS shRNA + anti-NK1.1, *Ehmt2* shRNA + IgG, and *Ehmt2* shRNA + anti-NK1.1)]. P values were calculated using unpaired two-tailed Student's *t*-test. (E) *Rag2* KO mice were injected with Panc02 cells expressing either NS shRNA or *Ehmt2* shRNA and were treated with either control IgG antibody or anti-NK1.1 antibody for NK cell depletion. Tumor volumes were measured at the indicated weeks. ($n = 6$). For the analysis of tumor progression in mice, the statistical assessment was performed using the area under the curve method, followed by unpaired two-tailed Student's *t*-tests. (F) Representative tumor images for the experiment shown in panel (E) under the indicated conditions. (G) Measurement of NK cells (%) in spleens from mice treated with IgG isotype control or anti-NK1.1 antibodies expressing either NS shRNA or *Ehmt2* shRNA using FACS analysis. ($n = 6$). P values were calculated using unpaired two-tailed Student's *t*-test. All quantitative data were shown as the mean ± SEM. Source data are available online for this figure.

aminotransferase (AST) and creatinine (Appendix Fig. S7B–D) and major organ morphologies (Appendix Fig. S7E–G) in all three mouse stains (C57BL/6, BALB/c, and NSG) used in our studies. Moreover, neither *EHMT2* knockdown, *EHMT2* knockout, nor treatment with UNC0642, A366, or BRD4770 affected cancer cell growth in a clonogenic assay, indicating that the impact of EHMT2 on tumor growth is largely NK cell-driven (cancer cell-extrinsic) (Appendix Fig. S8A–C).

Based on this, we also asked if, similar to EHMT2 inhibition, inhibition of TGF-β1, could induce a similar phenotype. Therefore, we used the TGF-β1 inhibitor vactosertib. Vactosertib specifically binds to the kinase domain of TGF-βR1 (Jin et al, 2014; Son et al, 2014; Yoon et al, 2013). By doing so, it blocks the receptor's phosphorylation and activation by TGF-βR2 (Jin et al, 2014; Son et al, 2014; Yoon et al, 2013). Vactosertib is undergoing multiple phase I and II clinical trials, often in combination with other treatments such as immune checkpoint inhibitors (e.g., PD-1/PD-L1 inhibitors) (Ahn et al, 2024; Malek et al, 2024). We found that the vactosertib blocked Panc02 tumor growth in C57BL/6 mice (Fig. 8E,F). Furthermore, vactosertib-treated tumors also showed increased NK cell infiltration (Fig. 8G). Finally, consistent with the important role of immune cells in the observed tumor suppression from EHMT2 inhibition and TGF-β1 inhibition in immunocompetent mice, treatment with either BRD4770 (Fig. 8H,I) or vactosertib (Fig. 8J,K) failed to inhibit tumor growth in immunodeficient NSG mice. Collectively, these results demonstrate that, similar to the genetic inhibition of EHMT2, pharmacological inhibition of EHMT2 and TGF-β1 results in tumor suppression and increased NK cell infiltration.

## Discussion

Cancer cells use complex, not fully understood mechanisms, to evade NK cell-mediated immunity. We found that the inhibition of EHMT2 in tumor cells increased AZGP1, which reduced TGF-β1 levels. TGF-β1 was both necessary and sufficient to suppress NK cell-mediated tumor eradication. TGF-β1 in an autocrine manner suppressed activating ligands MICB and ULBP3 on cancer cells, of which MICB was necessary for NK cell-induced cytotoxicity. While TGF-β1 in a paracrine manner also impaired NK cell migration by downregulating the expression of CXCL10

and CCL27 in cancer cells, and reduced NK cell-mediated cytotoxicity by inhibiting the expression of key transcription factors (e.g., Eomes, T-bet), cytokines (e.g., IFN-γ), and receptors like NKp30, NKG2D, and others in NK cells. In mouse models, EHMT2 loss suppressed tumors in immunocompetent and *Rag2* KO mice (NK cell-dependent, lacking B- and T-cells), but not in immunodeficient mice. Finally, EHMT2 and TGF-β1 inhibitors reduced tumor growth, identifying EHMT2 as a promising, druggable target that suppresses NK cell-mediated anti-tumor immunity via TGF-β1-mediated autocrine and paracrine effects (Fig. 8L).

EHMT2 is a histone methyltransferase that transfers the methyl group to the lysine residue of histone H3, specifically to lysine 9 of histone H3, which leads to transcriptional repression (Tachibana et al, 2001). Previous studies have shown the role of EHMT2 in a variety of cellular processes, including embryonic development and cell differentiation (Ciceri et al, 2024; Ideno et al, 2013), DNA repair and cell survival (Yang et al, 2017), immune regulation (Kato et al, 2020; Scheer and Zaph, 2017) and in cancer progression (Dutta et al, 2016; Kato et al, 2020; Kim et al, 2018; Wang et al, 2017; Yang et al, 2024).

In particular, EHMT2 has been shown to play important roles in a wide variety of cancer types. This includes a previous study that identified a role for the NKX3.1-EHMT2-KDM6C in prostate differentiation and showed that disruption of this network led to an increased predisposition to prostate cancer (Dutta et al, 2016). Additionally, other studies have identified the role of EHMT2 in promoting breast cancer (Kim et al, 2018; Wang et al, 2017). Similarly, EHMT2 has been reported to be mutated in cutaneous melanoma and shown to drive melanoma oncogenesis by regulating WNT signaling (Kato et al, 2020). Furthermore, the role of EHMT2 in neuroendocrine transformation in non-small cell lung cancer (Yang et al, 2024) and its role in controlling pluripotent-like identity and tumor-initiating function in human colorectal cancer (Bergin et al, 2021) has also been documented.

Some studies have also documented roles for EHMT2 in immune regulation. For example, in the case of bladder cancer, the EHMT2/DNMT network has been shown to trigger immune-mediated bladder cancer regression (Segovia et al, 2019). This study used the broad-spectrum inhibitor CM-272, which targets EHMT1 and 2, and DNA methyltransferases and showed that in combination with cisplatin or with anti-PD-L1-based therapies, this drug

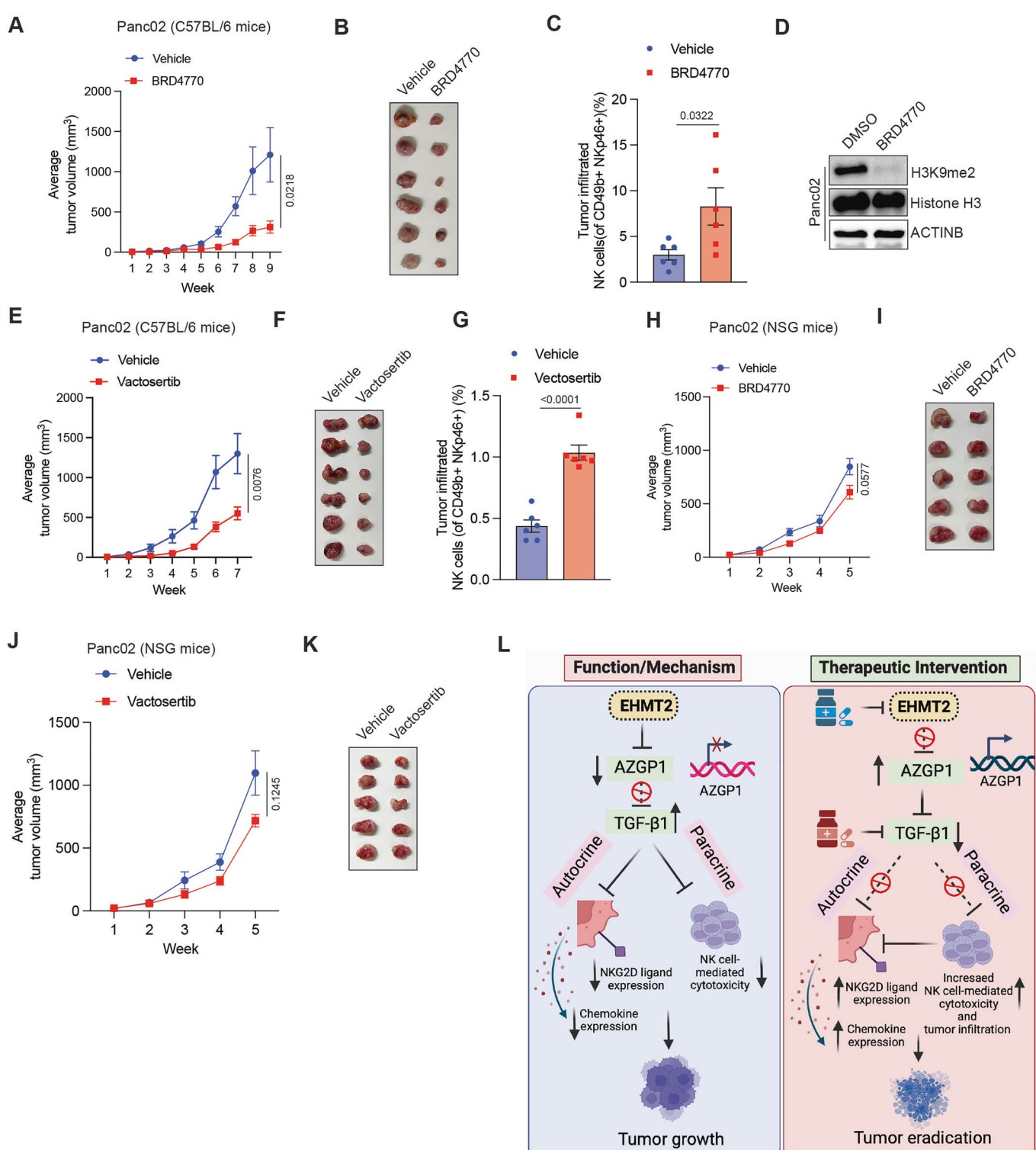

promoted apoptosis and immunogenic cell death and led to the conversion of immune non-responsive tumors to immune-responsive tumors (Segovia et al, 2019). Similarly, regulation of T-cell-mediated cytotoxicity by EHMT2 was also observed in the

case of cutaneous melanoma (Kato et al, 2020). Additional studies have also shown the role of EHMT2 in regulating macrophage function (Chen et al, 2009; Yoshida et al, 2015) and have also documented its role in regulating T cell differentiation during

**Figure 8. Pharmacological inhibition of EHMT2 and TGF-β1 induces tumor suppression in immunocompetent mice.**

(A) C57BL/6 mice were injected with Panc02 cells and treated with vehicle or BRD4770. Tumor volumes were measured at the indicated weeks. Average tumor volumes are plotted at the indicated time points. ($n = 6$). For the analysis of tumor progression in mice, the statistical assessment was performed using the area under the curve method, followed by unpaired two-tailed Student's $t$-tests. (B) Representative tumor images under the indicated conditions for the experiment shown in panel (A) under indicated conditions. (C) Measurement of tumor-infiltrated NK cells (%) in Panc02 tumors treated with either vehicle or BRD4770 using FACS analysis and are plotted. ($n = 6$). $P$ values were calculated using unpaired two-tailed Student's $t$-test. (D) Panc02 cells were treated with BRD4770 (5 μM) for 48 h. Indicated proteins were analyzed using immunoblotting. ACTINB was used as a loading control. (E) C57BL/6 mice were injected with Panc02 cells and treated with vehicle or vactosertib. Tumor volumes were measured at the indicated weeks. Average tumor volumes are plotted at the indicated weeks. ($n = 6$). For the analysis of tumor progression in mice, the statistical assessment was performed using the area under the curve method, followed by unpaired two-tailed Student's $t$-tests. (F) Representative tumor images for the experiment shown in panel (E) under the indicated conditions. (G) Measurement of tumor-infiltrated NK cells (%) in Panc02 tumors treated with either vehicle or vactosertib and are plotted. ($n = 6$). $P$ values were calculated using unpaired two-tailed Student's $t$-test. (H) NSG mice were injected with Panc02 cells and treated with vehicle or BRD4770. Tumor volumes were measured at the indicated weeks. Average tumor volumes are plotted at the indicated time points. ($n = 5$). For the analysis of tumor progression in mice, the statistical assessment was performed using the area under the curve method, followed by unpaired two-tailed Student's $t$-tests. (I) Representative tumor images for the experiment shown in panel (H) under the indicated conditions. (J) NSG mice were injected with Panc02 cells and treated with vehicle or vactosertib. Tumor volumes were measured at the indicated weeks. Average tumor volumes are plotted at the indicated time points. ($n = 5$). For the analysis of tumor progression in mice, the statistical assessment was performed using the area under the curve method following by unpaired two-tailed Student's $t$-tests. (K) Representative tumor images for the experiment shown in panel (J) under the indicated conditions. (L) Model showing the function and mechanism of EHMT2-driven suppression of NK cell-mediated anti-tumor immunity and nodes that can be therapeutically targeted for enhancing NK cell-mediated tumor clearance. All quantitative data are shown as the mean ± SEM. Source data are available online for this figure.

murine intestinal inflammation (Antignano et al, 2014). However, the role of EHMT2 in the regulation of NK cells-mediated anti-tumor immunity has remained unexplored. Here, we show that EHMT2 inhibition in uveal melanoma, breast cancer and pancreatic cancer cells promoted their NK cell-mediated tumor suppression. These results identify EHMT2 as a key suppressor of NK cell function.

The TGF-β pathway is known for its role in immune suppression, including its ability to suppress NK cell function (Batlle and Massague, 2019; Massague and Sheppard, 2023). We found that the loss of EHMT2 led to increased expression of AZGP1, culminating in the attenuation of TGF-β1, thereby facilitating an enhanced NK cell-mediated eradication of cancer cells. Interestingly, we found that the impact of the EHMT2-AZGP1-TGF-β1 axis was both autocrine and paracrine. Specifically, the inhibition of EHMT2 and TGF-β1 resulted in increased expression of NKG2D ligands (ULBP3 and MICB) on cancer cells in an autocrine manner, resulting in enhanced NK cell-mediated cytotoxicity. While TGF-β1 caused paracrine regulation of NK cell-mediated cytotoxicity by suppressing NK cell receptors NKG2D and NKp30, NK cell-specific transcriptional factors such as Eomes, T-Bet and cytokines such as IFN-γ and upregulating the ILC1 markers such as CD49a, CD56, and CXCR6. This is consistent with previous reports that TGF-β1 can induce epigenetic regulation in NK cells (Castriconi et al, 2003; Cortez et al, 2017; Regis et al, 2020; Viel et al, 2016). TGF-β has been shown to induce epigenetic remodeling in natural killer (NK) cells, notably by repressing the expression of activating receptors such as NKG2D through histone modifications and DNA methylation (Viel et al, 2016). This immunosuppressive cytokine also diminishes NK cell-mediated cytotoxicity and cytokine secretion via SMAD-dependent transcriptional repression pathways (Viel et al, 2016). Prolonged exposure to TGF-β can lead to the establishment of a stably dysfunctional NK cell phenotype, suggesting the involvement of heritable epigenetic silencing mechanisms (Cortez et al, 2017). Furthermore, previous studies have implicated TGF-β signaling in promoting active cytotoxic NK cells to transition to non-cytotoxic ILC1 cells (Cortez et al, 2017; Gao et al, 2017; Lopes et al, 2022). We found that treatment of NK cells with TGF-β1 showed increased expression of the markers of ILC1. Furthermore, Ehmt2

loss resulted in reduced ILC1 cell population and showed an increased number of infiltrated active cytotoxic NK cells in syngeneic tumors.

We also found increased expression of NK cell migration-promoting chemokines, such as CCL27 and CXCL10, following the inhibition of EHMT2 and TGF-β1. More importantly, using a variety of mouse models, including immunocompetent mice, *Rag2* KO mice (lacking T- and B cells) and immunodeficient NSG mice (lacking NK-, T-, and B cells), we demonstrated that EHMT2 loss promoted NK cell-mediated tumor suppression, regardless of the presence or absence of T- and B cells. Consistent with this, NK cell depletion reversed the Ehmt2 loss induced tumor suppression in both breast and pancreatic cancer syngeneic tumors in mice. These results highlight that NK cells are central regulators of EHMT2 loss-induced tumor suppression. Furthermore, in all the cancer cell line models that we tested in the immunodeficient mice, Ehmt2 loss alone was either less effective or not sufficient to induce tumor suppression, indicating that anti-tumor immunity, particularly NK cell-mediated cytotoxicity triggered by Ehmt2 loss, was key to in vivo tumor suppression.

We also demonstrated that small-molecule inhibitors of EHMT2 and TGF-β1 can be used as a strategy for unleashing the immune response against cancer cells. The implication of EHMT2 in suppressing NK cell-mediated cytotoxicity across various cancer types broadens the potential therapeutic applications of EHMT2 and TGF-β1 inhibitors. Our data demonstrate that targeting EHMT2 could serve as a promising strategy to enhance the efficacy of NK cell-based immunotherapies. Furthermore, the association between increased AZGP1 expression and improved overall patient survival in UM patients highlights the potential of AZGP1 as a prognostic marker.

In conclusion, our study identifies EHMT2 as a cell-extrinsic driver of tumor growth that largely works via suppressing NK cell-mediated anti-tumor immunity, primarily through the repression of AZGP1 and activation of the TGF-β1. The findings offer previously undocumented insights into the epigenetic regulation of NK cell-mediated cytotoxicity and pave the way for the translation of EHMT2 and TGF-β1-targeted therapies alone or in combination with other cancer immunotherapeutic agents for cancer treatment in the clinic.

# Methods

### Reagents and tools table

| Reagent/resource | Reference or source | Identifier or catalog number |
| --- | --- | --- |
| **Experimental models** | | |
| Mouse: NSG | Jackson Laboratory | Stock No. 005557 |
| Mouse: C57BL/6 | Jackson Laboratory | Stock No. 000664 |
| Mouse: BALB/c | Jackson Laboratory | Stock No. 000651 |
| Mouse: *Rag2* KO | Jackson Laboratory | Stock No. 008449 |
| **Cell Lines** | | |
| MP46 | ATCC | CRL-3298 |
| MP41 | ATCC | CRL-3297 |
| Mel92.1 | Sigma-Aldrich | 13012458 |
| HEK-293T | ATCC | CRL-3216 |
| NK92MI | ATCC | CRL-2408 |
| EMT6 | ATCC | CRL-2755 |
| PANC10.05 | ATCC | CRL-2547 |
| HCC70 | ATCC | CRL-2315 |
| SU.86.86 | ATCC | CRL-1837 |
| MDA-MB-231 | ATCC | HTB-26 |
| Panc02 | Cytion | 300501 |
| **Recombinant DNA** | | |
| pLX304-V5 | Horizon Discovery | 25890 RRID: Addgene_25890 |
| pLX304-AZGP1-V5 | Horizon Discovery | OHS6085-213578414 |
| pLX307-TGFB1-V5 | Addgene | 98377 RRID: Addgene_98377 |
| pLX307-V5 | Addgene | 41392 RRID: Addgene_41392 |
| Plasmid: piggyBac GFP-Luc | Ding et al, 2005 (PMID:16096065) | N/A |
| Plasmid: Act-PBase | Ding et al, 2005 (PMID:16096065) | N/A |
| Plasmid: psPAX2 | Addgene | 12260, RRID: Addgene_12260 |
| Plasmid: pMD2.G | Addgene | 12259, RRID: Addgene_12259 |
| Plasmid: lentiCRISPR v2 | Addgene | 52961 RRID: Addgene_52961 |
| **shRNAs** | | |
| *EHMT2* shRNA#1 | TRCN0000115667 | RHS3979-98058501 |
| *EHMT2* shRNA#2 | TRCN0000115668 | RHS3979-98058509 |
| *EHMT1* shRNA#1 | V3LHS_319788 | RHS4430-101513560 |
| *EHMT1* shRNA#2 | V3LHS_319785 | RHS4430-101519966 |
| *AZGP1* shRNA#1 | TRCN0000061360 | RHS3979-9628544 |
| *AZGP1* shRNA#2 | TRCN0000061362 | RHS3979-9628546 |

| Reagent/resource | Reference or source | Identifier or catalog number |
| --- | --- | --- |
| *MICB* shRNA#1 | TRCN0000061334 | RHS3979-9628518 |
| *MICB* shRNA#2 | TRCN0000061336 | RHS3979-9628520 |
| *ULBP3* shRNA#1 | V3LHS_340503 | RHS4430-100989651 |
| *ULBP3* shRNA#2 | V3LHS_340501 | RHS4430-100992303 |
| *TGFB1* shRNA#1 | TRCN0000003316 | RHS3979-9571962 |
| *TGFB1* shRNA#2 | TRCN0000003319 | RHS3979-9571965 |
| *Ehmt2* shRNA#1 | TRCN0000054546 | RMM3981-9621745 |
| **Antibodies** | | |
| EHMT2 (Diluted at 1:1000 for Western Blot (WB), 1:100 for Immunoprecipitation, 1:50 for CUT& RUN) | Cell Signaling Technology | 68851 RRID: AB_2799755 |
| EHMT1 (Diluted at 1:1000 for WB) | Cell Signaling Technology | 35005 RRID: AB_2799068 |
| H3K9me2 (Diluted at 1:1000 for WB, 1:50 for CUT&RUN) | Cell Signaling Technology | 4658 RRID: AB_10544405 |
| Histone H3 (Diluted at 1:2000 for WB) | Cell Signaling Technology | 9715 RRID: AB_331563 |
| MICB (Diluted at 1:1000 for WB) | Cell Signaling Technology | 77296 RRID: NA |
| ULBP3 (Diluted at 1:1000 for WB) | Proteintech | 30058-1-AP RRID: AB_3086222 |
| AZGP1 (Diluted at 1:500 for WB) | Proteintech | 66178-1-Ig RRID: AB_2881573 |
| TGF-β (Diluted at 1:1000 for WB) | Cell Signaling Technology | 3711 RRID: RRID: AB_2063354 |
| β-Actin (D6A8) (Diluted at 1:2000 for WB) | Cell Signaling Technology | 8457 RRID: AB_10950489 |
| V5-Tag (Diluted at 1:1000 for WB) | Cell Signaling Technology | 13202 RRID: AB_2687461 |
| Anti-Mouse Asialo-GM1 (Injected 100 μg/mouse, once a week) | Biolegend | 146002 RRID: AB_2562206 |
| Anti-Mouse NK1.1 (Injected 200 μg/mouse twice a week) | BioXcell | BE0036 RRID: AB_1107737 |
| In Vivo MAb mouse IgG2a isotype control (Injected 200 μg/mouse twice a week) | BioXcell | BE0085 RRID: NA |
| PE-anti-mouse NKp46 (5 μl/sample for FACS) | Biolegend | 137604 RRID: AB_2235755 |
| APC-anti-mouse CD49b (5 μl/sample for FACS) | Biolegend | 103516 RRID: AB_2566101 |
| PE-anti-mouse CD49a (5 μl/sample for FACS) | Biolegend | 142604 RRID: AB_10945158 |
| APC-anti-mouse CD8a (5 μl/sample for FACS) | Biolegend | 100712 RRID: AB_312751 |
| PE-anti-mouse CD4 (5 μl/sample for FACS) | Biolegend | 100408 RRID: AB_312693 |
| Brilliant Violet 650 Anti-mouse CD45 (5 μl/sample for FACS) | Biolegend | 109836 RRID: AB_2563065 |
| Anti-mouse CD16/32 (Diluted 1:100 for FACS) | Biolegend | 101302 RRID: AB_312801 |

| Reagent/resource | Reference or source | Identifier or catalog number |
|---|---|---|
| APC-anti-MICB (5 µl/sample for FACS) | R&D systems | FAB1599A RRID: AB_2297703 |
| APC-anti-ULBP3 (5 µl/sample for FACS) | R&D systems | FAB1517A RRID: AB_3083730 |
| APC-mouse IgG2B isotype (5 µl/sample for FACS) | R&D systems | IC0041A RRID: AB_357246 |
| TGF-beta-1,2,3 Mab(1D11) (2 µg/ml for Neutralization) | Thermo Fisher Scientific | MA5-23795 RRID: AB_2609812 |
| T-bet/TBX21 (Diluted at 1:1000 for WB) | Cell Signaling Technology | 13232 RRID: AB_2616022 |
| NCR3/NKp30 (Diluted at 1:1000 for WB) | Cell Signaling Technology | 87214 RRID: NA |
| EOMES (Diluted at 1:1000 for WB) | Cell Signaling Technology | 81493 RRID: AB_2799974 |
| CD49a (Diluted at 1:1000 for WB) | Cell Signaling Technology | 71747 RRID: NA |
| CD103 (Diluted at 1:1000 for WB) | Novus Biologicals | NBP3-03545 RRID: AB_3532135 |
| NKG2D (Diluted 1 µg/mL for WB) | R&D systems | MAB1391-SP RRID: AB_2133263 |
| CD56 (Diluted at 1:1000 for WB) | Cell Signaling Technology | 99746 RRID: AB_2868490 |
| CXCR6 (Diluted 0.5 µg/mL for WB) | Thermo Fisher Scientific | PA5-19936 RRID: AB_11154796 |
| CD69 (Diluted at 1:1000 for WB) | Cell Signaling Technology | 39275 RRID: NA |
| Ki-67 (Diluted at 1:300 for IHC) | Abcam | ab15580 RRID: AB_443209 |
| **Oligonucleotides and other sequence-based reagents** | | |
| Primers | This study | Table EV2 |
| CUT& RUN Primers | This study | Table EV2 |
| sgRNAs | This study | Table EV2 |
| **Chemicals, enzymes and other reagents** | | |
| A366 | Cayman chemicals | 16081 |
| UNC0642 | Selleckchem | S7230 |
| BRD4770 | Selleckchem | S7591 |
| Vactosertib (TEW-7197) | Selleckchem | S7530 |
| Calcein-AM | Sigma-Aldrich | 206700 |
| DMEM | Sigma-Aldrich | D5796 |
| RPMI-1640 | Sigma-Aldrich | R8758 |
| MEM | Sigma-Aldrich | M4526 |
| Opti-MEM | GIBCO | 31985070 |
| Fetal Bovine Serum | GIBCO | 10437-028 |
| Horse serum | Sigma-Aldrich | H1138 |
| Trypsin-EDTA | GIBCO | 25200-056 |
| Penicillin-Streptomycin | GIBCO | 15140-122 |
| Effectene Transfection Reagent | QIAGEN | 301427 |
| CountBright™ Absolute Counting Beads, for flow cytometry | Thermo Fisher Scientific | C36950 |
| Transwell 24-well plates | Costar | 3421 |

| Reagent/resource | Reference or source | Identifier or catalog number |
|---|---|---|
| **Software** | | |
| Prism 10.0 | GraphPad | www.graphpad.com/scientific software/prism |
| ImageJ | ImageJ | https://imagej.net/ij/ |
| **Other** | | |
| Invitrogen™ CyQUANT™ LDH Cytotoxicity Assay | Invitrogen | C20301 |
| CUT&RUN Assay Kit | Cell Signaling Technology | 86652 |
| Structural Genome Consortium's epigenetic chemical probe inhibitor library | Cayman chemicals | 17525 |
| Chemokine array | RayBiotech | AAH-CHE-1 |
| Recombinant Human TGF-beta-1 (Human Cell-expressed) Protein | R&D Systems | 7754-BH-005/CF |
| Human IFN-gamma Quantikine QuicKit ELISA | R&D Systems | QK285 |
| Tumor Dissociation Kit | Miltenyi Biotec | 130-096-730 |
| Spleen Dissociation Kit | Miltenyi Biotec | 130-095-926 |
| Alanine Transaminase (ALT) Assay Kit | Cayman chemicals | 700260 |
| Aspartate Aminotransferase (AST) Assay Kit | Cayman chemicals | 701640 |
| Creatinine (serum) Assay Kit | Cayman chemicals | 700460 |

## Cell lines and cell culture

The human uveal melanoma cell lines (MP46 and MP41 cells), breast cancer cell lines (MDA-MB-231, HCC70, and EMT6 cells), and pancreatic cancer cell lines (PANC10.05, SU.86.86, and Panc02 cells), NK92MI and 293T cells were purchased from the American Type Culture Collection (ATCC). Mel92.1 cells were purchased from Sigma-Aldrich. OMM2.5 cells were kindly provided by Hans Grossniklaus' laboratory (Emory University). MP46 and MP41 cells were cultured in RPMI-1640 media supplemented with 25% FBS and 1% penicillin-streptomycin. The HCC70, PANC10.05, SU.86.86, OMM2.5, and Mel92.1 cells were cultured in RPMI-1640 media supplemented with 10% FBS and 1% penicillin-streptomycin. MDA-MB-231, SU.86.86, Panc02, EMT6, and 293T cells were grown in DMEM supplemented with 10% FBS and 1% penicillin-streptomycin. The NK92MI cells were grown in Alpha Minimum Essential Medium without ribonucleosides and deoxyribonucleosides with 2 mM L-glutamine, 1.5 g/L sodium bicarbonate, 0.2 mM myo-inositol, 0.1 mM 2-mercaptoethanol, 0.02 mM folic acid, 12.5% horse serum, 12.5% FBS, and 1% penicillin-streptomycin. All provided cell lines were confirmed via STR profiling status by respective vendors or confirmed as needed. Mycoplasma-negative status of all cell lines was ensured using the universal mycoplasma detection kit from ATCC (Cat# 30-1012 K). Details of the cell lines are presented in the Reagents and tools table.

## Chemical genetics screen using small-molecule inhibitors targeting specific epigenetic regulators

The chemical genetics screen was performed using the Structural Genome Consortium's (SGC) epigenetic chemical probe inhibitor library containing 36 small-molecule inhibitors targeting epigenetic regulators (Cayman Chemical; Cat. No. 17525). All inhibitors were dissolved in DMSO to prepare 10 mM stocks. The inhibitors and their targets are listed in Table EV1. MP46 cells were seeded in six-well plates ($3 \times 10^5$/well) and treated were with the indicated concentrations of small-molecule inhibitors as listed in Table EV1 or DMSO as a negative control. After 48 h of treatment with the inhibitors, MP46 cells were co-cultured with NK92MI cells with 1:10 ratio [Target (Cancer cells): Effector (NK cells)] for 3 h followed by measurement of NK cell-mediated cytotoxicity using LDH cytotoxicity assay as described below in this Methods section.

## LDH cytotoxicity assay

The LDH cytotoxicity assay was performed using the CyQUANT LDH Cytotoxicity Assay Kit (Thermo Fisher Scientific, USA; Cat. No. C20301), as previously described (Chava et al, 2020). NK92MI cells ($0.5 \times 10^5$, $1 \times 10^5$, $2 \times 10^5$, and $4 \times 10^5$ cells; volume 100 µL) served as effector cells and were incubated with nonspecific sgRNA-expressing, gene-specific sgRNA-expressing, nonspecific shRNA-expressing, gene specific shRNA-expressing or DMSO– or inhibitor-treated "target" cancer cells ($1 \times 10^4$ cells; volume 100 µL) in 96-well ultra-low attachment tissue culture plates in different cancer cell:NK cell ratios (1:5, 1:10, 1:20, and 1:40). The plates were incubated at 37 °C in a $CO_2$ incubator for 3 h followed by centrifuging the 96-well plates at 1500 rpm for 5 min. The supernatants were then collected from each well into a fresh 96-well plate, and 50 µL of LDH substrate mixture was added to each well. The plate was incubated for 10–20 min at room temperature in the dark, and absorbance at 490 and 680 nm was measured using the Biotek Synergy MX Multi Format Microplate Reader (Biotek, USA). The absorbance at 680 nm was subtracted from the absorbance at 490 nm to calculate the percent (%) cytotoxicity using the formula below.

We used the following controls for all the LDH cytotoxicity assays: (1) NK cells alone. (2) Cancer cells alone. (3) NK cell medium alone. (4) Cancer cells medium alone. (5) NK cell medium + cancer cell medium. To calculate the percent (%) cytotoxicity using the formula below and converted to fold changes and plotted in the graphs.

$$\frac{\text{LDH experimental} - \text{LDH effector cells} - \text{LDH spontaneous} \times 100}{\text{LDH maximal}}$$

## Calcein-AM cytotoxicity assay

Mel92.1 and MP41 cells expressing either nonspecific shRNA or *EHMT2* shRNA or treated with vehicle or EHMT2 inhibitors were labeled with 1 µM of Calcein-AM (Sigma-Aldrich, Cat. No. C1359) in 1x PBS and incubated for 30 min at room temperature. Calcein-AM-labeled cells were washed twice with 1× PBS and resuspended in complete DMEM medium. NK92MI cells ($1 \times 10^5$ cells; volume 100 µL) served as effector cells and were incubated with Calcein-AM-labeled target cancer cells ($1 \times 10^4$ cells; volume 100 µL) in a

1:10 cancer cell:NK cell ratio in 96-well ultra-low attachment tissue culture plates, and as a maximum release control 2% Triton X-100 in culture medium were used. The plates were incubated at 37 °C in $CO_2$ for 4 h. After incubation, 96-well tissue culture plates were centrifuged at 1000 rpm for 3 min. About 100 µL supernatants were then collected from each well and transferred into new plates. Samples' absorbance was measured using the Biotek Synergy MX Multi Format Microplate Reader (Biotek, USA) using GFP filter (Ex/Em:485 nm/520 nm). To calculate the percent (%) cytotoxicity using the formula below and converted to fold changes and plotted in the graphs.

$$\frac{\text{Test release} - \text{Spontaneous release}. \times 100}{\text{Maximal release} - \text{Spontaneous release}}$$

## Plasmids and preparation of the lentiviral and retroviral stable cell lines

Gene-specific lentiviral shRNAs were obtained from Open Biosystems, and ORF constructs were obtained from Horizon Discovery. The catalog numbers for the shRNAs and ORF constructs are provided in the Reagents and Tools Table. Gene-specific lentiviral sgRNAs were cloned into the pLenti-CRISPR-V2 vector (Clone ID. 52961) obtained from Addgene; the sgRNA sequences are provided in the Reagents and Tools Table. For lentivirus production, plasmids were transfected into 293 T cells along with the PDM2.G and psPAX2 packaging plasmids were transfected using Effectene Transfection Reagent (Qiagen, USA) per the manufacturer's instructions. After 48 h, the lentivirus-containing supernatants were harvested, filtered, and used for infections. Lentiviral shRNA-infected Mel92.1 and MP41 cells were selected using 0.5 µg/mL puromycin. Lentiviral shRNA-infected EMT6 and Panc02 cells are selected using 2 µg/mL puromycin. For the pLX304-Blast-V5-based lentivirus, 4 µg/mL blasticidin (Thermo Fisher Scientific, USA) was used to select successfully transduced Mel92.1 and MP41 cells.

## RNA preparation, complementary DNA (cDNA) preparation, reverse transcription (RT), and quantitative PCR (qPCR) analysis

Total RNA was extracted with TRIzol® (Invitrogen, USA) and purified with RNeasy mini columns (Qiagen, USA) for the mRNA expression analyses. The cDNA was generated using the M-MuLV first-strand cDNA synthesis kit (New England Biolabs, USA) according to the manufacturer's instructions. Quantitative reverse transcription polymerase chain reaction (RT-qPCR) was performed using the Power SYBR® Green kit (Applied Biosystems, USA) according to the manufacturer's instructions. Beta actin (*ACTINB*) was used as a normalization control. Primer sequences used in the study are provided in Table EV2.

## Immunoblot analysis

Immunoblot analysis was performed as described previously (Santra et al, 2009). Briefly, protein extracts were prepared in RIPA lysis buffer (Thermo Fisher Scientific, USA; Cat. No. 89901) supplemented with protease inhibitors (Roche, USA)) and phosphatase inhibitors (Sigma-Aldrich, USA). Protein

concentrations were estimated using Bradford Protein Assay Reagent (Bio-Rad Laboratories, USA) according to the manufacturer's instructions. Protein extracts were separated using 8, 10, 12, or 15% sodium dodecyl sulfate polyacrylamide gel electrophoresis (SDS-PAGE) gels and were transferred onto a polyvinylidene difluoride (PVDF) membrane using a wet-transfer apparatus (Bio-Rad). Membranes were blocked in 5% non-fat dry milk prepared in Tris-buffered saline containing 0.1% Tween-20 and were probed with primary antibodies. After washing, the membranes were incubated with the appropriate horseradish peroxidase-conjugated secondary antibodies (1:5000) (GE Healthcare Life Sciences, USA). The blots were developed using SuperSignal West Pico or Femto Chemiluminescent Substrate (Thermo Fisher Scientific, USA). All antibodies used for immunoblotting are listed in the Reagents and Tools Table.

## Immunohistochemistry (IHC) for Ki-67 staining

Panc02 and EMT6 tumor tissues collected from experimental mice were fixed overnight in 10% formalin, embedded in paraffin, and cut into 5 μm sections. Briefly, following deparaffinization of formalin-fixed, paraffin-embedded slides containing tumor tissue sections, antigen retrieval was performed in citrate buffer (pH 6.0) at 97 °C for 20 min using the Lab Vision PT Module (Thermo Scientific, USA). Endogenous peroxides were blocked using hydrogen peroxide for 30 min. The slides were then washed with 1× Tris-buffered saline (TBS), and proteins were blocked using 0.3% bovine serum albumin (BSA) for 30 min. Slides were incubated in Ki-67 antibody (dilution 1:300), followed by incubation with secondary anti-rabbit HRP-conjugated antibody (Dako, Germany). Slides were then stained using the Dako Liquid DAB+ Substrate Chromogen System and counterstained with Automation Hematoxylin Histological Staining Reagent (Thermo Fisher Scientific, USA). 20× images were captured using light microscope, and Ki-67 quantification was performed using ImageJ software (NIH; https://imagej.nih.gov/ij/). The Ki-67 antibody used for immunohistochemistry analysis is listed in the Reagents and tools table.

## RNA-sequencing (RNA-seq) and data analysis

Mel92.1 and MP41 cells treated with EHMT2 inhibitors A366 (1 μM) and UNC0642 (1 μM) for 48 h were used to prepare total RNA and which was then used for gene expression analysis on an Illumina HiSeq 2500 system. Total RNA was extracted using Trizol reagent (Invitrogen) according to the manufacturer's instructions and then purified on RNAeasy Mini Columns (Qiagen) according to the manufacturer's instructions. mRNA was purified from ~500 ng total RNA using oligo-dT beads and then sheared by incubation at 94 °C. Following first-strand synthesis with random primers, second-strand synthesis was performed with dUTP to generate strand-specific sequencing libraries. The cDNA library was then end-repaired and A-tailed. Adapters were then ligated, and second-strand digestion was performed using uracil-DNA-glycosylase. Indexed libraries that met appropriate cut-offs for both were quantified by RT-qPCR using a commercially available kit (KAPA Biosystems). The insert size distribution was determined using LabChip GX or an Agilent Bioanalyzer. Samples with a yield ≥0.5 ng/μl were used for sequencing on the Illumina HiSeq 2500 system. Images generated by the sequencers were converted

into nucleotide sequences by the base-calling pipeline RTA 1.18.64.0 and stored in FASTQ format. The raw sequencing data in the FASTQ files were subjected to a quality check (FastQC), removal of adapter content, and quality thresholding (removal of reads with Phred score <30). Reads that passed the quality thresholds were mapped to the latest stable version of the Mouse reference genome mm10 (GRCm38/mm10, Ensembl) using Bowtie2 and Tophat 2.1.1. The expression of the assembled transcriptomes was estimated using Cufflinks 2.2.1 (Trapnell et al, 2012). Briefly, the quality of the assemblies was assessed, and the normalized gene and transcript expression profiles were computed for each sample. The normalization was performed using the classic fragments per kilobases per million fragments (FPKM) method followed by Log2 transformation. The gene-level differential expression between conditions was estimated using the Log2-transformed FPKM values of transcripts sharing each gene ID. The uncorrected $p$ value of the test statistic and the false discovery rate (FDR)-adjusted $p$ value of the test statistic ($q$ value) were estimated for differentially expressed genes (DEGs). Any gene with a $p$ value greater than the FDR after Benjamini-Hochberg correction for multiple testing was deemed to be differentially expressed between the test condition and control condition. The RNA-seq data have been submitted to GEO (Accession No. GSE196624).

## Survival analysis using uveal melanoma TCGA data

TCGA data for uveal melanoma has been previously published (Robertson et al, 2017). The R package curatedTCGAData was used to download the gene expression and associated clinical data from the TCGA database. The multiAssayExperiment object was retrieved using the curatedTCGAData function in the package. We retrieved the expression data of AZGP1 mRNA and survival-related clinical features from this multiAssayExperiment object to perform the survival analysis. The survival analysis was performed using the R package survminer. The coxph function in the package was used to calculate the hazard ratios and the $p$ value. The ggsurvplot function was used for the Kaplan–Meier survival plot.

## Cleavage under targets and release using nuclease (CUT&RUN) assay

CUT&RUN assays were performed in Mel92.1 cells using the CUT&RUN Assay Kit (Cell Signaling Technology, USA; Cat No. 86652) according to the manufacturer's instructions. Briefly, $5 \times 10^5$ cells were harvested, washed, bound to activated Concanavalin A-coated magnetic beads, and permeabilized. The bead–cell complexes were incubated overnight with the appropriate antibody at 4 °C. Then, the complexes were washed three times, and the cells were resuspended in 100 μl protein A and G/micrococcal nuclease (pAG/MNase) and incubated for 1 h at room temperature. The samples were then washed three times with digitonin buffer with protease inhibitors, resuspended in 150 μL digitonin buffer, and incubated for 5 min on ice. pAG/MNase was activated by adding calcium chloride, and the samples were incubated at 4 °C for 30 min. The reaction was stopped by adding 150 μL stop buffer, and the samples were incubated at 37 °C for 10 min to release the DNA fragments. The DNA was extracted using the DNA purification columns included in the CUT&RUN Assay Kit. Quantitative PCR was then performed using AZGP1 promoter-specific primers, and

relative fold change was calculated as the ratio of EHMT2– or H3K9me2-immunoprecipitated DNA to IgG-precipitated DNA. The primer sequences and antibodies used for the CUT&RUN assays are listed in Table EV2.

## Mouse studies

Details of individual animal studies are presented below. All mouse strains were obtained from Jackson Laboratory (see Reagents and Tools Table). All mouse studies were performed in accordance with protocols and guidelines approved by the Institutional Animal Care and Use Committee (IACUC) of the University of Alabama at Birmingham (UAB) (IACUC protocol number: IACUC-21856). Animals were housed in a temperature-controlled environment (22 °C, 30–70% humidity) with a 12-h light/dark cycle. Food and water were available ad libitum.

## Orthotopic mammary fat pad-based breast tumorigenesis experiments

EMT6 cells stably expressing firefly luciferase under the control of a cytomegalovirus promoter were generated by co-transfection of the transposon vector piggyBac GFP-Luc and the helper plasmid Act-PBase as described previously (Ding et al, 2005). Cells with stable transposon integration were selected using blasticidin S (Thermo Fisher Scientific). EMT6 cells expressing nonspecific (NS) shRNA or *Ehmt2* shRNA-expressing ($0.5 \times 10^6$ cells in 5–6-week-old female BALB/c (stock no. 000651, Jackson Laboratory) and $2.5 \times 10^4$ cells in 5–6-week-old female NSG mice (stock no. 005557, Jackson Laboratory) murine breast cancer cells were orthotopically injected into the mouse mammary fat pads. The mice were imaged weekly by In Vivo Imaging System (PerkinElmer, Waltham, MA, USA). Total luminescence counts of the tumor-bearing areas were measured using the Living Image in vivo imaging software (PerkinElmer). At the end of the experiment, the mice were sacrificed. For NK cell depletion, of Ultra-LEAF™ Purified anti-Asialo-GM1 clone Poly21460 (BioLegend, USA) antibodies were injected intraperitoneally (100 µg/mouse) into BALB/c mice once a week for 3 weeks. As a control, 100 µg of Ultra-LEAF purified mouse IgG1, κ-isotype control antibody (clone MG1-45, BioLegend, USA) were administrated intraperitoneally (100 µg/mouse) once a week for 3 weeks. At the end of the experiment, the mice were sacrificed, the spleens were collected, and the NK cell quantitation was performed by FACS analysis. All the antibodies used for this experiment are listed in the Reagents and tools table.

## Mouse subcutaneous tumorigenesis experiments

Panc02 cells ($5 \times 10^6$) expressing NS shRNA control and *Ehmt2* shRNA were injected subcutaneously into 5–6-week-old male NSG mice (stock no. 005557, Jackson Laboratory) or 5–6-week-old male C57BL/6 mice (stock no. 000664, Jackson Laboratory) or 5–6-week-old male B6.Cg-*Rag2*^*tm1.1Cgn*/J (Rag2 KO mice; stock no. 008449, Jackson Laboratory). Tumor volume was measured every week and plotted, and tumor size was calculated using the following formula: length × width$^2$ × 0.5. Subcutaneous tumors from individual groups were harvested and imaged. For NK cell depletion, monoclonal purified in vivo anti-NK1.1 clone PK136 (Bioxcell, USA) antibodies

were injected intraperitoneally into 5–6-week-old male C57BL/6 mice twice every week for 6 weeks. As a control, 200 µg of Ultra-LEAF purified mouse IgG1, κ-isotype control antibody (BioLegend; clone MG1-45) were administrated intraperitoneally (200 µg/ mouse) twice every week for 6 weeks. At the end of the experiment, the spleens were collected and analyzed the NK cell quantitation by FACS analysis. All the antibodies used for this experiment are listed in the Reagents and tools table.

## Mouse subcutaneous tumorigenesis experiments with EHMT2 inhibitor and TGF-β1 inhibitor

Panc02 cells ($5 \times 10^6$) were injected subcutaneously into 5–6-week-old male C57BL/6 mice (stock no. 000664, Jackson Laboratory) or 5–6-week-old male NSG mice (stock no. 005557, Jackson Laboratory). Mice were treated with EHMT2 inhibitor (BRD4770) 15 mg/kg body weight or vehicle (corn oil) intraperitoneally, or with TGFβ1 inhibitor (Vactosertib, also known as TEW-7197) 50 mg/kg body weight or vehicle (artificial gastric fluid formulation: 900 mL ddH$_2$O containing 7 mL of conc. HCl, 2.0 g NaCl, and 3.2 g pepsin) via oral gavage. Tumor volume was measured every week and plotted, and tumor size was calculated using the following formula: length × width$^2$ × 0.5. Subcutaneous tumors from individual groups were harvested and imaged.

## Measurement of NK cells, ILC1, and T cells in spleen and tumors using flow cytometry

Tumor or spleen dissociation was performed using the Tumor Dissociation Kit, Mouse (Miltenyi Biotec, USA; Cat. No. 130-096-730), according to the manufacturer's instructions. Briefly, tumors were excised, and necrotic areas were trimmed and excluded from the analysis. Tumors were cut into 3 mm × 3 mm pieces, and those from different areas of the tumor were added to MACS C tubes, containing RPMI-1640 medium with enzyme mix. Tissues were dissociated using a gentleMACS Octo dissociator with heaters. Cell suspensions were passed through a MACS SmartStrainer (70 µm) that was placed on a 15-mL tube to collect the single-cell suspensions. After washing with FACS buffer (PBS with 2% FBS), cells were used for flow cytometry analysis. In brief, washed cells were incubated with Fc block (anti-mouse CD16/32 antibody) at a dilution of 1:100 for 10 min. Cells were then stained with anti-mouse CD49b and anti-mouse CD335 (NKp46) for NK cells; anti-mouse CD45 Antibody, anti-mouse CD8a antibody and anti-mouse CD4 antibody used for measurement of T-cells; anti-mouse CD45, anti-mouse CD49a and CD49b for ILC1 cells and active NK cells for 30 min, followed by washing and then subjected to flow cytometry analysis. CD49b and CD335 (NKp46) double-positive NK cells were measured in mouse tumors and Spleen. CD45 and CD49b double-positive infiltrated NK cells were measured as active cytotoxic NK cells, and CD45 and CD49a double-positive infiltrated NK cells were measured as ILC1 cells. CD45 and CD8a double-positive infiltrated T cells were cytotoxic T cells, and CD45 and CD4 double-positive infiltrated T cells were Helper T cells. FACS analyses were performed using a BD LSRFortessa (BD Biosciences). All FACS data were analyzed using FlowJo software. All the antibodies used for these analyses are listed in the Reagents and tools table.

## Flow cytometry for measuring MICB and ULBP3 cell surface expression

Mel92.1 and MP41 cells were plated ($3 \times 10^5$ per well) and after 24 h cells were treated with EHMT2 inhibitors indicated concentrations, after 72 h incubation period, surface expression of ULBP3 and MICB proteins was analyzed by flow cytometry with APC-labeled ULBP3 and APC-labeled MICB antibodies. DMSO control and cells stained with nonspecific isotype-matched antibody control were used as negative controls. Data were analyzed using FlowJo software. Antibodies used for this experiment are listed in the Reagents and tools table.

## Conditioned medium preparation from cancer cells and measurement of NK cell-mediated cytotoxicity

Uveal melanoma cells were cultured at a density of $3 \times 10^5$ cells/well in six-well plates in RPMI-1640 medium supplemented with 10% FBS. Once the cells reached ~80% confluence, the cells were gently washed twice with 1× PBS, the complete medium was replaced with 3 ml/well of Opti-MEM along with DMSO or EHMT2 inhibitor, and the conditioned medium was harvested following 72 h of incubation. The conditioned medium was passed through 0.45 μm filters to remove cell debris. In selected experiments, conditioned medium was concentrated ~50-fold by centrifugation using Amicon Ultra-15 centrifugal filters with 3 kDa molecular weight cut-off (Millipore, USA). Total conditioned medium was used to treat NK92MI cell lines. For the TGF-β1 neutralization experiment, conditional media containing anti-TGF-β1 (1D11) antibody (2 μg/ml) or anti-mice IgG (2 μg/ml) treated to the NK cells. In parallel, Mel92.1 and MP41 cells were grown, and after 24 h of TGF-β1 neutralization, the LDH cytotoxicity assay was performed using the LDH cytotoxicity assay kit from Thermo Fisher Scientific (Cat. No. C20301), as described above.

## Human chemokine expression analysis using a human chemokine array

Human Chemokine Array C1 kits (Cat. No. AAH-CHE-1) were purchased from RayBiotech, and these were performed according to the manufacturer's instructions. Briefly, $2 \times 10^5$ Mel92.1 cells per well were plated into six-well plates in Opti-MEM Reduced-Serum Medium (Invitrogen). After 36 h, cells were treated with either DMSO or EHMT2 inhibitor UNC0642 (1 μM) for 72 h. Conditioned media were then collected from DMSO or UNC0642 (1 μM)-treated cells and concentrated using Centricon tubes (3-kDa cutoff). Array membranes were blocked in blocking buffer at room temperature (RT) for 1 h and were incubated overnight at 4 °C with 1.5 mL of concentrated conditioned media collected from DMSO- or UNC0642 (1 μM) treated cells. The next day, membranes were washed and incubated with 1 mL of primary biotin-conjugated antibody mixture provided by the kit at 4 °C overnight. Membranes were then washed and incubated with 2 mL of a 1:1000 dilution of horseradish peroxidase (HRP)–conjugated streptavidin at room temperature for 2 h. After incubation, array membrane signals were detected using the SuperSignal West Pico or Femto Chemiluminescent Substrate Kit (Thermo Fisher Scientific, USA). Spot signal intensities were quantified using ImageJ software (NIH; https://imagej.nih.gov/ij/). Positive controls on each array membrane were used to normalize spot intensities across different array membranes.

## NK cell migration assay

Mel92.1 and MP41 cells were treated with EHMT2 inhibitors DMSO or A366 (3 μM) or UNC0642 (3 μM) or TGF-β1 inhibitor vactosertib (1 μM) for 72 h in Opti-MEM Reduced-Serum Medium (Gibco, USA). Cell culture supernatants (conditioned media) were collected and concentrated using Amicon Ultra-15 centrifugal filters with 3 kDa molecular weight cut-off (Millipore-Sigma, USA). Concentrated conditioned media were collected, and 600 μl of it was added to the bottom chamber of a Corning Transwell system (6.5-mm diameter inserts and 5.0 μm pore size; Cat. No. 3421. Millipore-Sigma, USA). In parallel, NK cells (NK92MI) ($2.5 \times 10^5$) were seeded in the upper Transwell chamber in serum-free medium. Cells were incubated at 37 °C for 4 h, after which 500 μl of non-adherent cells from the bottom chamber were collected in fluorescence-activated cell sorting (FACS) tubes. A known number of fluorescent CountBright Absolute Counting Beads (Invitrogen, USA; Cat. No. C36950) in a volume of 50 μL were added to the migrated NK cells in each tube, cells were analyzed by flow cytometry. The absolute number of NK cells in 500 μL was calculated using the following formula: A*(B/C), where A = number of NK cell events, B = the number of beads added to each culture, and C = number of bead events. All neutralizing antibodies used for these experiments are listed in the Reagents and tools table.

## Clonogenic assays

Clonogenic assays were performed in Mel92.1 ($2.5 \times 10^3$ cells/well), MP41 ($5 \times 10^3$ cells/well), SU.86.86 ($2.5 \times 10^3$ cells/well), PANC10.05 ($2.5 \times 10^3$ cells/well), MDA-MB-231($2.5 \times 10^3$ cells/well), HCC70 ($2.5 \times 10^3$ cells/well) EMT6 ($1 \times 10^3$ cells), and Panc02 ($1 \times 10^3$ cells) or *EHMT2* knockdown or knockout cell lines were seeded in a 6-well plate for clonogenic assay. For EHMT2 inhibitor treatment conditions, after 48 h of plating, the cells were treated with DMSO or EHMT2 inhibitors and the medium containing inhibitors was changed every 3 days. The cells were allowed to grow for up to two weeks. Surviving colonies were fixed and stained with a solution containing a solution of 40% methanol, 10% acetic acid, and 0.005% Coomassie Brilliant Blue R-250 (Sigma-Aldrich), and the plates were scanned with an Epson Perfection V850 Pro Photo Scanner (USA).

## IFN-γ measurement using ELISA

IFN-γ protein levels were measured in NK92MI cells using Human IFN-gamma Quantikine ELISA Kit (Cat. No. QK285) from R&D Systems (Bio-Techne, USA) as per the manufacturer's instructions. Briefly, $5 \times 10^6$ NK92MI cells were plated in 100-mm cell culture dishes in Opti-MEM Reduced-Serum Medium in the presence or absence of TGF-β1 (10 ng/ml) for 24 h. Thereafter, conditioned medium was collected. Next, 50 μL of each of the samples, standards, and controls per well were added to the 96-well microplate strips provided. Then, 50 μL Antibody Cocktail was added to each well, and the plate was incubated for 1 h at room temperature on a horizontal orbital microplate shaker. Next, each

**The paper explained**

**The medical problem**

Cancer cells can evade the immune system, including natural killer (NK) cells, which are a vital part of the body's innate defense against tumors. While NK cell-based therapies are being investigated in clinical trials, the molecular mechanisms that allow cancer cells to suppress or escape NK cell surveillance remain poorly understood. Identifying these mechanisms is essential for designing new therapeutic strategies that enhance NK cell-mediated tumor destruction.

**Results**

This study identifies the histone methyltransferase EHMT2 as a key suppressor of NK cell-mediated anti-tumor immunity. Inhibiting EHMT2 in cancer cells, using either pharmacological inhibitors or genetic approaches, enhanced NK cell-mediated killing in several cancer types, including uveal melanoma, triple-negative breast cancer, and pancreatic cancer.

EHMT2 loss led to increased expression of AZGP1, a protein that inhibits the immunosuppressive cytokine TGF-β1. This in turn elevated the levels of activating ligands MICB and ULBP3 on tumor cells, which are recognized by NK cells. Additionally, EHMT2 loss increased the expression of chemokines such as CXCL10 and CCL27 in cancer cells, promoting NK cell migration.

In immunocompetent mouse models, inhibition of EHMT2 suppressed tumor growth in an NK cell-dependent manner. These effects were lost in immune-deficient mice lacking NK cells, confirming that the anti-tumor activity of EHMT2 inhibition is mediated predominantly through NK cells.

**Impact**

This study reveals that EHMT2 is an epigenetic regulator that enables tumors to evade NK cell-mediated immunity. By suppressing AZGP1 and increasing TGF-β1, EHMT2 contributes to an immunosuppressive tumor environment. Inhibiting EHMT2 reactivates NK cell responses by reversing these effects. The findings demonstrate that EHMT2 and TGF-β1 inhibitors could be developed as immunotherapeutic agents, especially for cancers that are resistant to conventional T cell-targeted therapies. These results broaden the scope of NK cell-based immunotherapies and identify EHMT2 as a promising therapeutic target.

well was aspirated and was washed, repeating the process twice for a total of three washes with Wash Buffer (400 μL each wash). Further, 100 μL of Substrate Solution was added to each well, and incubated for 30 min at room temperature on the benchtop, protected from light. Then, 50 μL of Stop Solution was added to each well. Optical density was determined at 450 and 540 nm using the Biotek Synergy MX Multi Format Microplate Reader (Biotek, USA), and the readings at 540 nm were subtracted from readings at 450 nm, followed by calculating IFN-γ protein levels in the samples was calculated using the equation obtained from the linear regression of the standard curve.

## Toxicity assessment in BRD4770-treated mice

To evaluate if BRD4770 exerts any generalized- or organ-specific toxicity, vehicle- or BRD4770-treated mice (C57BL/6, BALB/c, and NSG) were subjected to weekly measurement of body weight. The body weights were recorded until the end of the experimental period.

At the end of the experiment, blood was collected by retro-orbital sinus puncture, centrifuged at 6000 rpm for 15 min at room temperature, and the serum was transferred to fresh microcentrifuge tubes. Thereafter, mice were euthanized, their lungs, liver, heart and kidney were collected and fixed in 10% formalin overnight at room temperature. Next, fine sections (5 μm) from tissue/organ samples were prepared and stained with hematoxylin and eosin (H&E). 4× and 20× images were captured using a light microscope.

Creatinine levels in serum collected from vehicle- or BRD4770-treated mice were measured using Creatinine Colorimetric Assay Kit (Cat. No. 700460) from Cayman Chemical, USA, as per the manufacturer's instructions. Briefly, blood was collected by retro-orbital sinus puncture, centrifuged at 6000 rpm for 15 min at room temperature, and the serum was transferred to fresh microcentrifuge tubes. Next, 15 μL of each of the samples, and standards per well were added to a 96-well microplate. Then, 100 μL of Creatinine Reaction Buffer to each well, followed by the addition of 100 μL of Creatinine (serum) Color Reagent to each well and immediately started timing the reaction. Immediately, the absorbance was read and recorded (0 min) at 490 nm using the Biotek Synergy MX Multi Format Microplate Reader (Biotek, USA). The plate was continued incubating at room temperature. At 7 min, the absorbance was read and recorded at 490 nm. The initial (0 min) readings were subtracted from the final (7 min) readings. The creatinine concentration in the samples was calculated using the equation obtained from the linear regression of the standard curve.

Alanine transaminase (ALT) levels in serum collected from vehicle- or BRD4770-treated mice were measured using Alanine Transaminase Colorimetric Activity Assay Kit (Catalog No. 700260) from Cayman Chemical (MI, USA) as per the manufacturer's instructions. Briefly, blood was collected by retro-orbital sinus puncture, centrifuged at 6000 rpm for 15 min at room temperature, and the serum was transferred to fresh microcentrifuge tubes. Next, 150 μL of ALT Substrate, 20 μL of ALT Cofactor, and 20 μL of sample (or ALT positive control) were added to the designated wells on the 96-well microplate. The plate was covered and was incubated at 37 °C for 15 min. The plate cover was removed and quickly initiated the reactions by adding 20 μL of ALT Initiator to all the wells being used. Immediately, the absorbance was measured at 340 nm once every minute for 20 min at 37 °C. The change in absorbance ($\Delta A_{340}$) per minute was determined, and the ALT activity in the samples was calculated.

Aspartate aminotransferase (AST) levels in serum collected from vehicle- or BRD4770-treated mice were measured using Aspartate Aminotransferase Colorimetric Activity Assay Kit (Catalog No. 701640) from Cayman Chemical (MI, USA) as per the manufacturer's instructions. Briefly, blood was collected by retro-orbital sinus puncture, centrifuged at 6000 rpm for 15 min at room temperature, and the serum was transferred to fresh microcentrifuge tubes. Next, 150 μL of AST Substrate, 20 μL of AST Cofactor, and 20 μL of sample (or AST positive control) were added to the designated wells on the 96-well microplate. The plate was covered and was incubated at 37 °C for 15 min. The plate cover was removed and quickly initiated the reactions by adding 20 μL of AST Initiator to all the wells being used. Immediately the absorbance was measured at 340 nm once every minute for 10 min at 37 °C. The change in absorbance ($\Delta A_{340}$) per minute was determined, and the AST activity in the samples was calculated.

## Quantification and statistical analysis

No animal, sample, or data were excluded from the analysis. Experimental animal groups were assigned randomly to vehicle or treatment groups. No blinding was performed. The number of samples included in each experiment is indicated in the figure legends. All experiments were conducted with at least three biological replicates. Results for individual experiments are expressed as mean ± standard error of the mean (SEM). For the analysis of tumor progression in mice, the statistical assessment was performed using the area under the curve method followed by unpaired two-tailed Student's $t$-tests on GraphPad Prism, version 10.0, for Macintosh (GraphPad Software, San Diego, CA, USA; www.graphpad.com). For the survival analysis in Fig. 2H, the log-rank test was used to calculate the $P$ value of comparing Kaplan–Meier curves and the Wald test was used to calculate the $p$ value of Cox proportional hazards regression analysis. The $P$ values for the other experiments were calculated using unpaired two-tailed Student's $t$-tests in GraphPad Prism software, version 10.0, for Macintosh. A $P$ value ≤0.05 was considered statistically significant.

## Data availability

RNA-seq data of this study have been submitted to the National Center for Biotechnology Information (NCBI) Gene Expression Omnibus (GEO), http://www.ncbi.nlm.nih.gov/geo/ (Accession No. GSE196624, URL: https://www.ncbi.nlm.nih.gov/geo/query/acc.cgi?acc=GSE196624).

The source data of this paper are collected in the following database record: biostudies:S-SCDT-10_1038-S44321-025-00357-6.

## Peer review information

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

## Acknowledgements

We would like to acknowledge the National Institutes of Health (NIH) grants R01CA271673 (to NW) and R00CA241110 (to MR) and Department of Defense grant HT9425-24-1-0593 (to RG). We also acknowledge the Small Animal Imaging and Flow cytometry cores of the O'Neal Comprehensive Cancer Center at the University of Alabama at Birmingham.

## Author contributions

**Suresh Chava**: Conceptualization; Data curation; Formal analysis; Validation; Investigation; Visualization; Methodology; Writing—original draft; Writing—review and editing. **Suresh Bugide**: Data curation; Formal analysis; Validation; Investigation; Visualization; Methodology; Writing—original draft; Writing—review and editing. **Parmanand Malvi**: Data curation; Formal analysis; Validation; Investigation; Visualization; Methodology; Writing—original draft;

Writing—review and editing. **Kelly D DeMarco**: Data curation; Formal analysis. **Boyang Ma**: Data curation; Formal analysis. **Chaitanya N Parikh**: Data curation; Formal analysis. **Marcus Ruscetti**: Supervision; Methodology; Writing—original draft; Project administration; Writing—review and editing. **Allan Zajac**: Supervision; Methodology; Writing—original draft; Writing—review and editing. **Guoping Cai**: Conceptualization; Resources; Data curation; Formal analysis; Supervision; Funding acquisition; Validation; Visualization; Methodology; Writing—original draft; Project administration; Writing—review and editing. **Romi Gupta**: Conceptualization; Resources; Formal analysis; Supervision; Funding acquisition; Validation; Visualization; Methodology; Writing—original draft; Project administration; Writing—review and editing. **Narendra Wajapeyee**: Conceptualization; Resources; Data curation; Formal analysis; Supervision; Funding acquisition; Validation; Visualization; Methodology; Writing—original draft; Project administration; Writing—review and editing.

Source data underlying figure panels in this paper may have individual authorship assigned. Where available, figure panel/source data authorship is listed in the following database record: biostudies:S-SCDT-10_1038-S44321-025-00357-6.

## Disclosure and competing interests statement

The authors declare no competing interests.

# Expanded View Figures

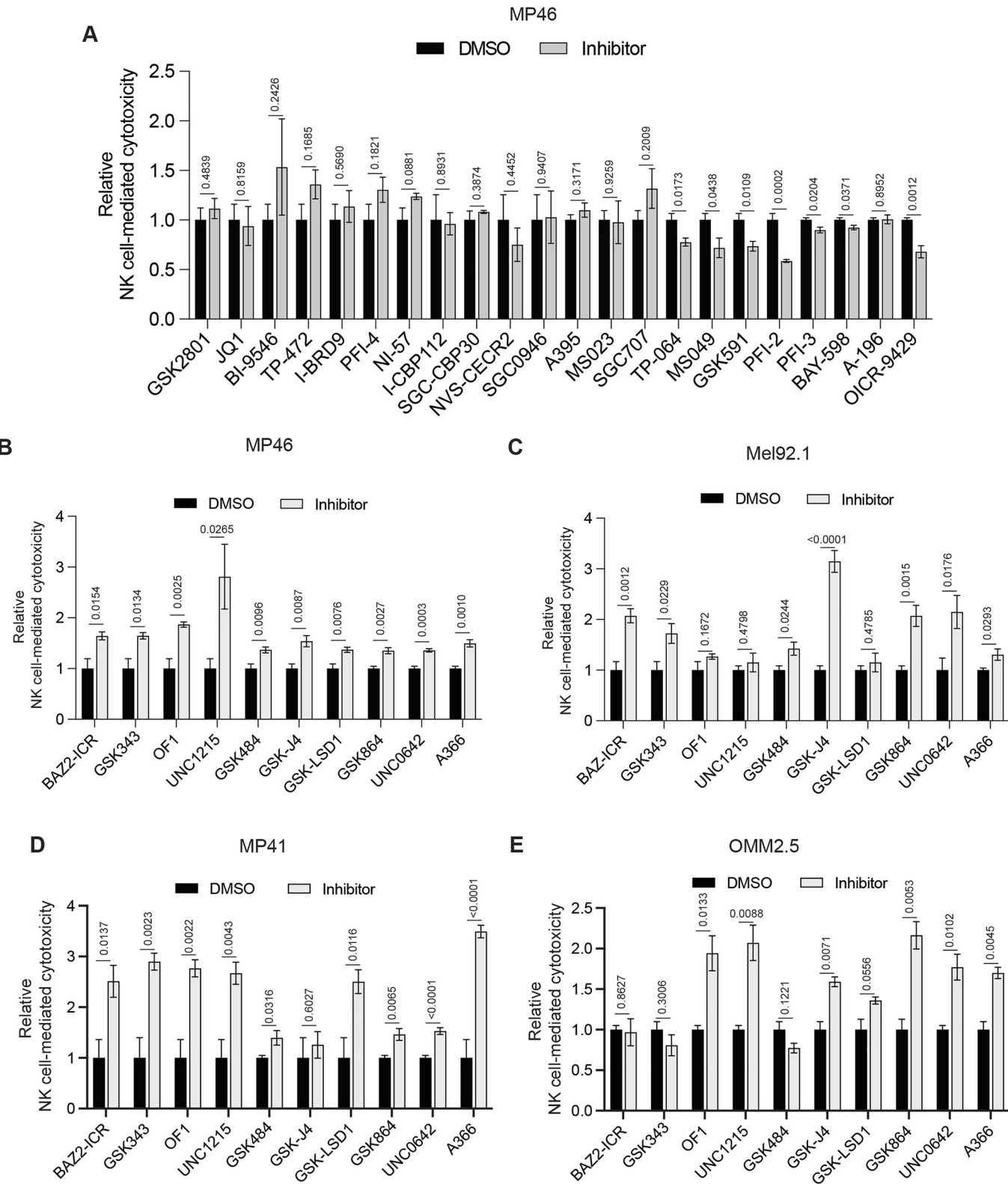

◀ **Figure EV1.    An epigenetic regulator inhibitor screen identifies suppressors of NK cell-mediated cytotoxicity.**

(A) MP46 cells were analyzed for NK cell-mediated cytotoxicity using an LDH-based cytotoxicity assay after treatment with DMSO or indicated small-molecule epigenetic inhibitors for 48 h. Relative NK cell-mediated cytotoxicity for the indicated inhibitors is plotted. ($n = 5$ for DMSO and $n = 3$ for BI-9546; $n = 4$ for DMSO and $n = 3$ for TP-472; $n = 4$ DMSO and $n = 4$ for I-BRD9, $n = 4$ PFI-1, $n = 4$ for I-CBP112, $n = 4$ for NVS-CECR2, $n = 4$ for SGC0946, $n = 4$ for MS023, $n = 4$ for SGC707; $n = 4$ DMSO and $n = 5$ A395; for all other sample DMSO and inhibitors conditions ($n = 5$ each). P values were calculated using unpaired two-tailed Student's *t*-test. (B) MP46 cells were analyzed for NK cell-mediated cytotoxicity using an LDH-based cytotoxicity assay after treatment with DMSO or indicated small-molecule epigenetic inhibitors for 48 h. Relative NK cell-mediated cytotoxicity for the indicated inhibitors is plotted. ($n = 5$ for DMSO and BAZ2-ICR, $n = 5$ for GSK343, $n = 5$ for OF-1 and $n = 5$ UNC1215; for all other samples DMSO $n = 4$ and inhibitors $n = 5$)) P values were calculated using unpaired two-tailed Student's *t*-test. (C) Mel92.1 cells were analyzed for NK cell-mediated cytotoxicity using an LDH-based cytotoxicity assay after treatment with indicated DMSO or small-molecule epigenetic inhibitors for 48 h. Relative NK cell-mediated cytotoxicity for the indicated inhibitors is plotted. ($n = 6$ for DMSO, $n = 6$ UNC0642 and $n = 6$ A366; for all other samples $n = 5$ for DMSO and $n = 5$ for inhibitors). P values were calculated using unpaired two-tailed Student's *t*-test. (D) MP41 cells were analyzed for NK cell-mediated cytotoxicity using an LDH-based cytotoxicity assay after treatment with DMSO or indicated small-molecule epigenetic inhibitors for 48 h. Relative NK cell-mediated cytotoxicity for the indicated inhibitors is plotted. ($n = 6$ for DMSO, $n = 6$ UNC0642 and $n = 6$ A366; for all other samples $n = 5$ for DMSO and $n = 5$ for inhibitors). P values were calculated using unpaired two-tailed Student's *t*-test. (E) OMM2.5 cells were analyzed for NK cell-mediated cytotoxicity using an LDH-based cytotoxicity assay after treatment with indicated DMSO or small-molecule epigenetic inhibitors for 48 h. Relative NK cell-mediated cytotoxicity for the indicated inhibitors is plotted. ($n = 3$ for DMSO and $n = 3$ for inhibitors). P values were calculated using unpaired two-tailed Student's *t*-test. All quantitative data were shown as the mean ± SEM.

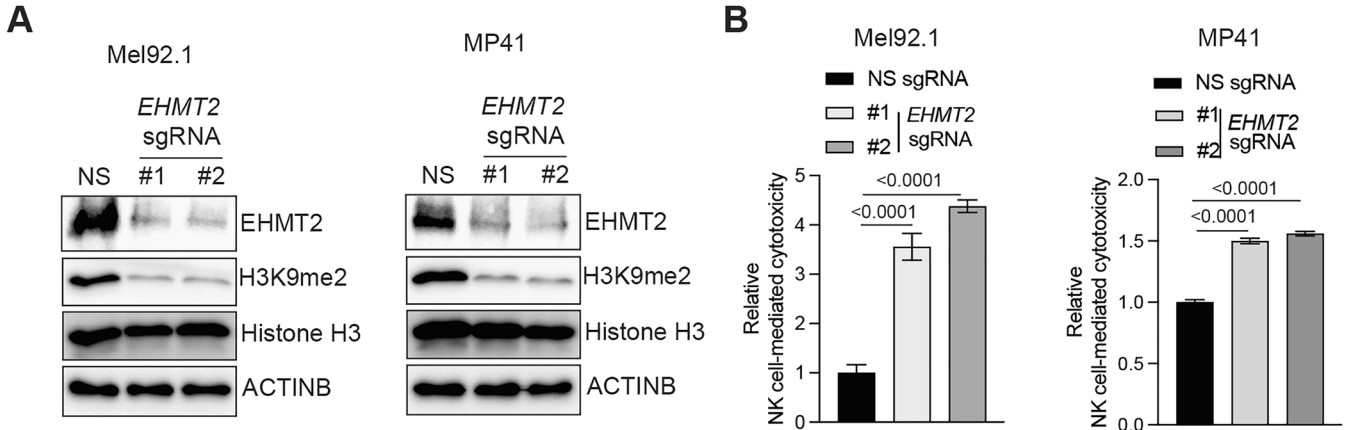

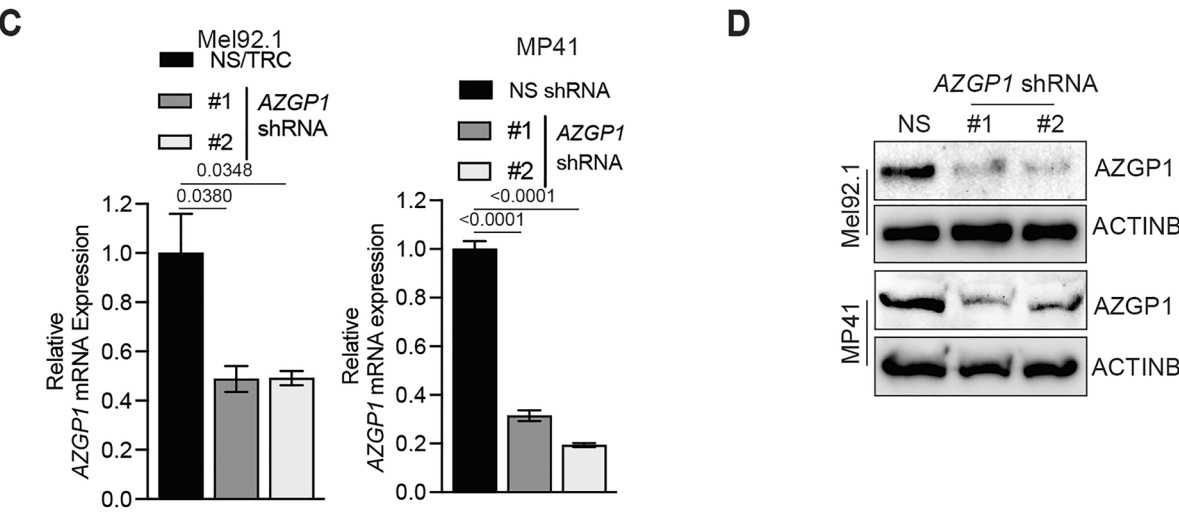

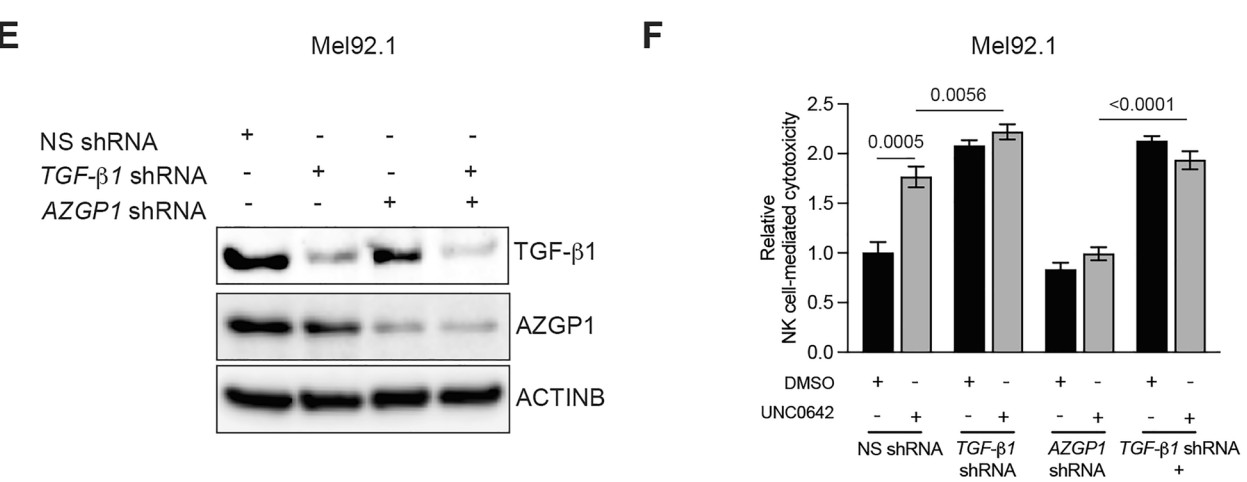

**Figure EV2.   Analysis of EHMT2 knockout cancer cells in NK cell-mediated cytotoxicity assay and validation of AZGP1 knockdown.**

(A) Mel92.1 and MP41 cells expressing either nonspecific single-guide RNA (NS sgRNA) or *EHMT2* sgRNAs were analyzed for the indicated proteins using immunoblotting. Histone H3 and ACTINB were used as loading controls. (B) Mel92.1 and MP41 cells expressing either NS sgRNA or *EHMT2* sgRNAs were analyzed for NK cell-mediated cytotoxicity using an LDH-based cytotoxicity assay. Relative NK cell-mediated cytotoxicity under indicated conditions are plotted. ($n = 6$). P values were calculated using unpaired two-tailed Student's *t*-test. (C) Mel92.1 and MP41 cells expressing either nonspecific (NS) shRNA or *AZGP1* shRNAs were analyzed for mRNA expression for *AZGP1* mRNA were analyzed using RT-qPCR analysis. Relative mRNA expression for indicated genes are plotted. *ACTINB* was used for normalization. ($n = 3$). P values were calculated using unpaired two-tailed Student's *t*-test. (D) Mel92.1 and MP41 cells expressing either NS shRNA or *AZGP1* shRNAs were analyzed for the expression of AZGP1 by immunoblotting. ACTINB was used as a loading control. (E) Mel92.1 cells expressing nonspecific NS shRNA, *AZGP1* shRNA, *TGF-β1* shRNA or both *AZGP1* shRNA and *TGF-β1* shRNA were analyzed by immunoblotting for the indicated proteins. ACTINB was used as a loading control. (F) Mel92.1 cells expressing either NS shRNA, *AZGP1* shRNA, *TGF-β1* shRNA or both *AZGP1* shRNA and *TGF-β1* shRNA were treated with DMSO or EHMT2 inhibitor UNC0642 (1 μM) for 48 h and were analyzed for NK cell-mediated cytotoxicity using an LDH-based cytotoxicity assay. Relative NK cell-mediated cytotoxicity under the indicated conditions is plotted. ($n = 6$). P values were calculated using unpaired two-tailed Student's *t*-test. All quantitative data were presented as the mean ± SEM.

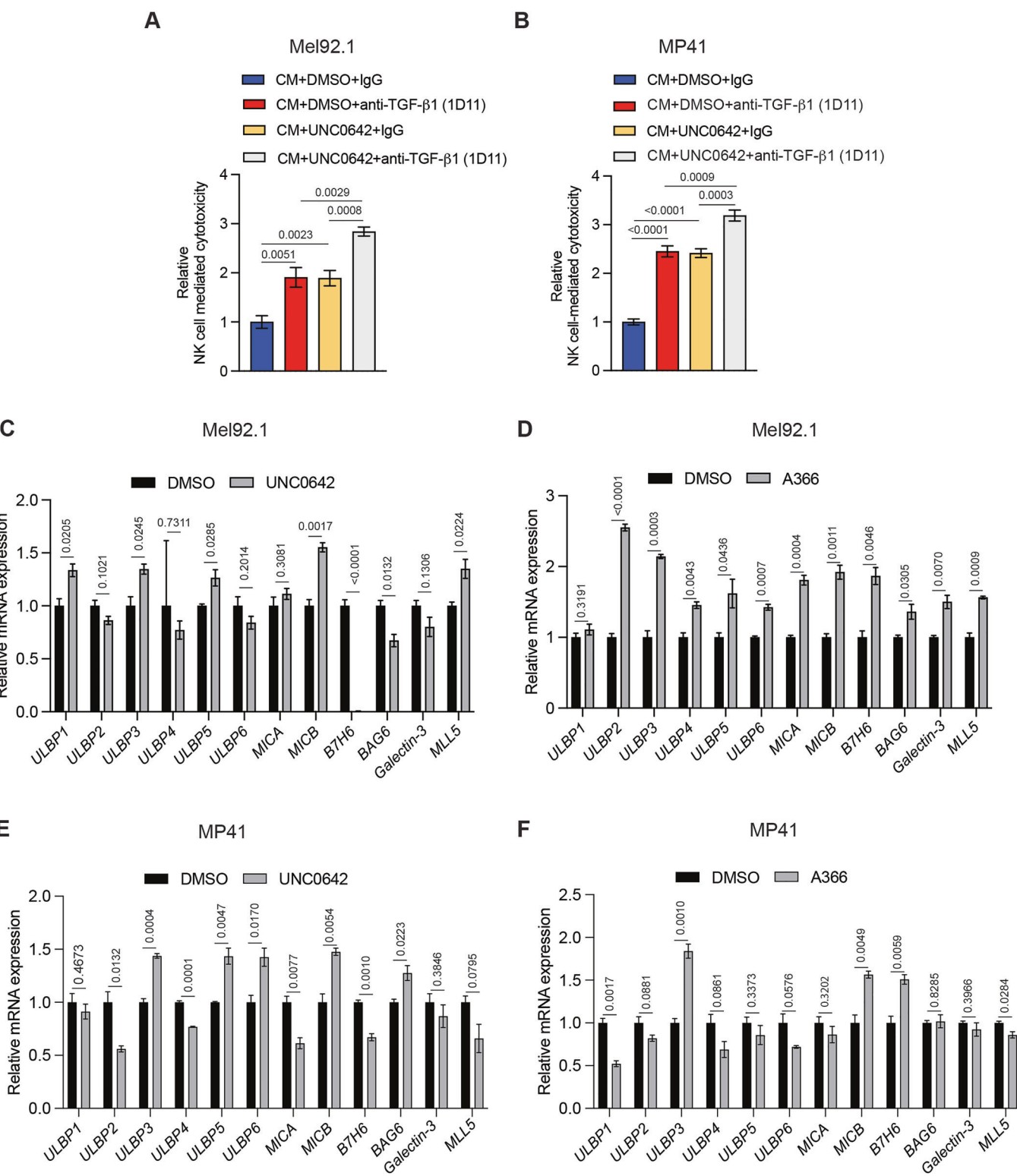

◄ **Figure EV3. Simultaneous inhibition of EHMT2 and neutralization of TGF-β1 augments NK cell-mediated cytotoxicity, and analysis of NK cell ligands in UM cell lines following EHMT2 inhibition.**

(A) Mel92.1 cells were treated with the EHMT2 inhibitor UNC0642 (1 μM) or DMSO for 72 h in Opti-MEM. Following this, conditioned media (CM) was then collected, concentrated, and used to treat NK cells for 24 h in the presence of either control IgG or a TGF-β1-neutralizing antibody (1D11). NK cells were then assessed for cytotoxic activity against Mel92.1 cells using an LDH-based cytotoxicity assay. The relative NK cell-mediated cytotoxicity under the indicated conditions is shown. ($n = 5$). P values were calculated using unpaired two-tailed Student's t-test. (B) MP41 cells were treated with the EHMT2 inhibitor UNC0642 (1 μM) or DMSO for 72 h in Opti-MEM. Following this, conditioned media (CM) was then collected, concentrated, and used to treat NK cells for 24 h in the presence of either control IgG or a TGF-β1-neutralizing antibody (1D11). NK cells were then assessed for cytotoxic activity against MP41 cells using an LDH-based cytotoxicity assay. The relative NK cell-mediated cytotoxicity under the indicated conditions is shown. ($n = 6$). P values were calculated using unpaired two-tailed Student's t-test. (C) Mel92.1 cells treated with DMSO or UNC0642 (3 μM) for 48 h were analyzed for the indicated mRNAs using RT-qPCR. mRNA expression relative to DMSO-treated cells is plotted. ACTINB was used for normalization. ($n = 3$). P values were calculated using unpaired two-tailed Student's t-test. (D) Mel92.1 cells treated with DMSO or A366 (3 μM) for 48 h were analyzed for the indicated mRNAs using RT-qPCR. mRNA expression relative to DMSO-treated cells is plotted. ACTINB was used for normalization. ($n = 3$). P values were calculated using unpaired two-tailed Student's t-test. (E) MP41 cells treated with DMSO or UNC0642 (3 μM) for 48 h were analyzed for the indicated mRNAs using RT-qPCR. mRNA expression relative to DMSO-treated cells is plotted. ACTINB was used for normalization. ($n = 3$). P values were calculated using unpaired two-tailed Student's t-test. (F) MP41 cells treated with DMSO or A366 (3 μM) for 48 h were analyzed for the indicated mRNAs using RT-qPCR. mRNA expression relative to DMSO-treated cells is plotted. ACTINB was used for normalization. ($n = 3$). P values were calculated using unpaired two-tailed Student's t-test. All quantitative data were presented as mean ± SEM.

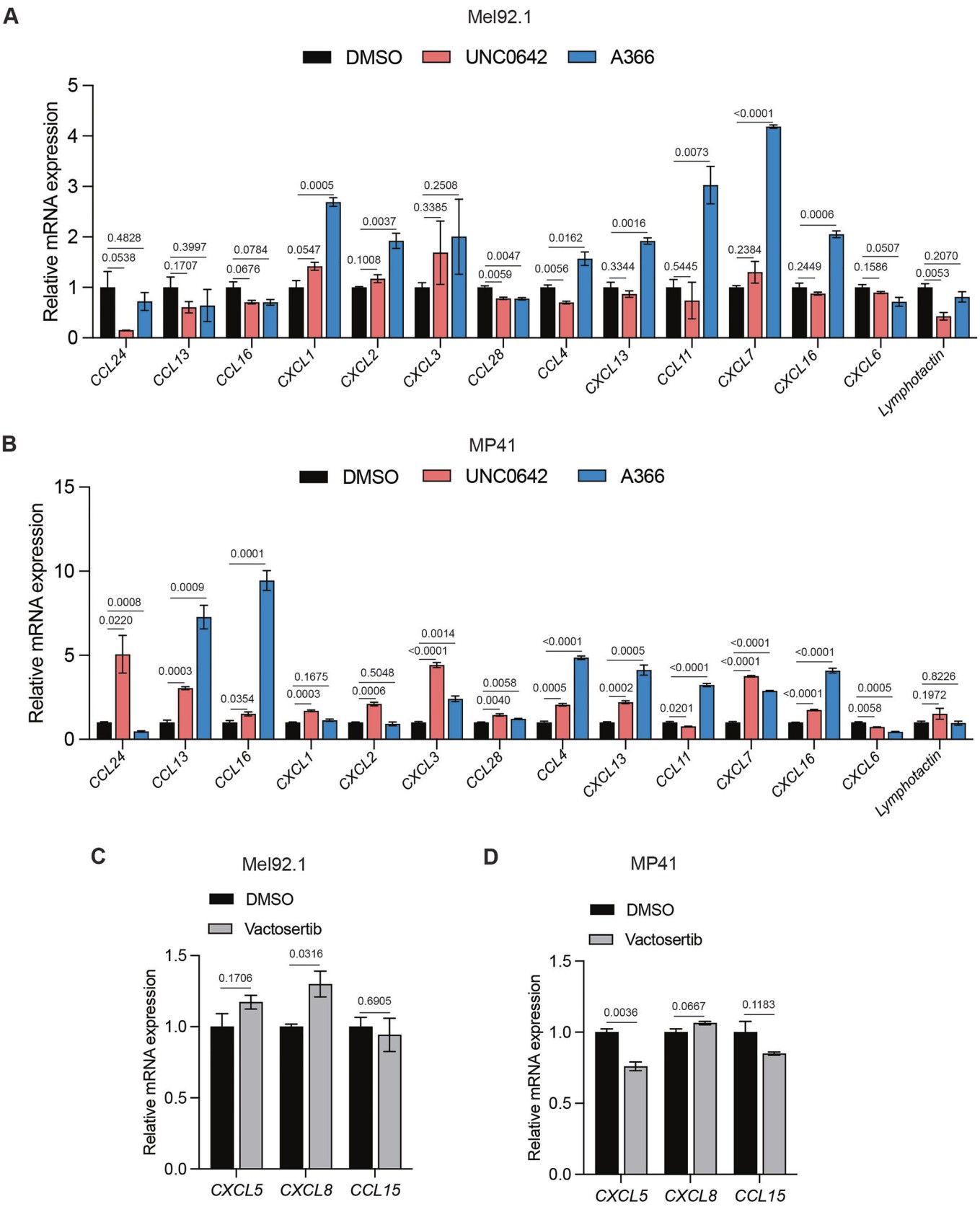

◀  **Figure EV4. Analysis of human chemokines expression in uveal melanoma cell lines following EHMT2 inhibitor treatment.**

(A) Mel92.1 cells treated with DMSO, or UNC0642 (1 μM), A366 (1 μM) for 72 h were analyzed for indicated mRNAs using RT-qPCR. mRNA expression relative to DMSO-treated cells is plotted. *ACTINB* was used for normalization. ($n = 3$). *P* values were calculated using unpaired two-tailed Student's *t*-test. (B) MP41 cells treated with DMSO, or UNC0642 (1 μM), A366 (1 μM) for 72 h were analyzed for indicated mRNAs using RT-qPCR. mRNA expression relative to DMSO-treated cells is plotted. *ACTINB* was used for normalization. ($n = 3$). *P* values were calculated using unpaired two-tailed Student's *t*-test. (C) Mel92.1 cells treated with vactosertib (1 μM) for 48 h were analyzed for indicated mRNAs using RT-qPCR. mRNA expression relative to DMSO-treated cells is plotted. *ACTINB* was used for normalization. ($n = 3$). *P* values were calculated using unpaired two-tailed Student's *t*-test. (D) MP41 cells were treated with vacotosertib (1 μM) for 48 h were analyzed for the indicated mRNAs using RT-qPCR. mRNA expression relative to DMSO-treated cells is plotted. *ACTINB* was used for normalization. ($n = 3$). *P* values were calculated using unpaired two-tailed Student's *t*-test. All quantitative data were shown as the mean ± SEM.

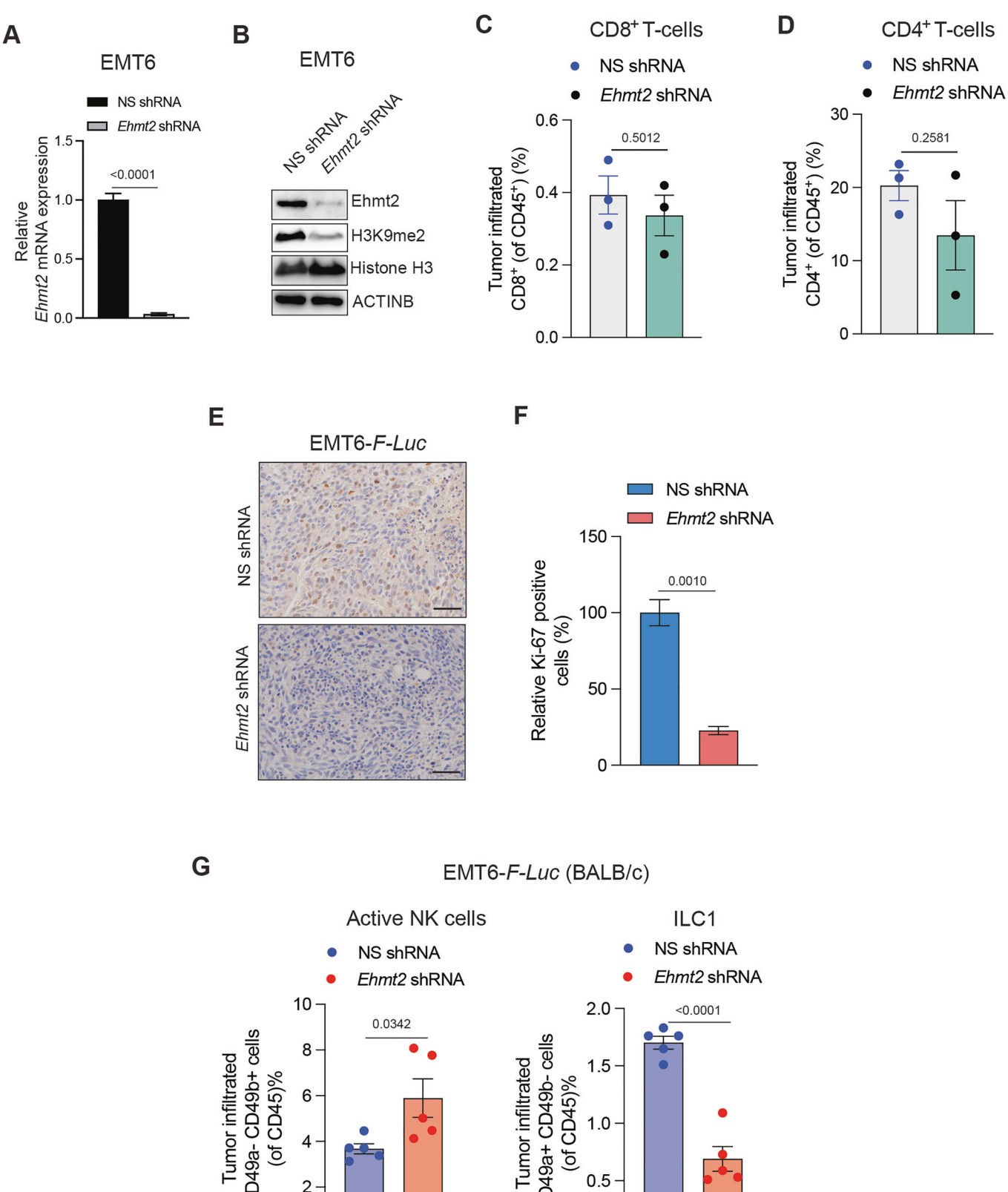

◀ **Figure EV5.  Analysis of EMT6 tumors.**

(**A**) EMT6 cells expressing either nonspecific (NS) shRNA or *Ehmt2* shRNA were analyzed for mRNA expression for *Ehmt2* mRNA were analyzed using RT-qPCR analysis. Relative mRNA expression is plotted as the mean ± SEM. *Actinb* was used for normalization. ($n = 3$). *P* values were calculated using unpaired two-tailed Student's *t*-test. (**B**) EMT6 cells expressing either NS shRNA or *Ehmt2* shRNA were analyzed for the expression of the indicated proteins by immunoblotting. Histone H3 and ACTINB were used as loading controls. (**C**) Firefly luciferase (*F-Luc*)-labeled EMT6 cells expressing either NS shRNA or *Ehmt2* shRNA were orthotopically injected into the mammary fat pad of female BALB/c mice. Measurement of tumor-infiltered CD8$^+$ T-cells (%) in the EMT6 tumors under indicated conditions using FACS analysis. ($n = 3$). *P* values were calculated using unpaired two-tailed Student's *t*-test. (**D**) Firefly luciferase (*F-Luc*)-labeled EMT6 cells expressing either NS shRNA or *Ehmt2* shRNA were orthotopically injected into the mammary fat pad of female BALB/c mice. Measurement of tumor-infiltered CD4$^+$ T-cells (%) in the EMT6 tumors under indicated conditions using FACS analysis and plotted. ($n = 3$). *P* values were calculated using unpaired two-tailed Student's *t*-test. (**E**) Ki-67 expression was analyzed by immunohistochemistry (IHC) in EMT6 tumor sections expressing NS shRNA and *Ehmt2* shRNA. Representative Ki-67 staining images of EMT6 tumor sections expressing NS shRNA and *Ehmt2* shRNA at 20× magnifications are shown. Scale bar, 50 µm. (**F**) Quantitation of Ki-67 staining for the experiment presented in panel (**E**). ($n = 3$). *P* values were calculated using unpaired two-tailed Student's *t*-test. (**G**) EMT6 cells expressing NS shRNA or *Ehmt2* shRNAs were injected subcutaneously into BALB/c mice. Measurement of tumor-infiltrated NK cells (Lin⁻CD49a⁻CD49b⁺) and ILC1 (Lin⁻CD49a⁺CD49b⁻) in the EMT6 tumors under the indicated conditions using FACS analysis and plotted. ($n = 5$). *P* values were calculated using unpaired two-tailed Student's *t*-test. All quantitative data were shown as the mean ± SEM.

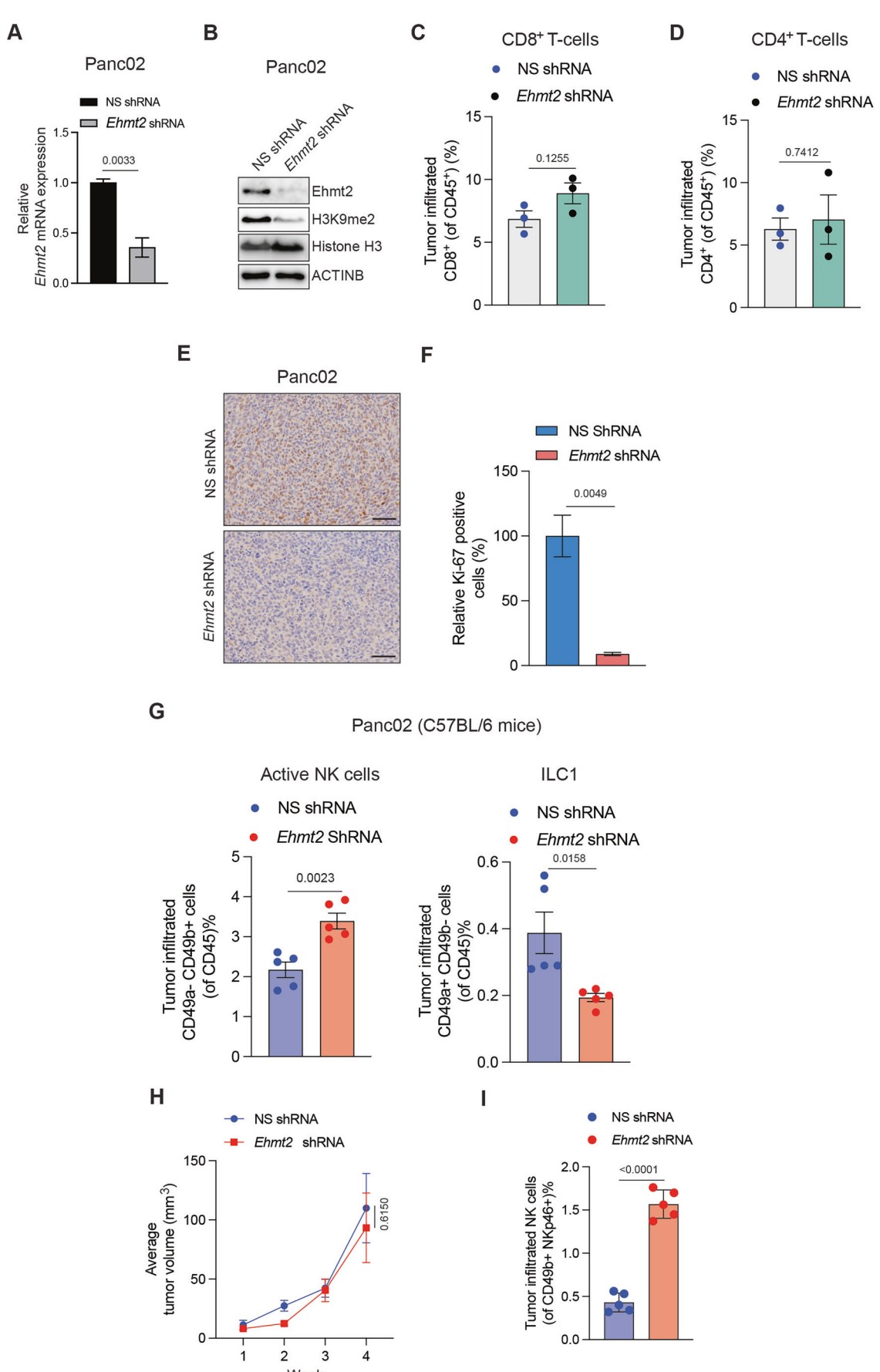

◀ **Figure EV6.   Analysis of Panc02 tumors.**

(A) Panc02 cells expressing either nonspecific (NS) shRNA or *Ehmt2* shRNA were analyzed for mRNA expression for *Ehmt2* mRNA were analyzed using RT-qPCR analysis. Relative mRNA expression is plotted. *Actinb* was used for normalization. ($n = 3$). $P$ values were calculated using unpaired two-tailed Student's $t$-test. (B) Panc02 cells expressing either NS shRNA or *Ehmt2* shRNA were analyzed for the expression of the indicated proteins by immunoblotting. Histone H3 and ACTINB were used as loading controls. (C) Panc02 cells expressing NS shRNA or *Ehmt2* shRNAs were injected subcutaneously into C57BL/6 mice. Measurement of tumor-infiltered CD8$^+$ T-cells (%) in the Panc02 tumors under indicated conditions using FACS analysis and plotted. ($n = 3$). $P$ values were calculated using unpaired two-tailed Student's t-test. (D) Panc02 cells expressing NS shRNA or *Ehmt2* shRNAs were injected subcutaneously into C57BL/6 mice. Measurement of tumor-infiltered CD4$^+$ T-cells cells (%) in the Panc02 tumors under indicated conditions using FACS analysis and plotted. ($n = 3$). $P$ values were calculated using unpaired two-tailed Student's $t$-test. (E) Ki-67 expression was analyzed by immunohistochemistry (IHC) in Panc02 tumor sections expressing NS shRNA and *Ehmt2* shRNA. Representative Ki-67 staining images of Panc02 tumor sections expressing NS shRNA and *Ehmt2* shRNA at 20× magnifications are shown. Scale bar, 50 μm. (F) Quantitation of Ki-67 staining for the experiment presented in panel (E) and plotted. ($n = 3$). $P$ values were calculated using unpaired two-tailed Student's $t$-test. (G) Panc02 cells expressing NS shRNA or *Ehmt2* shRNAs were injected subcutaneously into C57BL/6 mice. Measurement of tumor-infiltrated NK cells (Lin$^-$CD49a$^-$CD49b$^+$), and ILC1 (Lin$^-$CD49a$^+$CD49b$^-$) in the Panc02 tumors under the indicated conditions using FACS analysis and plotted. ($n = 5$). $P$ values were calculated using unpaired two-tailed Student's $t$-test. (H) Panc02 cells expressing either NS shRNA or *Ehmt2* shRNA were injected subcutaneously into C57BL/6 mice. Tumor volumes at the indicated times are plotted. ($n = 5$). For the analysis of tumor progression in mice, the statistical assessment was performed using the area under the curve method, followed by unpaired two-tailed Student's $t$-tests. (I) Measurement of tumor-infiltrated NK cells (%) in the Panc02 tumors under the indicated conditions using FACS analysis and plotted. ($n = 5$). $P$ values were calculated using unpaired two-tailed Student's $t$-test. All quantitative data were shown as the mean ± SEM.

**A**

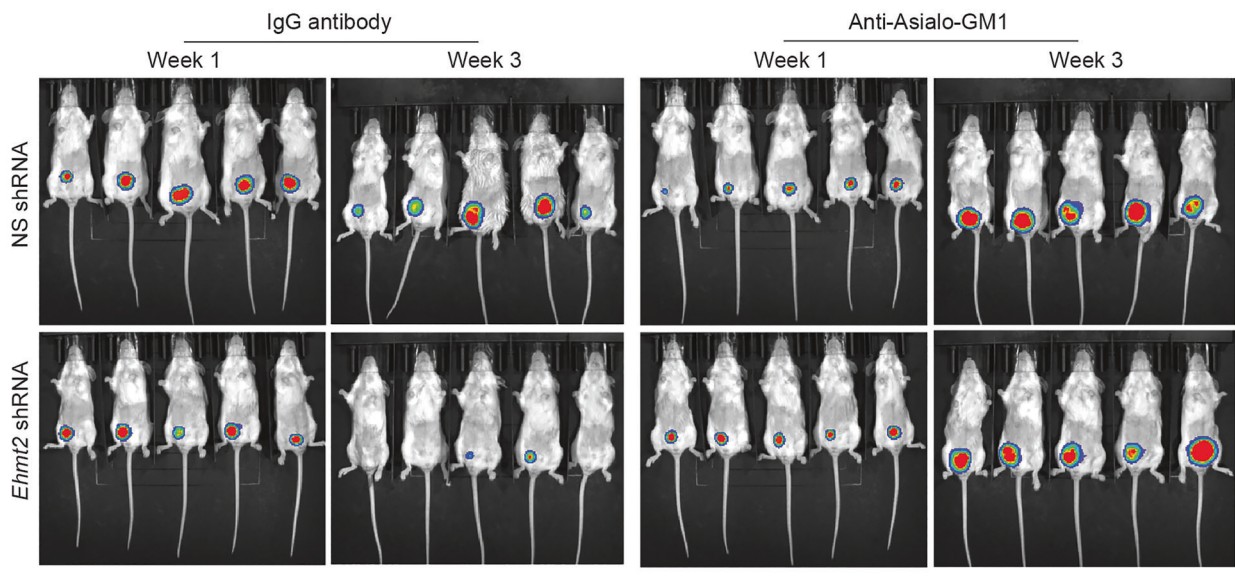

**B**

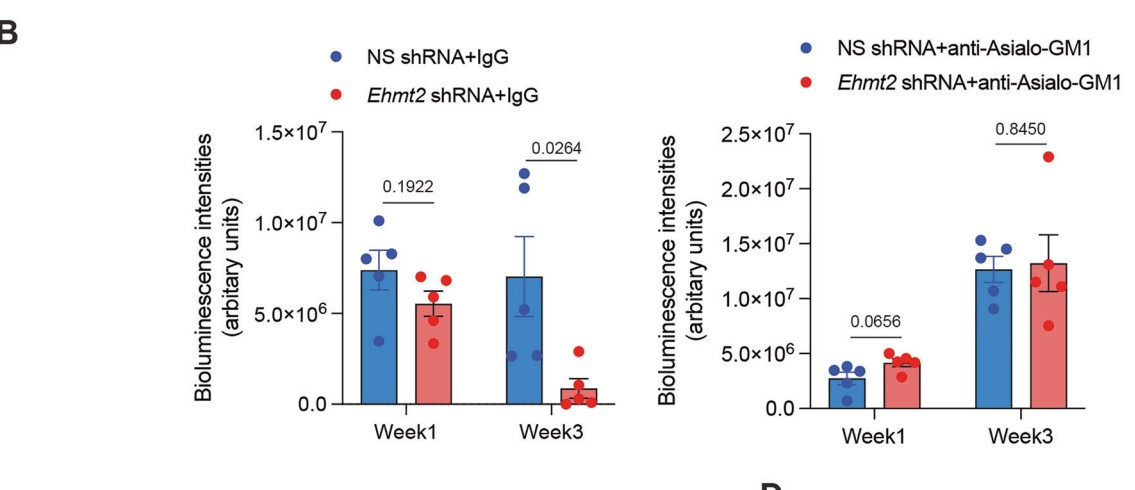

**C**

Tumor

**D**

Spleen

◀ **Figure EV7. NK cells are necessary for *Ehmt2* knockdown-mediated tumor suppression in the immunocompetent syngeneic mouse model of EMT6.**

(A) Firefly luciferase (*F-Luc*)-labeled EMT6 cells expressing either nonspecific (NS) shRNA or *Ehmt2* shRNA were injected orthotopically into the mammary fat pad of female BALB/c mice ($n = 5$) and treated with either IgG isotype control antibody (100 μg/mouse) or anti-Asialo-GM1 antibody (100 μg/mouse) one day before EMT6 cell line injections and then once a week during the course of the experiment. Bioluminescence images of mice at the indicated weeks after injection are shown. (B) Bioluminescence intensities of the mice at the indicated weeks under the indicated conditions for the experiment presented in panel (A) are plotted. ($n = 5$). P values were calculated using unpaired two-tailed Student's *t*-test. (C) Measurement of tumor-infiltrated NK cells (%) in EMT6 tumors expressing either NS shRNA or *Ehmt2* shRNA from BALB/c mice treated with IgG isotype control or anti-Asialo-GM1 antibodies using FACS analysis. ($n = 5$). P values were calculated using unpaired two-tailed Student's *t*-test. (D) Measurement of relative NK cells (%) in spleens from mice harboring EMT6 tumors expressing either NS shRNA or *Ehmt2* shRNA from mice treated with IgG isotype control or anti-Asialo-GM1 antibodies using FACS analysis. ($n = 5$). P values were calculated using unpaired two-tailed Student's *t*-test. All quantitative data were shown as the mean ± SEM.

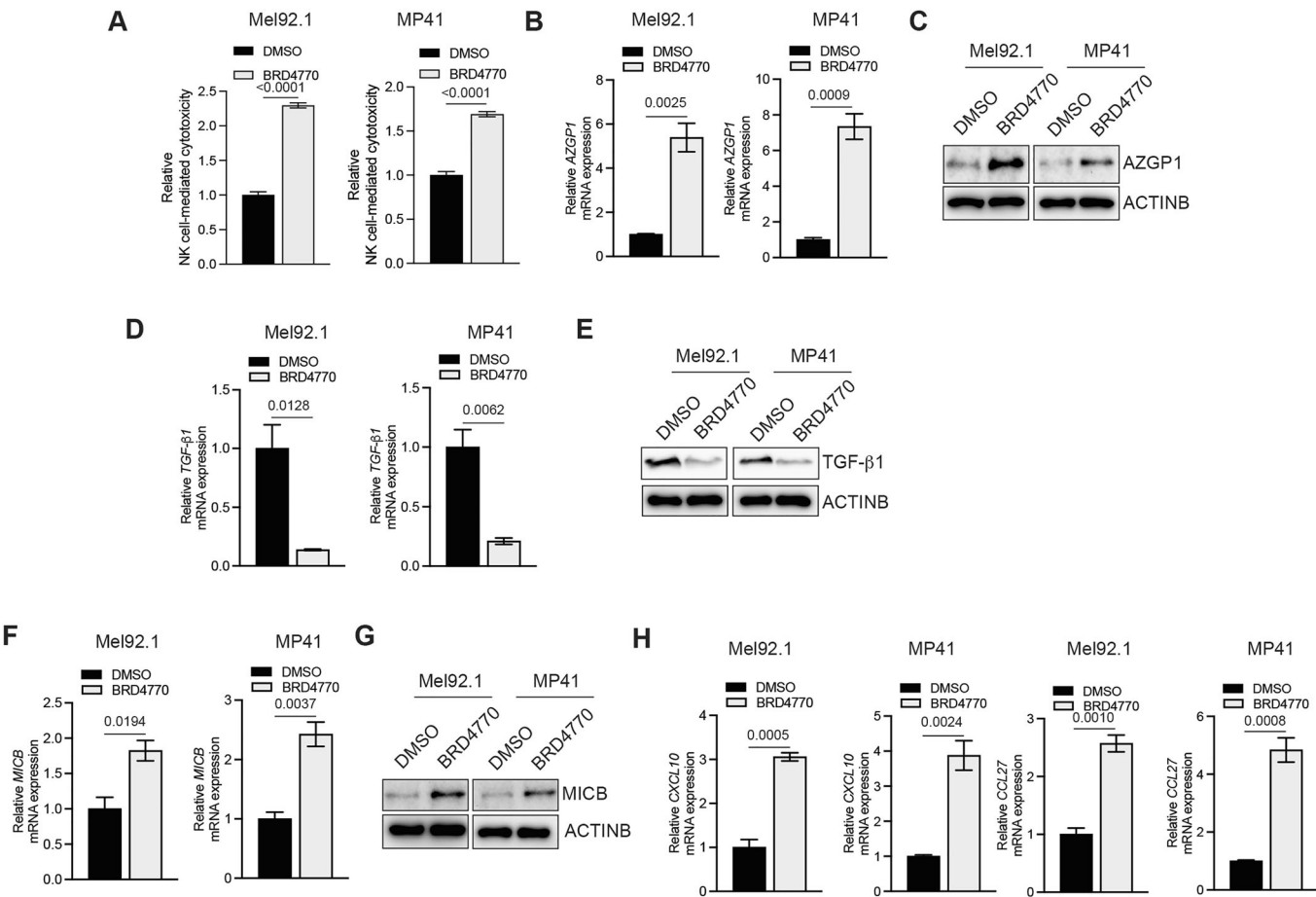

**Figure EV8. The EHMT2 inhibitor BRD4770 enhances NK cell-mediated cytotoxicity, upregulates AZGP1, suppresses TGF-β1, and induces CXCL10 and CCL27 expression.**

(A) Mel92.1 and MP41 cells were treated with DMSO or BRD4770 (5 μM) for 48 h and NK cell-mediated cytotoxicity was measured using an LDH-based cytotoxicity assay. Relative NK cell-mediated cytotoxicity under the indicated conditions for indicated UM cell lines is plotted. ($n = 5$). $P$ values were calculated using unpaired two-tailed Student's $t$-test. (B) Mel92.1 and MP41 cells treated with DMSO or BRD4770 (5 μM) for 48 h and were analyzed for *AZGP1* mRNA using RT-qPCR. mRNA expression relative to DMSO-treated cells is plotted. *ACTINB* was used for normalization. (n = 3). $P$ values were calculated using unpaired two-tailed Student's $t$-test. (C) Mel92.1 and MP41 cells were treated with DMSO or BRD4770 (5 μM) for 48 h and were analyzed for AZGP1 expression by immunoblotting. ACTINB was used as a loading control. (D) Mel92.1 and MP41 cells were treated with DMSO or BRD4770 (5 μM) for 48 h and were analyzed for *TGF-β1* mRNA using RT-qPCR. mRNA expression relative to DMSO-treated cells is plotted. *ACTINB* was used for normalization. (n = 3). $P$ values were calculated using unpaired two-tailed Student's $t$-test. (E) Mel92.1 and MP41 cells were treated with DMSO or BRD4770 (5 μM) for 48 h and were analyzed for TGF-β1 expression by immunoblotting. ACTINB was used as a loading control. (F) Mel92.1 and MP41 cells were treated with DMSO or BRD4770 (5 μM) for 48 h and were analyzed for *MICB* mRNA using RT-qPCR. mRNA expression relative to DMSO-treated cells is plotted. *ACTINB* was used for normalization. (n = 3). $P$ values were calculated using unpaired two-tailed Student's $t$-test. (G) Mel92.1 and MP41 cells were treated with DMSO or BRD4770 (5 μM) for 48 h and were analyzed for MICB expression by immunoblotting. ACTINB was used as a loading control. (H) Mel92.1 and MP41 cells were treated with DMSO or BRD4770 (5 μM) for 48 h and were analyzed for *CXCL10* and *CCL27* mRNAs using RT-qPCR. mRNA expression relative to DMSO -treated cells is plotted. *ACTINB* was used for normalization. (n = 3). $P$ values were calculated using unpaired two-tailed Student's $t$-test. All quantitative data were shown as the mean ± SEM.

