## [Peer Review File · EMBO Molecular Medicine]

Loss of EHMT2 Enhances NK Cell-Driven Anti-Tumor Immunity through TGF- β 1 Suppression

Suresh Chava, Suresh Bugide, Parmanand Malvi, Kelly DeMarco, Boyang Ma, Chaitanya Parikh, Marcus Ruscetti, Allan Zajac, Guoping Cai, Romi Gupta, and Narendra Wajapeyee

Corresponding author(s): Narendra Wajapeyee (nwajapey@uab.edu) , Romi Gupta (romigup@uab.edu)

Review Timeline:

Submission Date:	21st Feb 25
Editorial Decision:	19th Mar 25
Revision Received:	2nd Oct 25
Editorial Decision:	3rd Nov 25
Revision Received:	14th Nov 25
Accepted:	24th Nov 25

Editor: Lise Roth

Transaction Report:

19th Mar 2025

Dear Prof. Wajapeyye,

Thank you for the submission of your manuscript to EMBO Molecular Medicine. We have now received feedback from the three reviewers who agreed to evaluate your manuscript. As you will see from the reports below, the referees acknowledge the interest of the study and are overall supporting publication of your work pending appropriate revisions.

Addressing the reviewers' concerns in full will be necessary for further considering the manuscript in our journal, and acceptance of the manuscript will entail a second round of review. EMBO Molecular Medicine encourages a single round of revision only and therefore, acceptance or rejection of the manuscript will depend on the completeness of your responses included in the next, final version of the manuscript. For this reason, and to save you from any frustrations in the end, I would strongly advise against returning an incomplete revision.

We are expecting your revised manuscript within three to four months, if you anticipate any delay, please contact us.

We require:

4) A .docx formatted letter INCLUDING the reviewers' reports and your detailed point-by-point responses to their comments. As part of the EMBO Press transparent editorial process, the point-by-point response is part of the Review Process File (RPF), which will be published alongside your paper.

5) A complete author checklist, which you can download from our author guidelines (<https://www.embopress.org/page/journal/17574684/authorguide#submissionofrevisions>). Please insert information in the checklist that is also reflected in the manuscript. The completed author checklist will also be part of the RPF.

6) All Materials and Methods need to be described in the main text using our 'Structured Methods' format. According to this format, the Methods section includes a Reagents and Tools Table (listing key reagents, experimental models, software and relevant equipment and including their sources and relevant identifiers) followed by a Methods and Protocols section describing the methods, ideally using a step-by-step protocol format. The aim is to facilitate adoption of the methodologies across labs. Please download and fill our Reagents and Tools Table template (.docx), which you can find in our author guidelines:

<https://www.embopress.org/doi/10.15252/msb.20178071>

7) Please note that all corresponding authors are required to supply an ORCID ID for their name upon submission of a revised manuscript.

8) It is mandatory to include a 'Data Availability' section after the Materials and Methods. Before submitting your revision, primary datasets produced in this study need to be deposited in an appropriate public database, and the accession numbers and database listed under 'Data Availability'. Please remember to provide a reviewer password if the datasets are not yet public (see <https://www.embopress.org/page/journal/17574684/authorguide#dataavailability>).

9) For data quantification: please specify the name of the statistical test used to generate error bars and P values, the number (n) of independent experiments (specify technical or biological replicates) underlying each data point and the test used to calculate p-values in each figure legend. The figure legends should contain a basic description of n, P and the test applied. Graphs must include a description of the bars and the error bars (s.d., s.e.m.). Please provide exact p values.

10) Our journal encourages inclusion of *data citations in the reference list* to directly cite datasets that were re-used and obtained from public databases. Data citations in the article text are distinct from normal bibliographical citations and should directly link to the database records from which the data can be accessed. In the main text, data citations are formatted as follows: "Data ref: Smith et al, 2001" or "Data ref: NCBI Sequence Read Archive PRJNA342805, 2017". In the Reference list, data citations must be labeled with "[DATASET]". A data reference must provide the database name, accession number/identifiers and a resolvable link to the landing page from which the data can be accessed at the end of the reference. Further instructions are available at .

11) We replaced Supplementary Information with Expanded View (EV) Figures and Tables that are collapsible/expandable online. EV Figures should be cited as 'Figure EV1, Figure EV2' etc... in the text and their respective legends should be included in the main text after the legends of regular figures.

12) The paper explained: EMBO Molecular Medicine articles are accompanied by a summary of the articles to emphasize the major findings in the paper and their medical implications for the non-specialist reader. Please provide a draft summary of your article highlighting

13) Author contributions: CRedit has replaced the traditional author contributions section because it offers a systematic machine readable author contributions format that allows for more effective research assessment. Please remove the Authors Contributions from the manuscript and use the free text boxes beneath each contributing author's name in our system to add specific details on the author's contribution. More information is available in our guide to authors.

Please also suggest a visual abstract to illustrate your article as a PNG file 550 px wide x 300-600 px high. A cropped portion of this image will serve as thumbnail for the table of content on our webpage.

16) As part of the EMBO Publications transparent editorial process initiative (see our Editorial at <http://embomolmed.embopress.org/content/2/9/329>), EMBO Molecular Medicine will publish online a Review Process File (RPF) to accompany accepted manuscripts.

In the event of acceptance, this file will be published in conjunction with your paper and will include the anonymous referee reports, your point-by-point response and all pertinent correspondence relating to the manuscript. Let us know whether you agree with the publication of the RPF and as here, if you want to remove or not any figures from it prior to publication. Please note that the Authors checklist will be published at the end of the RPF.

I look forward to receiving your revised manuscript.

Yours sincerely,

Lise Roth
Lise Roth, PhD
Senior Editor
EMBO Molecular Medicine

***** Reviewer's comments *****

Referee #1 (Remarks for Author):

This is a well-written manuscript with a cohesive narrative well-supported by the data. Rigor with regard to validating findings across multiple cell lines using multiple drugs and complementary genetic approaches is noted and appreciated.

1. Authors should move the last paragraph of the Introduction to the Discussion.
2. Authors should define the impact of EHMT2 inhibition on tumor cell-intrinsic growth properties, so the relative contribution of cell intrinsic versus NK cell-mediated killing can be fully understood.
3. Western blots are sub-publication quality. Messy blots need to be re-run to generate improved images.
4. Phenotype of NK cell mediated cytotoxicity reported in Figure 1 is much stronger in experiments using pharmacological agents as compared to shRNA. Presumably this difference is due to incomplete inactivation with shRNAs; CRISPR-mediated KO could provide an alternative approach that may generate a stronger phenotype. This would strengthen the argument about EHMT2's importance in this context.
5. In Figures 3 and 4, the manuscript could be strengthened by proving that the effect of TGFb1 in the context of AZGP1 depletion, which according to their model is required to unleash elevated TGFb1.
6. Figure 6 would be strengthened by quantitation of tumor growth rates in tumors from the mouse models (eg Ki67, BrdU, etc).
7. Authors could shorten the summary of their findings in the Discussion, as this is redundant.

Referee #2 (Comments on Novelty/Model System for Author):

Technical quality: some of the assays need better comparison side by side and not in separate experiments. Also the NK cell cytotoxicity should be assessed in different ratios to be convincing. Medical impact would be improved by using same inhibitors in vitro and in vivo.

Referee #2 (Remarks for Author):

The authors analyzed the impact of EHMT2 inhibition or knockdown in different cancer cell lines on the susceptibility of these cells to NK cell-mediated lysis. By employing different techniques, RNA sequencing, in vitro and in vivo approaches they could show an important contribution of EHMT2 to escape of tumor cells via induction of TGFb1. While overall it is an interesting and

complex manuscript, some experiments seem to lack direct connection and some results are not reproduced in same cell lines. Several issues need to be addressed:

In the introduction (last paragraph before summary of this study) it is unclear whether EHMT2 has these effects in the immune cells or in the tumor cells - please specify. In the summary at the end of introduction: In this study... - the authors need to specify from the beginning whether they focus on EHMT2 in tumor cells or NK cells. Loss of.. needs to be mentioned where the loss was.

Results:

General comment: The LDH based assay to assess NK cell cytotoxicity might be affected by differences in NK cell-derived LDH. They might show different levels dependent whether they are seeded alone or killing the tumor cells. The authors should include different ratios of NK to tumor cells for most crucial experiments to confirm the specificity of their signal.

Figure 2 I - please include AZGP1 antibody and not only V5

Figure 3L: This is well known that TGF β generally suppresses NK cell function. The better experiment would be to neutralize TGF β in the same experiment next to EZHT2 KD or inhibition - does it reach same level? Is there still room for synergy?

Figure 4: The transition here lack red line. It needs to be better explained why authors now look at the ligands on target cells while before claimed they treat NK cells with conditioned medium?

4A-C Levels of NKG2D ligands need to be shown on the surface using flow cytometry to make sure that they would have a functional consequence

4E The knockdown of MICB does not have any effect on NK susceptibility in cells not treated with the inhibitor? Could the authors comment on that? Or is the quantification misleading?

4F-K In this context it would mainly make sense to look at NKG2D. If the authors claim that NKG2DL are crucial in their system. Why NCR3 was chosen and not other receptor

Figure 5 It needs to be better explained why the chemokines were analyzed as in these experiments the migration does not seem relevant. The enhanced percentage of NK cells in tumor in vivo (Fig. 6) may be a consequence of smaller tumors and thereby a higher immune cell ratio.

The sentence needs to be more precise: We found that TNBC cells treated with EHMT2 inhibitors (A366 and UNC0642) showed increased NK cell-mediated cytotoxicity is not phrase correctly. Rather show increased susceptibility/sensitivity? Please correct through the manuscript.

In vivo: Why UM cell lines were not used in the mouse model?

6H and I: There is a clear significant difference between the tumor growth in NSG mice. To make a clear statement that the effect observed is not tumor intrinsic one would need to run this experiment side by side or even more clear perform the experiment in immunodeficient mice with or without adoptive NK cell transfer to check whether there is really an additive effect of NK cells and knockdown.

In vivo: It is unclear why the authors used a different inhibitor for in vivo treatment without including it in any in vitro assays. Please repeat key findings of your paper with the new inhibitor used in vivo.

Referee #3 (Comments on Novelty/Model System for Author):

One concern regarding the medical impact is whether systemic application of EHMT2 inhibitors will be tolerated by patients as well as in mice.

Referee #3 (Remarks for Author):

The manuscript by Chava et al. reports that inhibition or loss of the histone methyltransferase EHMT2 in tumor cells enhances enhanced NK cell-mediated elimination of diverse cancers, including uveal melanoma, breast cancer, and pancreatic cancer. EHMT2 loss increased AZGP1 and decreased TGF- β 1 levels, resulting in autocrine elevation of NKG2D ligands MICB and ULBP3, chemokines in cancer cells, and paracrine stimulation of NK cell function. The manuscript is written in a comprehensive manner, the experimental setups appear are sound and the proposed mechanism is potentially of interest.

Given that the authors claim to report a, at least partially, TGF- β 1-driven phenomenon, it will be important to take in account the existing conceptual framework regarding the impact of TGF- β 1 on the phenotype and function of Type I Innate Lymphoid Cells (ILCs), including NK cells. In addition, in certain setting the readouts to determine NK cell cytotoxicity need some refinement. Therefore, the impact of the study is likely to increase by addressing the questions below experimentally.

Major:

It is important to take in account the potential impact on the NK cell and ILC1 phenotypes respectively (doi: 10.1038/ni.3800.). In the murine setting, paracrine TGF- β -signaling drives conversion of NK cells (CD49a-CD49b+Eomes+) into intermediate type 1 innate lymphoid cell (intILC1) (CD49a+CD49b+Eomes+) populations and ILC1 (CD49a+CD49b-Eomesint) populations in the tumor microenvironment. Importantly, intILC1s and ILC1s are unable to control local tumor growth, whereas NK cells maintain tumor immunosurveillance. ILCs 1 are T-Bet High, produce cytokines like TNF and GM-CSF, NK cells are Eomes high and cytotoxic, both subsets are capable of producing high amounts of IFN- γ . Therefore, these features should be included in the in vitro and in vivo analysis of the manuscript. Even more so, since human tumor ILC1-like cells (Lin- (CD19-CD14-CD3-)CD56+CXCR6+ ILC1-like cells) have been described in human tumors as well.

Along these lines and concerning Figure 4: To sufficiently address NK cell function in this setting (exposure to TGF-beta), the authors should perform a cytotoxicity/killing assay in addition measuring IFN- γ levels and the expression of surface receptors in NK cells. Especially IFN- γ levels are not a good single readout of cytotoxicity in the context of TGF-beta exposure (see comments above).

One concern regarding the medical impact is whether systemic application of EHMT2 inhibitors will be tolerated by patients as well as in mice. The potential side effects of systemic EHMT2 inhibition in the mouse model could be monitored more thoroughly than just the mortality rates of the treatment. For instance, changes in body weight or any impacts on routine blood/serum parameters that would suggest organ dysfunction? Any changes on the morphology of major organs?

The authors should at least explain why shRNAs were used for knockdowns instead of other techniques, namely Crispr.

TGF-beta itself is known to induce epigenetic changes in NK cells. The authors should at least discuss this topic.

Minor:

By reading the abstract is initially not clear whether inhibition of EHMT2 in tumor cells or NK cells is responsible for the phenotype: "We identified the histone methyltransferase EHMT2 as a key suppressor of NK cell-mediated cytotoxicity. EHMT2 inhibition enhanced NK cell-mediated elimination of diverse cancers, including uveal melanoma, breast cancer, and pancreatic cancer."

I suggest the following: "We identified the histone methyltransferase EHMT2 as a key suppressor of NK cell-mediated cytotoxicity. EHMT2 inhibition IN CANCER CELLS enhanced NK cell-mediated elimination of diverse cancers, including uveal melanoma, breast cancer, and pancreatic cancer."

Response to reviewers' comments

Reviewer's comments

Referee #1 (Remarks for Author):

This is a well-written manuscript with a cohesive narrative well-supported by the data. Rigor with regard to validating findings across multiple cell lines using multiple drugs and complementary genetic approaches is noted and appreciated.

We thank the reviewer for noting several strengths of our paper. The reviewer mentioned, "This is a well-written manuscript with a cohesive narrative well-supported by the data". The reviewer also highlighted the rigor of our approach, stating that our work demonstrates rigor "with regard to validating findings across multiple cell lines using multiple drugs and complementary genetic approaches".

The reviewer also made several important suggestions to further strengthen our manuscript. We thank the reviewer for his/her time in reviewing our manuscript and providing valuable recommendations to improve it further. Our detailed point-by-point responses to the reviewer's suggestions are provided below:

1. Authors should move the last paragraph of the Introduction to the Discussion.

As recommended by the reviewer, we have now moved the last paragraph of the Introduction to the Discussion section to reduce redundancy.

2. Authors should define the impact of EHMT2 inhibition on tumor cell-intrinsic growth properties, so the relative contribution of cell intrinsic versus NK cell-mediated killing can be fully understood.

This is an important point. To address this more thoroughly, we have included several new results in the revised manuscript. First, we have performed clonogenic assays by treating various cell lines used in our study with EHMT2 inhibitors (BRD4770, A366 and UNC0642) as well as using *EHMT2* knockdown and knockout cells. We found no significant impact of pharmacological or genetic inhibition of EHMT2 on the growth of cell lines. These results demonstrate that EHMT2 likely plays a limited cell-intrinsic role in promoting the growth of cell models that we have tested in our study.

These new results are shown below and included in **Fig. EV16** of the revised manuscript:

Figure EV16. Clonogenic assays for measuring the impact of pharmacological and genetic inhibition of EHMT2 on the indicated cancer cell lines. **A.** Indicated human cancer cell lines

were treated with DMSO or EHMT2 inhibitors UNC0642 (1 μ M) and A366 (1 μ M) and were analyzed by clonogenic assay. Representative images of clonogenic assays for the indicated human cancer cell lines under the indicated conditions are shown. **B.** Indicated uveal melanoma cell lines expressing non-specific (NS) shRNA or *EHMT2* shRNAs, or non-specific (NS) sgRNAs or *EHMT2*-targeting sgRNAs were analyzed by clonogenic assay. Representative images of clonogenic assays for the indicated uveal melanoma cell lines under the indicated conditions are shown. **C.** Indicated mouse cancer cell lines were treated with DMSO or indicated EHMT2 inhibitors BRD4770 (1 μ M), UNC0642 (1 μ M) and A366 (1 μ M) or expressing NS shRNA or *Ehmt2* shRNAs and were analyzed by clonogenic assay. Representative images of clonogenic assays for the indicated mouse cancer cell lines under the indicated conditions are shown.

In addition to this, we would like to highlight that in our mouse studies using Panc02 cells, no significant tumor reduction was observed following *Ehmt2* knockdown in immunodeficient NSG mice (**Fig. 6M and 6N**). In contrast, in immunocompetent C57BL/6 mice, Panc02 tumors with *Ehmt2* knockdown grew significantly more slowly compared to tumors expressing a non-specific (NS) control shRNA (**Fig. 6J and 6K**). These results further support a predominantly cell-extrinsic, tumor-promoting role for *Ehmt2* in this model. This conclusion is further supported by our NK cell depletion experiments, in which depletion of NK cells restored the growth of tumors otherwise suppressed by *Ehmt2* knockdown (**Fig. 7**), underscoring the role of NK cells in mediating this effect.

Finally, based on a question from the Reviewer 2 related to this, we also performed an additional experiment using the EMT6 cell line, in which we found some reduction in tumor volumes even in NSG mice, indicating that *in vivo* *Ehmt2* knockdown might exert some minor cell-intrinsic effects in EMT6 model. To establish the importance of NK cells in this model, we first performed an NK cell depletion experiment and found that the depletion of NK cells restored the *Ehmt2*-loss-induced tumor growth inhibition of EMT6 tumors, indicating an important role of NK cells in mediating *Ehmt2*-loss-driven tumor suppression. These new results are shown below and in the **Fig. EV13** of the revised manuscript.

Figure EV13. NK cells are necessary for *Ehmt2* knockdown-mediated tumor suppression in immunocompetent syngeneic mouse model of EMT6. **A.** Firefly luciferase (*F-Luc*)-labeled EMT6 cells expressing either Non-silencing (NS) shRNA or *Ehmt2* shRNA were injected orthotopically into the mammary fat pad of female BALB/c mice ($n = 5$) and treated with either IgG isotype control antibody ($100 \mu\text{g}/\text{mouse}$) or anti-Asialo-GM1 antibody ($100 \mu\text{g}/\text{mouse}$) one day before EMT6 cell line injections and then once a week during the course of the experiment. Bioluminescence images of mice at the indicated weeks after injection are shown. **B.** Bioluminescence intensities of the mice at the indicated weeks under the indicated conditions for the experiment presented in panel A are plotted. **C.** Measurement of tumor-infiltrated NK cells (%) in EMT6 tumors ($n=5$) expressing either NS shRNA or *Ehmt2* shRNA from BALB/c mice

treated with IgG isotype control or anti-Asialo-GM1 antibodies using FACS analysis. **D.** Measurement of relative NK cells (%) in spleens from mice (n = 5) harboring EMT6 tumors expressing either NS shRNA or *Ehmt2* shRNA from mice treated with IgG isotype control or anti-AsialoGM1 antibodies using FACS analysis. All quantitative data are shown as the mean ± SEM, ns = not significant, *p<0.05, ***p<0.001, ****p<0.0001.

3. Western blots are sub-publication quality. Messy blots need to be re-run to generate improved images.

We thank the reviewer for pointing out concerns regarding the quality of some Western blots. In response, we have repeated and replaced 32 Western blots to provide higher-quality images in the revised manuscript.

These new Western blots include the following:

1. **Fig.1B:**

MP41 blots: H3K9me2, Histone H3 and ACTINB treated with A366 and UNC0642

MP46 blots: H3K9me2, Histone H3 and ACTINB treated with A366 and UNC0642

2. **Fig. 1D:** H3K9me2 and Histone H3 in Mel92.1 cells expressing NS shRNA or *EHMT2* shRNA.

3. **Fig. 2E:** MP41 blots: AZGP1 and ACTINB in MP41cells treated with DMSO, A366 and UNC0642

4. **Fig. 2I:** AZGP1 and ACTINB blots with AZGP1 antibody in Vector- and AZGP1-V5 ORF expressing Mel92.1 and MP41cells.

5. **Fig. 3C:** TGF-β1 and ACTINB in Vector- and AZGP1-V5 ORF expressing Mel92.1 and MP41cells.

6. **Fig. EV2B:** H3K9me2 and Histone H3 in Mel92.1 and MP41 cells expressing NS shRNA or *EHMT1* shRNA.

7. **Fig. EV4D:** AZGP1 and ACTINB in Mel92.1 cells expressing NS shRNA or *AZGP1* shRNAs.

8. **Fig. EV11B:** *Ehmt2*, H3K9me2, Histone H3 and ACTINB in EMT6 cells expressing NS shRNA or *Ehmt2* shRNA

9. **Fig. EV12B:** *Ehmt2*, H3K9me2, Histone H3 and ACTINB in Panc02 cells expressing NS shRNA or *Ehmt2* shRNA

4. Phenotype of NK cell-mediated cytotoxicity reported in Figure 1 is much stronger in experiments using pharmacological agents as compared to shRNA. Presumably this difference is due to incomplete inactivation with shRNAs; CRISPR-mediated KO could provide an alternative approach that may generate a stronger phenotype. This would strengthen the argument about *EHMT2*'s importance in this context.

The reviewer has raised a valid and important point. In response to this recommendation, we have now generated *EHMT2* knockout (KO) uveal melanoma cell lines. Using these cancer cells, we demonstrated that *EHMT2* knockout is indeed substantially more effective in inducing NK cell-mediated cytotoxicity against these cells compared to shRNAs. These new findings are now shown below and have been included in **Fig. EV4A and EV4B** of the revised manuscript.

5. In Figures 3 and 4, the manuscript could be strengthened by proving that the effect of TGF β 1 in the context of AZGP1 depletion, which according to their model is required to unleash elevated TGF β 1.

As the reviewer noted that, we documented that the inhibition of EHMT2 results in increased expression of AZGP1 that in turn inhibits TGF- β 1 levels. In previous version of the manuscript, we demonstrated that ectopic expression of AZGP1 in Mel92.1 and MP41 cells significantly downregulated the TGF- β 1 levels (**Fig. 3C**).

Based on the new recommendation from the reviewer, we knocked down both *AZGP1* and *TGF- β 1* in uveal melanoma cells and treated them with EHMT2 inhibitor UNC0642. We found, as expected, *AZGP1* knockdown inhibited the NK cell-mediated cytotoxicity following EHMT2 inhibitor UNC0642 treatment. However, this reduction was reversed in the double knockdown of *AZGP1* and *TGF- β 1*, indicating that the reduced NK cell-mediated cytotoxicity observed with *AZGP1* knockdown is dependent on the upregulation of TGF- β 1. These new results are shown below and in the **Fig. EV4E and EV4F** of the revised manuscript.

Figure EV4E and 4F. Double knockdown of both AZGP1 and TGF- β 1 enhances NK cell-mediated cytotoxicity in cancer cells. E. Mel92.1 cells expressing non-specific (NS) shRNA, *AZGP1* shRNA, *TGF- β 1* shRNA or both *AZGP1* shRNA and *TGF- β 1* shRNA were analyzed by immunoblotting for indicated proteins. ACTINB was used as a loading control. F. Mel92.1 cells expressing either NS shRNA, *AZGP1* shRNA, *TGF- β 1* shRNA or both *AZGP1* shRNA and *TGF- β 1* shRNA were treated with DMSO or EHMT2 inhibitor UNC0642 (1 μ M) for 48 hrs and analyzed for NK cell-mediated cytotoxicity using an LDH-based cytotoxicity assay. Relative NK cell-mediated cytotoxicity under the indicated conditions is plotted. All quantitative data are presented as the mean \pm SEM; ** p < 0.01, *** p < 0.001, **** p < 0.0001.

6. Figure 6 would be strengthened by quantitation of tumor growth rates in tumors from the mouse models (e.g., Ki67, BrdU, etc.).

To address this point, we have incorporated Ki-67 staining into the experiments corresponding to **Fig. 6E and 6J**. The resulting data are now presented in **Fig. EV11E, EV11F and Fig. EV12E, EV12F** of the revised manuscript.

Figure. EV11E, EV11F and Fig. EV12E, EV12F *Ehmt2* knockdown tumors display reduced Ki67 staining. **Fig. EV11E.** Ki-67 expression was analyzed by immunohistochemistry (IHC) in EMT6 tumor sections expressing NS shRNA and *Ehmt2* shRNA. Representative Ki-67 staining images of EMT6 tumor sections expressing NS shRNA and *Ehmt2* shRNA at 20× magnifications are shown. Scale bar, 50 μm. **Fig. EV11F.** Quantitation of Ki-67 staining for the experiment presented in panel EV11E. **Fig. EV12E.** Ki-67 expression was analyzed by immunohistochemistry (IHC) in Panc02 tumor sections expressing NS shRNA and *Ehmt2* shRNA. Representative Ki-67 staining images of Panc02 tumor sections expressing NS shRNA and *Ehmt2* shRNA at 20× magnifications are shown. Scale bar, 50 μm. **Fig. EV12F.** Quantitation of Ki-67 staining for the experiment presented in panel EV12E. All quantitative data are presented as the mean ± SEM; **p < 0.01.

7. Authors could shorten the summary of their findings in the Discussion, as this is redundant.

We agree with the reviewer's observation, and, in line with a similar comment (Comment 1), we have moved the final paragraph of the Introduction to the Discussion section and have also shortened the summary of our findings accordingly.

Referee #2 (Comments on Novelty/Model System for Author):

Technical quality: Some of the assays need better comparison side by side and not in separate experiments. Also, the NK cell cytotoxicity should be assessed in different ratios to be convincing. Medical impact would be improved by using the same inhibitors in vitro and in vivo.

We are grateful to the reviewer for the thoughtful comments and for the time and effort dedicated to evaluating our manuscript. Below, we provide our detailed point-by-point responses to the general note above, followed by detailed responses to the reviewer's specific comments.

Referee #2 (Remarks for Author):

The authors analyzed the impact of EHMT2 inhibition or knockdown in different cancer cell lines on the susceptibility of these cells to NK cell-mediated lysis. By employing different techniques—RNA sequencing, in vitro, and in vivo approaches—they could show an important contribution of EHMT2 to escape of tumor cells via induction of TGF β 1. While overall it is an interesting and complex manuscript, some experiments lack direct connection, and some results are not reproduced in the same cell lines. Several issues need to be addressed:

The reviewer noted strengths in our manuscript and stated that “By employing different techniques —RNA sequencing, in vitro, and in vivo approaches—they could show an important contribution of EHMT2 to escape of tumor cells via induction of TGF β 1.” We thank the reviewer for highlighting the strengths of our manuscript.

However, the reviewer also noted that “Several issues need to be addressed” and provided multiple excellent suggestions for further strengthening our manuscript.

We thank the reviewer for his/her valuable suggestions. Our detailed point-by-point responses to the reviewer's specific questions are presented below:

- In the introduction (last paragraph before summary), it is unclear whether EHMT2 has these effects in immune cells or in tumor cells. Please specify.

We thank the reviewer for bringing this ambiguity to our attention. We have revised this paragraph to clarify this important point and indicated whether EHMT2 influenced either cancer cell or immune cell functions.

- In the summary at the end of introduction: In this study... - the authors need to specify from the beginning whether they focus on EHMT2 in tumor cells or NK cells. Loss of.. needs to be mentioned where the loss was.

We apologize for the unclear nature of this statement. We have now clarified that the loss was in cancer cells in the revised manuscript.

Results:

- General comment: The LDH based assay to assess NK cell cytotoxicity might be affected by differences in NK cell-derived LDH. They might show different levels dependent whether they are seeded alone or killing the tumor cells. The authors should include different ratios of NK to tumor cells for most crucial experiments to confirm the specificity of their signal.

This is an important suggestion. We have added an explanation below, along with several new experiments to address this point in the revised manuscript, as described below:

We appreciate the reviewer's comment regarding the possible effect of NK cell-derived LDH on the cytotoxicity assay. However, we have carefully designed our experiments by including appropriate controls and subtracting the background readings before calculating the cytotoxicity. We used the following controls for all of our NK cell cytotoxicity experiments: **1.** NK cells alone. **2.** Cancer cells alone **3.** NK cell medium alone **4.** Cancer cell medium alone. **5.** NK cell medium+cancer cell medium. As shown in the formula below that we used to calculate NK cell-mediated cytotoxicity, LDH experimental value is subtracted from both LDH effector cells (NK cells alone) and LDH spontaneous (Cancer cells alone).

$$\frac{\text{LDH experimental} - \text{LDH effector cells} - \text{LDH spontaneous} \times 100}{\text{LDH maximal}}$$

Additionally, guided by the suggestion from the reviewer, we have now included new experiments that were performed using different cancer cell (target cells): NK cell (effector cells) ratios. For these studies, we either treated the uveal melanoma cell lines Mel92.1 and MP41 cells with the EHMT2 inhibitors (UNC0642 or A366) or used Mel92.1 and MP41 cells expressing *EHMT2* shRNAs compared to cells expressing non-specific (NS) shRNAs, at various cancer cell:NK cell ratios (1:5, 1:10, 1:20 and 1:40). As the reviewer will see, even with variations in cancer cell:NK cell ratios, pharmacological or genetic inhibition of EHMT2 resulted in significantly increased NK cell-mediated cytotoxicity. These new results are shown below and have been included in the **Fig. EV3B and EV3C** of the revised manuscript.

Figure EV3B and EV3C. Pharmacological and genetic inhibition of EHMT2 results in increased NK cell-mediated cytotoxicity, independent of NK cell (effector cell) and cancer cell (target cell) ratios. B. Mel92.1 and MP41 cells were analyzed for NK cell-mediated cytotoxicity using LDH-based cytotoxicity assay at indicated cancer cell: NK cell ratios (1:5, 1:10, 1:20 or 1:40) after treatment with DMSO or indicated small molecule inhibitors of EHMT2 for 48 hr. NK cell-mediated cytotoxicity under indicated conditions is plotted relative to DMSO treatment at 1:5 cancer cell: NK cell ratio. **C.** Mel92.1 and MP41 cells expressing either NS shRNA or *EHMT2* shRNA were analyzed for NK cell-mediated cytotoxicity using LDH-based cytotoxicity assay at indicated cancer cell: NK cell ratios (1:5, 1:10, 1:20 or 1:40). NK cell-mediated cytotoxicity for under indicated conditions is plotted relative to DMSO treatment at 1:5 cancer cell: NK cell ratio. All quantitative data are presented as the mean \pm SEM; * $p < 0.05$, ** $p < 0.01$, *** $p < 0.001$, **** $p < 0.0001$.

Moreover, to provide further support for our finding, we have also performed an additional assay to measure NK cell-mediated cytotoxicity. For this purpose, we used a fluorescence-based quantitative Calcein-AM assay to monitor NK cell-mediated cytotoxicity (1, 2). Consistent with the results of our LDH-based assay, the quantitative Calcein-AM-based assay also showed increased NK cell-mediated cytotoxicity against cancer cells following genetic or pharmacological inhibition of EHMT2. These new results are shown below and in the **Fig. EV3D, E** of the revised manuscript.

Figure EV3D and EV3E. Calcein-AM-based NK cell-mediated cytotoxicity assay shows that pharmacological or genetic inhibition of EHMT2 enhances cancer cell eradication by NK cells. **D.** Mel92.1 and MP41 cells were analyzed for NK cell-mediated cytotoxicity using calcein-AM-based quantitative NK cell-mediated cytotoxicity assay after treatment with DMSO or indicated inhibitors of EHMT2 for 48 hr. Relative NK cell-mediated cytotoxicity under indicated conditions is plotted. **E.** Mel92.1 and MP41 cells expressing either NS shRNA or *EHMT2* shRNA were analyzed for NK cell-mediated cytotoxicity using calcein-AM-based quantitative NK cell-mediated cytotoxicity assay. Relative NK cell-mediated cytotoxicity for under indicated conditions is plotted. All quantitative data are presented as the mean \pm SEM; **** $p < 0.0001$.

- Figure 2I– please include AZGP1 antibody, not just V5.

As recommended by the reviewer, we have now included the AZGP1 Western blot in the Figure 2I of the revised manuscript. The revised Fig. 2I is shown below.

Figure 2I. Western blot analysis for indicated proteins. Mel92.1 and MP41 cells expressing AZGP1-V5 ORF were analyzed for the expression of indicated proteins by immunoblotting. ACTINB was used as a loading control.

- **Figure 3L**– This is well known that TGFb generally suppresses NK cell function. The better experiment would be to neutralize TGFb in the same experiment next to EZHT2 KD or inhibition - does it reach same level? Is there still room for synergy?

We thank the reviewer for this important suggestion. We agree that the simultaneous suppression of TGF- β 1 and EHMT2 may produce a synergistic effect on NK cell-mediated cytotoxicity. To investigate this, we neutralized TGF- β 1 in the conditioned media obtained from cancer cells treated with the EHMT2 inhibitor UNC0642. NK cells were then exposed to this media, and cytotoxicity was assessed. As anticipated by the reviewer, we observed a significantly enhanced NK cell-mediated cytotoxic response when both EHMT2 was inhibited and TGF- β 1 was neutralized, compared to either treatment alone. These results are now presented below and included in **Fig. EV5A and 5B** of the revised manuscript.

Figure EV5A and EV5B. Simultaneous inhibition of EHMT2 and TGF- β 1 neutralization augments NK cell-mediated cytotoxicity. **A.** Mel92.1 cells were treated with the EHMT2 inhibitor UNC0642 (1 μ M) or DMSO for 72 hr in Opti-MEM. Following this, conditioned media (CM) was then collected, concentrated, and used to treat NK cells for 24 hr in the presence of either control IgG or a TGF- β 1-neutralizing antibody (1D11). Then, NK cells were assessed for cytotoxic activity against Mel92.1 cells using an LDH-based cytotoxicity assay. The relative NK cell-mediated cytotoxicity under the indicated conditions is shown. **B.** MP41 cells were treated with the EHMT2 inhibitor UNC0642 (1 μ M) or DMSO for 72 hr in Opti-MEM. Following this, conditioned media (CM) was then collected, concentrated, and used to treat NK cells for 24 hr in the presence of either control IgG or a TGF- β 1-neutralizing antibody (1D11). Then, NK cells were assessed for cytotoxic activity against MP41 cells using an LDH-based cytotoxicity assay. The relative NK cell-mediated cytotoxicity under the indicated conditions is shown. All quantitative data are presented as mean \pm SEM; **p < 0.01, ***p < 0.001, ****p < 0.0001.

- **Figure 4**– The transition here lack red line. It needs to be better explained why authors now look at the ligands on target cells while before claimed they treat NK cells with conditioned medium?

We apologize to the reviewer for the ambiguity. As the reviewer may be well aware that

NKG2D is an activating receptor on NK cells and plays crucial role in immune surveillance against cancer cells. Numerous studies have documented the expression of NKG2D ligands in different cancer types to be important in NK cell-mediated tumor eradication (3-8). Furthermore, it has been established that the quantitative level of NKG2D ligand expression defines the “threat” level of target cell, and it correlates with NK cell-dependent tumor elimination (9, 10).

Thus, the previously documented role of NKG2D ligands in NK cell-mediated cytotoxicity and our own NK cell-mediated cytotoxicity assays, which showed that pharmacological or genetic inhibition of EHMT2 or knockdown of TGF- β 1 enhances NK cell-mediated killing of uveal melanoma cells and other cancer cell types (**Fig. 1C, 1E, 1G, 3E, 6A and 6C**). Further, considering previous studies that established the role of TGF- β 1 in downregulating NKG2D ligands (11) led us to investigate the NKG2D ligands expression.

Conditioned medium experiments (**Fig. 3J-L**) were performed to investigate the impact of tumor-derived TGF- β 1 on NK cell function and cytotoxicity. For this, we also showed in our manuscript TGF- β 1 regulates IFN- γ , T-bet, Eomes, NCR3, NKG2D, CD49a CD103, CD56 and CXCR6 expression in NK92MI cells (**Fig. 4I-K and Fig. EV9**), that plays crucial role in NK cell-mediated cytotoxicity.

Based on this comment from the review, we have revised our manuscript to better highlight our rationale.

- **Figure 4A-C**—Levels of NKG2D ligands need to be shown on the surface using flow cytometry to make sure that they would have a functional consequence.

We appreciate the reviewer’s suggestion and concur that evaluating the surface expression of NKG2D ligands would strengthen the study. Accordingly, we conducted flow cytometry (FACS) analyses to quantify MICB and ULBP3 surface levels corresponding to the experiments shown in Figures 4A–4C. The resulting data have been included in **Fig. EV6** and **EV7** of the revised manuscript and are presented below.

A**B**
Figure EV6. Monitoring the cell surface expression of NKG2D ligands ULBP3 and MICB on cancer cells under EHMT2 inhibition and *TGF- β 1* knockdown using flow cytometry. **A.** Mel92.1 and MP41 cells were treated with DMSO, UNC0642 (3 μ M) or A366 (3 μ M) for 72 hrs

and were analyzed for the cell surface expression of ULBP3 and MICB using flow cytometry. The representative histograms under the indicated conditions are shown. **B.** Mel92.1 and MP41 cells expressing non-specific (NS) shRNA or *TGF-β1* shRNA were analyzed for the cell surface expression of ULBP3 and MICB using flow cytometry. The representative histograms under the indicated conditions are shown.

Figure EV7. Monitoring the cell surface expression of NKG2D ligands ULBP3 and MICB under EHMT2 inhibition and *TGF-β1* overexpression using flow cytometry. **C.** Mel92.1 and MP41 cells expressing either an empty vector or V5-tagged *TGF-β1* ORF were treated with DMSO or UNC0642 (3 μM) for 72 hrs and analyzed for the cell surface expression of ULBP3 and MICB using flow cytometry. The representative histograms under the indicated conditions are shown.

- **Figure 4E**– The knockdown of MICB does not have any effect on NK susceptibility in cells not treated with the inhibitor? Could the authors comment on that? Or is the quantification misleading.

The reviewer makes a good point. As the reviewer noted, MICB is expressed at a lower level in the presence of EHMT2. Thus, this may, in part, explain why MICB knockdown without treatment with a EHMT2 inhibitor (which restores MICB expression) does not have a significant impact on NK cell susceptibility of cancer cells when MICB is knocked down under basal (e.g., when cancer cell are not treated with an EHMT2 inhibitor).

- **Figures 4F–K**– In this context it would mainly make sense to look at NKG2D. If the authors claim that NKG2DL are crucial in their system. Why NCR3 was chosen and not other receptor.

The reviewer raises this important point. NCR3 was selected based on several previously published studies demonstrating a critical role for TGF- β 1 in the transcriptional downregulation of NCR3, which encodes the NKp30 protein. This downregulation has been shown to significantly impair NK cell-mediated cytotoxicity (see references (12-14)). These studies highlight a mechanism by which TGF- β 1-producing cells can evade immune surveillance by suppressing NKp30 expression on NK cells. We have incorporated this rationale into the revised manuscript to clarify our choice of NCR3 as a gene of interest.

In addition, we fully agree with the reviewer that it would be logical and valuable to assess the expression of NKG2D, another key activating receptor on NK cells. In response to this thoughtful suggestion, we analyzed NKG2D expression following TGF- β 1 treatment. We found that TGF- β 1 treatment significantly reduced both mRNA and protein levels of NKG2D in NK92MI cells. These new data further support the immunosuppressive role of TGF- β 1 in downregulating activating NK cell receptors. These results are shown below and in **Fig. 4J** and **EV9B** of the revised manuscript.

Figure 4J and EV9B. TGF- β 1 suppresses the expression of NKG2D on NK cells. 4J. NK92MI cells were treated with TGF- β 1 (10 ng/ml) for 24 hr and analyzed for NKG2D protein using immunoblotting. ACTINB was used as a loading control. **EV9B.** NK92MI cells were treated with TGF- β 1 (10 ng/ml) for 24 hr and analyzed for *NKG2D* mRNA levels using RT-qPCR. Relative mRNA levels are presented under the indicated conditions. *ACTINB* was used for normalization. All quantitative data are presented as the mean \pm SEM; **p< 0.01.

- **Figure 5**– It needs to be better explained why the chemokines were analyzed as in these experiments the migration does not seem relevant. The enhanced percentage of NK cells in tumor in vivo (Fig. 6) may be a consequence of smaller tumors and thereby a higher immune cell ratio.

This is an important point. We undertook chemokine analysis because NK cell infiltration into the tumor microenvironment is one of the key determinants of NK cell-mediated tumor eradication. Thus, examining whether EHMT2 loss influences NK cell migration represents a relevant line of investigation.

The reviewer may have also noticed that our assertion that EHMT2 loss might also promote NK cell migration is not solely based on NK cell-infiltration measured in tumors themselves but was also validated in cell culture studies (shown in **Fig. 5**), in which we showed that EHMT2 loss or

TGF- β 1 inhibition resulted in increased expression of chemokine CCL27 and CXCL10. Both CCL27 and CXCL10 are known to increase the NK cell migration (15-18).

We also showed the direct impact of EHMT2 inhibition and TGF- β 1 inhibition on their ability to promote NK cell migration (see **Fig. 5H and 5I**).

However, we recognize the concern of the reviewer. Therefore, to determine whether the difference observed in NK cell infiltration was due to tumor volume differences or was a result of *Ehmt2* loss, we performed an experiment using Panc02 cells. In this new experiment, we analyzed the Panc02 tumors expressing either *Ehmt2* shRNA or non-specific (NS) shRNA at an early stage of tumor progression, when the differences in tumor volumes were not significant.

We found that, even at early stages, before the tumor volume differences started to become significant, a significantly larger number of NK cells infiltrated *Ehmt2* shRNA-expressing Panc02 tumors compared to control NS shRNA-expressing Panc02 tumors. These results further support the notion that NK cell infiltration is dependent on loss of *Ehmt2* expression rather than solely driven by differences in tumor volume. These new results are shown below and in the **Fig. EV12H and 12I** of this revised manuscript.

Figure EV12H and EV12I. *Ehmt2* knockdown promotes NK cell infiltration in Panc02 tumors in mice. **H.** Panc02 cells expressing either NS shRNA or *Ehmt2* shRNA were injected subcutaneously into C57BL/6 mice (n = 5). Tumor volumes at the indicated times are plotted. **I.** Measurement of tumor-infiltrated NK cells (%) in the Panc02 tumors (n=5) under indicated conditions using FACS analysis. All quantitative data are shown as the mean \pm SEM, ns = not significant and ****p<0.0001.

Collectively, these new results demonstrate the roles of EHMT2 and TGF- β 1 in inhibiting NK cell migration.

- The sentence needs to be more precise: We found that TNBC cells treated with EHMT2 inhibitors (A366 and UNC0642) showed increased NK cell-mediated cytotoxicity is not phrase correctly. Rather show increased susceptibility/sensitivity? Please correct through the manuscript.

As per the recommendation of the reviewer, we have revised this sentence to “We found that TNBC cells treated with EHMT2 inhibitors (A366 and UNC0642) showed increased sensitivity

to NK cell-mediated cytotoxicity.”

- **In vivo:** Why UM cell lines were not used in the mouse model?

A reliable and widely accepted syngeneic mouse model representative of uveal melanoma (UM) is not available, which is why we could not use the UM cell lines in mice for our in vivo studies. We have explained this limitation in the revised version of the manuscript.

- **Figures 6H and 6I**– There is a clear significant difference between the tumor growth in NSG mice. To make a clear statement that the effect observed is not tumor intrinsic one would need to run this experiment side by side or even more clear perform the experiment in immunodeficient mice with or without adoptive NK cell transfer to check whether there is really an additive effect of NK cells and knockdown.

This is an important experiment. To address this, we performed two new sets of experiments using EMT6 cell lines, in which we found a reduction in tumor volume even in NSG mice, indicating that, in this model Ehmt2 loss might suppress tumor growth.

To establish the importance of NK cells in this model, we first performed an NK cell depletion experiment and found that the depletion of NK cells restored the Ehmt2-loss-induced tumor growth inhibition of EMT6 tumors, indicating an important role of NK cells in mediating Ehmt2-loss-driven tumor suppression. These new results are shown below and in the **Fig. EV13** of the revised manuscript.

Figure EV13. NK cells are necessary for *Ehmt2* knockdown-mediated tumor suppression in immunocompetent syngeneic mouse model of EMT6. **A.** Firefly luciferase (F-Luc)-labeled EMT6 cells expressing either Non-silencing (NS) shRNA or *Ehmt2* shRNA were injected orthotopically into the mammary fat pad of female BALB/c mice ($n = 5$) and treated with either IgG isotype control antibody (100 $\mu\text{g}/\text{mouse}$) or anti-AsialoGM1 antibody (100 $\mu\text{g}/\text{mouse}$) one day before EMT6 cell line injections and then once a week during the course of the experiment. Bioluminescence images of mice at the indicated weeks after injection are shown. **B.**

Bioluminescence intensities of the mice at the indicated weeks under the indicated conditions for the experiment presented in panel A are plotted. **C.** Measurement of tumor-infiltrated NK cells (%) in EMT6 tumors (n=5) expressing either NS shRNA or *Ehmt2* shRNA from BALB/c mice treated with IgG isotype control or anti-AsialoGM1 antibodies using FACS analysis. **D.** Measurement of relative NK cells (%) in spleens from mice (n = 5) harboring EMT6 tumors expressing either NS shRNA or *Ehmt2* shRNA from mice treated with IgG isotype control or anti-AsialoGM1 antibodies using FACS analysis. All quantitative data are shown as the mean \pm SEM, ns = not significant, *p<0.05, ***p<0.001, ****p<0.0001.

Next, we performed an adoptive NK cell transfer experiment, in which we adoptively transferred mouse NK cells into NSG mice, then injected EMT6-*F-Luc* cells orthotopically injected into the mammary fat pad of female NSG mice and compared their tumor growth with or without adoptive NK cell transfer.

The schematic for the experiment is shown in the figure below, and details are as follows: Adoptive NK cell transfer was done as described previously, with minor modifications (19). Spleens were dissected from female BALB/c mice, and a single-cell suspension was prepared using the Spleen Dissociation Kit (Miltenyi Biotec, Cat#130-095-926). NK cells were isolated using a Mouse NK Cell Isolation Kit (Miltenyi Biotec, Cat# 130-115-818) following the manufacturer's protocol. Before adoptive transfer, the NK cells were cultured at a density of 5×10^6 cells per ml with RPMI 1640 medium containing 10% FBS, 2 mM L-glutamine, 1% penicillin–streptomycin, 50 μ M 2-mercaptoethanol (Sigma-Aldrich), 50 ng/ml IL-2 (BioLegend), 50 ng/ml IL-15 (BioLegend), and 50 ng/ml IL-18 (BioLegend). The NK cell purity was analyzed by flow cytometry. The purity of NK cells (CD49b⁺ NKp46⁺) prior to adoptive transfer was greater than 90%. Before adoptive transfer in mice, the NK cells were washed twice with $1 \times$ PBS, and 1×10^6 NK cells were injected intravenously into NSG mice one day before the injection of EMT6-F-Luc cells, and then once every week (a total of 3 additional times) during the course of the experiment.

However, we found that EMT6 cells grew too aggressively in NSG mice, and NK cells were insufficient to control the growth of these cells in NSG (immunodeficient) mice. Thus, based on our experiments, we found that the EMT6 model in the NSG mice background may not be suitable for the experiment recommended by the reviewer. The result of this experiment is shown below:

However, we believe that the above results with the EMT6 model with and without NK cell depletion in BALB/c mice (shown in **Fig. EV13**) along with our studies in the Panc02 model with and without NK cell depletion in C57BL/6 and *Rag2* KO mice (**Fig. 7**) provide ample evidence that NK cell-mediated tumor control is key to *Ehmt2*-loss-driven tumor suppression.

Figure legend. Monitoring the effect of Adoptive NK cell transfer in NSG mice on EMT6 tumor growth. **A.** Schematic representation of NK cells isolation from spleens harvested from BALB/c mice. **B.** Purity of NK cells analyzed by FACS using NK cell specific markers CD49b and NKp46 antibodies. **C.** Schematic representation of adoptive NK cell transfer in NSG mice. **D.** EMT6-F-Luc cells expressing were adoptively transferred with NK cells and EMT6-F-Luc cells were injected orthotopically into the mammary fat pad of female NSG mice (n = 5) as indicated in the schematic in panel A. Bioluminescence images of mice at the indicated weeks after cells injection are shown. **E.** Bioluminescence intensities of the mice at the indicated weeks from panel D is plotted. All quantitative data are shown as the mean ± SEM, ns = not significant p-value.

- **In vivo**– It is unclear why the authors used a different inhibitor for in vivo treatment without including it in any in vitro assays. Please repeat key findings of your paper with the new inhibitor used in vivo.

As indicated in the manuscript, we used BRD4770 due to its more favorable toxicity profile in mouse models. In relation to this, please see additional new results below, including toxicity profile of BRD4770 and new data demonstrating that BRD4770 functions similarly to the EHMT2 inhibitors. These new results are shown below and in the **Fig. EV15** of the revised manuscript.

Figure EV15. Analysis of BRD4770 induced toxicity in mice. **A.** C57BL/6, BALB/c and

NSG mice were treated with vehicle or BRD4770 (15 mg/kg body weight) for 7 weeks. Body weights were measured at the indicated weeks. Average body weights (n=6) are plotted at the indicated time points. **B.** C57BL/6, BALB/c and NSG mice were treated with vehicle or BRD4770 (15 mg/kg body weight). Serum ALT levels were measured at the end of the experiment and is plotted under the indicated conditions. **C.** C57BL/6, BALB/c and NSG mice were treated with vehicle or BRD4770 (15 mg/kg body weight). Serum AST levels were measured at the end of the experiment and is plotted under the indicated conditions. **D.** C57BL/6, BALB/c and NSG mice were treated with vehicle or BRD4770 (15 mg/kg body weight). Serum creatinine levels were measured at the end of the experiment and is plotted under the indicated conditions. **E.** Histopathological examination of vital organs (heart, lung, liver, and kidney) from C57BL/6 mice treated with vehicle or BRD4770 (15 mg/kg body weight) as shown by H&E staining images (Scale bar, 4×: 250 μ m, 20×: 50 μ m). **F.** Histopathological examination of vital organs (heart, lung, liver, and kidney) from BALB/c mice treated with vehicle or BRD4770 (15 mg/kg body weight) shown by H&E staining images (Scale bar, 4×: 250 μ m, 20×: 50 μ m). **G.** Histopathological examination of vital organs (heart, lung, liver, and kidney) from NSG mice treated with vehicle or BRD4770 (15 mg/kg body weight) shown by H&E staining images (Scale bar, 4×: 250 μ m, 20×: 50 μ m). All quantitative data are presented as the mean \pm SEM; ns, not significant.

Furthermore, as recommended by the reviewer, we have repeated the key findings using the new inhibitor BRD4770. We found that, similar to other EHMT2 inhibitors (UNC0642 and A366), BRD4770 enhances NK cell-mediated cancer cell eradication, upregulates AZGP1 expression and suppresses TGF- β 1 expression. Moreover, treatment with BRD4770 induced the expression of NKG2D ligand MICB and the chemokines CXCL10 and CCL27 in cancer cells. These new results are shown below and in the **Fig. EV14** of the revised manuscript.

Figure EV14. The EHMT2 inhibitor BRD4770 enhances NK cell-mediated cytotoxicity, upregulates AZGP1, suppresses TGF- β 1, and induces CXCL10 and CCL27 expression. **A.** Mel92.1 and MP41 cells were treated with DMSO or BRD4770 (5 μ M) for 48 hr and NK cell-mediated cytotoxicity was measured using an LDH-based cytotoxicity assay. Relative NK cell-mediated cytotoxicity under the indicated conditions for indicated UM cell lines is plotted. **B.** Mel92.1 and MP41 cells treated with DMSO or BRD4770 (5 μ M) for 48 hr and were analyzed for *AZGP1* mRNA using RT-qPCR. mRNA expression relative to DMSO treated cells is plotted. *ACTINB* was used for normalization. **C.** Mel92.1 and MP41 cells were treated with DMSO or BRD4770 (5 μ M) for 48 hr and were analyzed for *AZGP1* expression by immunoblotting. *ACTINB* was used as a loading control. **D.** Mel92.1 and MP41 cells were treated with DMSO or BRD4770 (5 μ M) for 48 hr and were analyzed for *TGF- β 1* mRNA using RT-qPCR. mRNA expression relative to DMSO treated cells is plotted. *ACTINB* was used for normalization. **E.** Mel92.1 and MP41 cells were treated with DMSO or BRD4770 (5 μ M) for 48 hr and were analyzed for TGF- β 1 expression by immunoblotting. *ACTINB* was used as a loading control. **F.** Mel92.1 and MP41 cells were treated with DMSO or BRD4770 (5 μ M) for 48 hr and were analyzed for *MICB* mRNA using RT-qPCR. mRNA expression relative to DMSO treated cells is plotted. *ACTINB* was used for normalization. **G.** Mel92.1 and MP41 cells were treated with DMSO or BRD4770 (5 μ M) for 48 hr and were analyzed for *MICB* expression by immunoblotting. *ACTINB* was used as a loading control. **H.** Mel92.1 and MP41 cells were treated with DMSO or BRD4770 (5 μ M) for 48 hr and were analyzed for *CXCL10* and *CCL27* mRNAs using RT-qPCR. mRNA expression relative to DMSO treated cells is plotted. *ACTINB* was used for normalization. All quantitative data are shown as the mean \pm SEM, * p <0.05, ** p <0.01, *** p <0.001, **** p <0.0001.

Referee #3 (Comments on Novelty/Model System for Author):

One concern regarding the medical impact is whether systemic application of EHMT2 inhibitors will be tolerated by patients as well as in mice.

This is an important question. We cannot comment on tolerability in human patients because it must be empirically tested in a Phase I clinical trial, which is beyond the scope of our current manuscript.

However, based on the comments from the reviewer, we have presented a large set of new data to show that BRD4770 is well tolerated in mice (see our detailed point-by-point responses to the relevant reviewer comments in this regard below).

Referee #3 (Remarks for Author):

The manuscript by Chava et al. reports that inhibition or loss of the histone methyltransferase EHMT2 in tumor cells enhances enhanced NK cell-mediated elimination of diverse cancers, including uveal melanoma, breast cancer, and pancreatic cancer. EHMT2 loss increased AZGP1 and decreased TGF- β 1 levels, resulting in autocrine elevation of NKG2D ligands MICB and ULBP3, chemokines in cancer cells, and paracrine stimulation of NK cell function. The manuscript is written in a comprehensive manner, the experimental setups appear are sound and the proposed mechanism is potentially of interest.

Given that the authors claim to report a, at least partially, TGF- β 1-driven phenomenon, it will be important to take in account the existing conceptual framework regarding the impact of TGF- β 1 on the phenotype and function of Type I Innate Lymphoid Cells (ILCs), including NK cells. In addition, in certain setting the readouts to determine NK cell cytotoxicity need some refinement. Therefore, the impact of the study is likely to increase by addressing the questions below experimentally.

We thank the reviewer for his/her positive comment and for noting that “the manuscript is written in a comprehensive manner, the experimental setups are sound and the proposed mechanism is potentially of interest.” The reviewer also made several excellent suggestions to further strengthen our manuscript. Our detailed, point-by-point responses to the reviewer’s questions are presented below:

Major:

It is important to take in account the potential impact on the NK cell and ILC1 phenotypes respectively (doi: 10.1038/ni.3800.). In the murine setting, paracrine TGF- β -signaling drives conversion of NK cells (CD49a-CD49b+Eomes+) into intermediate type 1 innate lymphoid cell (intILC1) (CD49a+CD49b+Eomes+) populations and ILC1 (CD49a+CD49b-Eomesint) populations in the tumor microenvironment. Importantly, intILC1s and ILC1s are unable to control local tumor growth, whereas NK cells maintain tumor immunosurveillance. ILCs 1 are T-Bet High, produce cytokines like TNF and GM-CSF, NK cells are Eomes high and cytotoxic, both subsets are capable of producing high amounts of IFN- γ . Therefore, these features should be included in the in vitro and in vivo analysis of the manuscript. Even more so, since human tumor ILC1-like cells (Lin- (CD19-CD14-CD3-)CD56+CXCR6+ ILC1-like cells) haven been described in human tumors as well.

Along these lines and concerning Figure 4: To sufficiently adress NK cell function in this setting

(exposure to TGF-beta), the authors should perform a cytotoxicity/killing assay in addition measuring IFN- γ levels and the expression of surface receptors in NK cells. Especially IFN- γ levels are not a good single readout of cytotoxicity in the context of TGF-beta exposure (see comments above).

We thank the reviewer for bringing this valid point to our attention. We agree with the reviewer that this is an important issue and appreciate the suggestion to investigate it further. As per the recommendation from the reviewer, we have now measured the human NK cell-specific markers of ILC1, including CD49a, CD103, CD69, CD56 and CXCR6 (20-22) in NK92MI cells following treatment with TGF- β 1.

We have now performed additional NK cell-mediated cytotoxicity assays after treating NK92MI cells with TGF- β 1. We found that treatment of NK cells (NK92MI) with TGF- β 1 resulted in reduced NK cell-mediated cytotoxicity against cancer cells (**Fig. 4F**). We have also measured the expression of NKG2D, a key activating receptor on NK cells, after TGF- β 1 treatment and found that NKG2D is downregulated following TGF- β 1 treatment in NK92MI cells (**Fig. 4J** and **Fig. EV9B**).

We also found increased expression of the CD49a and CD103, CD56 and CXCR6 while CD69 expression remained unchanged (**Fig. 4K** and **Fig. EV9C**).

Furthermore, we also analyzed the population of active NK cells (cytotoxic) and ILC1 (non-cytotoxic) in tumors derived from Panc02 and EMT6 cells expressing *Ehmt2* shRNA or a control non-specific (NS) shRNA. We found that ILC1 cells (Lin-CD49b-CD49a⁺) were significantly fewer in Panc02 and EMT6 tumors expressing *Ehmt2* shRNA compared to those expressing NS shRNA. Conversely, active NK cells (Lin-CD49b⁺CD49a-) were significantly higher in Panc02 and EMT6 tumors expressing *Ehmt2* shRNA compared to those expressing NS shRNA.

These new results are shown below and **Fig. 4F, 4J, EV9B, 4K, EV9C** and **Fig. EV11G** and **Fig. EV12G** of this revised manuscript.

Fig. 4F

Fig. EV9B

Fig. 4J

Fig. 4K

Fig. EV9C

EV11G

EMT6-*F-Luc*/BALB/c

EV12G

Panc02/C57BL/6

Figure. 4F, 4J, 4K, Fig. EV9B, EV9C and Fig. EV11G and Fig. EV12G. **4F.** NK92MI cells were treated with TGF- β 1 (10 ng/ml) for 24 hr followed by treated NK cells were used for NK cell-mediated cytotoxicity assay against Mel92.1 Cells. Relative NK cell-mediated cytotoxicity under indicated conditions are plotted. **4J.** NK92MI cells were treated with TGF- β 1 (10 ng/ml) for 24 hr and analyzed for NKG2D protein using immunoblotting. ACTINB was used as a loading control. **EV9B.** NK92MI cells were treated with TGF- β 1 (10 ng/ml) for 24 hr and

analyzed for *NKG2D* mRNA levels using RT-qPCR. Relative mRNA levels are presented under the indicated conditions. *ACTINB* was used for normalization. **4K.** NK92MI cells were treated with TGF- β 1 (10 ng/ml) for 24 hr and analyzed for CD69, CD49a, CD103 CD56 and CXCR6 proteins using immunoblotting. *ACTINB* was used as a loading control. **EV9C.** NK92MI cells were treated with TGF- β 1 (10 ng/ml) for 24 hr and analyzed for *CD69*, *CD49a*, *CD103*, *CD56* and *CXCR6* mRNA levels using RT-qPCR. Relative mRNA levels are presented under the indicated conditions. *ACTINB* was used for normalization. **EV11G.** EMT6 cells expressing NS shRNA or *Ehmt2* shRNAs were injected subcutaneously into BALB/c mice (n = 5). Measurement of tumor-infiltrated NK cells (Lin-CD49b⁺CD49a⁻) and ILC1 (Lin-CD49b⁻CD49a⁺) in the EMT6 tumors (n=5) under indicated conditions using FACS analysis. **EV12G.** Panc02 cells expressing NS shRNA or *Ehmt2* shRNAs were injected subcutaneously into C57BL/6 mice (n = 5). Measurement of tumor-infiltrated NK cells (Lin-CD49b⁺CD49a⁻), and ILCs (Lin-CD49b⁻CD49a⁺) in the Panc02 tumors (n=5) under indicated conditions using FACS analysis. All quantitative data are presented as the mean \pm SEM; ns, not significant, *p< 0.05, **p< 0.01.

One concern regarding the medical impact is whether systemic application of EHMT2 inhibitors will be tolerated by patients as well as in mice. The potential side effects of systemic EMHT2 inhibition in the mouse model could be monitored more thoroughly than just the mortality rates of the treatment. For instance, changes in body weight or any impacts on routine blood/serum parameters that would suggest organ dysfunction? Any changes on the morphology of major organs?

This is an important question. While we cannot comment on the tolerability of BRD4770 in human patients, an assessment that would require Phase I clinical trials and is beyond the scope of this manuscript, we have conducted additional studies in response to the reviewer's comments.

Based on the reviewer's comments, because we only used BRD4770 for the presented studies in our manuscript we now present a large set of new data demonstrating that BRD4770, the EHMT2 inhibitor used in our in vivo studies, is well tolerated in mice.

As the reviewer will see, there are no significant adverse effects associated with BRD4770 treatment in the mouse strains used in our studies (C57BL/6, BALB/c and NSG). These new results are shown below and in **Figure EV15** of the revised manuscript.

Figure EV15. Analysis of BRD4770 induced toxicity in mice. **A.** C57BL/6, BALB/c and

NSG mice were treated with vehicle or BRD4770 (15 mg/kg body weight) for 7 weeks. Body weights were measured at the indicated weeks. Average body weights (n=6) are plotted at the indicated time points. **B.** C57BL/6, BALB/c and NSG mice were treated with vehicle or BRD4770 (15 mg/kg body weight). Serum ALT levels were measured at the end of the experiment and is plotted under the indicated conditions. **C.** C57BL/6, BALB/c and NSG mice were treated with vehicle or BRD4770 (15 mg/kg body weight). Serum AST levels were measured at the end of the experiment and is plotted under the indicated conditions. **D.** C57BL/6, BALB/c and NSG mice were treated with vehicle or BRD4770 (15 mg/kg body weight). Serum creatinine levels were measured at the end of the experiment and is plotted under the indicated conditions. **E.** Histopathological examination of vital organs (heart, lung, liver, and kidney) from C57BL/6 mice treated with vehicle or BRD4770 (15 mg/kg body weight) as shown by H&E staining images (Scale bar, 4×: 250 µm, 20×: 50 µm). **F.** Histopathological examination of vital organs (heart, lung, liver, and kidney) from BALB/c mice treated with vehicle or BRD4770 (15 mg/kg body weight) shown by H&E staining images (Scale bar, 4×: 250 µm, 20×: 50 µm). **G.** Histopathological examination of vital organs (heart, lung, liver, and kidney) from NSG mice treated with vehicle or BRD4770 (15 mg/kg body weight) shown by H&E staining images (Scale bar, 4×: 250 µm, 20×: 50 µm). All quantitative data are presented as the mean ± SEM; ns, not significant.

The authors should at least explain why shRNAs were used for knockdowns instead of other techniques, namely CRISPR.

The reviewer raises a valid question. Our lab has access to knockdown-validated shRNAs targeting several thousand mouse and human genes, which we have used over more than a decade for research in our laboratory. Since we were able to achieve a potent EHMT2 knockdown using these existing shRNAs in our laboratory, we did not see a strong scientific rationale for using a sgRNA-based CRISPR knockout approach, as the two are similar.

However, motivated by reviewer's comment, as well as a similar suggestion from Reviewer 1, we generated a CRISPR knockout of *EHMT2* using single-guide RNAs (sgRNAs). We now show that, similar to shRNA-based approach, CRISPR-mediated knockout of *EHMT2* promotes NK cell-mediated cytotoxicity. These new findings are now shown below and have been included in **Fig. EV4A and EV4B** of the revised manuscript.

Figure EV4A and EV4B. CRISPR-based genetic knockout of *EHMT2* induces NK cell-mediated eradication of uveal melanoma cells. **A.** Mel92.1 and MP41 cells expressing either non-specific single-guide RNA (NS sgRNA) or *EHMT2* sgRNAs were analyzed for indicated proteins using immunoblotting. Histone H3 and ACTINB were used as loading controls. **B.** Mel92.1 and MP41 cells expressing either NS sgRNA or *EHMT2* sgRNAs were analyzed for NK cell-mediated cytotoxicity using LDH-based cytotoxicity assay. Relative NK cell-mediated cytotoxicity under indicated conditions are plotted. All quantitative data are presented as the mean \pm SEM; **** $p < 0.0001$.

TGF-beta itself is known to induce epigenetic changes in NK cells. The authors should at least discuss this topic.

We thank the reviewer for bringing this point to our attention. We have included this relevant topic to the Discussion section of the revised manuscript.

Minor revision:

By reading the abstract is initially not clear whether inhibition of *EHMT2* in tumor cells or NK cells is responsible for the phenotype: "We identified the histone methyltransferase *EHMT2* as a key suppressor of NK cell-mediated cytotoxicity. *EHMT2* inhibition enhanced NK cell-mediated elimination of diverse cancers, including uveal melanoma, breast cancer, and pancreatic cancer."

I suggest the following: "We identified the histone methyltransferase *EHMT2* as a key suppressor of NK cell-mediated cytotoxicity. *EHMT2* inhibition IN CANCER CELLS enhanced NK cell-mediated elimination of diverse cancers, including uveal melanoma, breast cancer, and pancreatic cancer."

We apologize for the ambiguity in the statement, and, as recommended by the reviewer, we have now revised this sentence to improve clarity.

References

1. Neri S, Mariani E, Meneghetti A, Cattini L, Facchini A. Calcein-acetyoxymethyl cytotoxicity assay: standardization of a method allowing additional analyses on recovered effector cells and supernatants. *Clin Diagn Lab Immunol.* 2001;8(6):1131-5.
2. Lorenzo-Herrero S, Sordo-Bahamonde C, Gonzalez S, Lopez-Soto A. Evaluation of NK cell cytotoxic activity against malignant cells by the calcein assay. *Methods Enzymol.* 2020;631:483-95.
3. Cho H, Chung JY, Kim S, Braunschweig T, Kang TH, Kim J, et al. MICA/B and ULBP1 NKG2D ligands are independent predictors of good prognosis in cervical cancer. *BMC Cancer.* 2014;14:957.
4. Friese MA, Platten M, Lutz SZ, Naumann U, Aulwurm S, Bischof F, et al. MICA/NKG2D-mediated immunogene therapy of experimental gliomas. *Cancer Res.* 2003;63(24):8996-9006.
5. Salih HR, Antropius H, Gieseke F, Lutz SZ, Kanz L, Rammensee HG, et al. Functional expression and release of ligands for the activating immunoreceptor NKG2D in leukemia. *Blood.* 2003;102(4):1389-96.
6. Vetter CS, Groh V, thor Straten P, Spies T, Brocker EB, Becker JC. Expression of stress-induced MHC class I related chain molecules on human melanoma. *J Invest Dermatol.* 2002;118(4):600-5.
7. Watson NF, Spendlove I, Madjd Z, McGilvray R, Green AR, Ellis IO, et al. Expression of the stress-related MHC class I chain-related protein MICA is an indicator of good prognosis in colorectal cancer patients. *Int J Cancer.* 2006;118(6):1445-52.
8. Bugide S, Green MR, Wajapeyee N. Inhibition of Enhancer of zeste homolog 2 (EZH2) induces natural killer cell-mediated eradication of hepatocellular carcinoma cells. *Proc Natl Acad Sci U S A.* 2018;115(15):E3509-E18.
9. Raulet DH, Gasser S, Gowen BG, Deng W, Jung H. Regulation of ligands for the NKG2D activating receptor. *Annu Rev Immunol.* 2013;31:413-41.
10. Diefenbach A, Jensen ER, Jamieson AM, Raulet DH. Rae1 and H60 ligands of the NKG2D receptor stimulate tumour immunity. *Nature.* 2001;413(6852):165-71.
11. Lee YS, Choi H, Cho HR, Son WC, Park YS, Kang CD, et al. Downregulation of NKG2DLs by TGF-beta in human lung cancer cells. *BMC Immunol.* 2021;22(1):44.
12. Castriconi R, Dondero A, Bellora F, Moretta L, Castellano A, Locatelli F, et al. Neuroblastoma-derived TGF-beta1 modulates the chemokine receptor repertoire of human resting NK cells. *J Immunol.* 2013;190(10):5321-8.
13. Han B, Mao FY, Zhao YL, Lv YP, Teng YS, Duan M, et al. Altered NKp30, NKp46, NKG2D, and DNAM-1 Expression on Circulating NK Cells Is Associated with Tumor Progression in Human Gastric Cancer. *J Immunol Res.* 2018;2018:6248590.
14. Castriconi R, Cantoni C, Della Chiesa M, Vitale M, Marcenaro E, Conte R, et al. Transforming growth factor beta 1 inhibits expression of NKp30 and NKG2D receptors: consequences for the NK-mediated killing of dendritic cells. *Proc Natl Acad Sci U S A.* 2003;100(7):4120-5.
15. Bugide S, Gupta R, Green MR, Wajapeyee N. EZH2 inhibits NK cell-mediated antitumor immunity by suppressing CXCL10 expression in an HDAC10-dependent manner. *Proc Natl Acad Sci U S A.* 2021;118(30).

16. Wennerberg E, Kremer V, Childs R, Lundqvist A. CXCL10-induced migration of adoptively transferred human natural killer cells toward solid tumors causes regression of tumor growth in vivo. *Cancer Immunol Immunother.* 2015;64(2):225-35.
17. Gao JQ, Tsuda Y, Han M, Xu DH, Kanagawa N, Hatanaka Y, et al. NK cells are migrated and indispensable in the anti-tumor activity induced by CCL27 gene therapy. *Cancer Immunol Immunother.* 2009;58(2):291-9.
18. Wang X, Zhang Y, Wang S, Ni H, Zhao P, Chen G, et al. The role of CXCR3 and its ligands in cancer. *Front Oncol.* 2022;12:1022688.
19. Peng L, Renauer PA, Sferruzza G, Yang L, Zou Y, Fang Z, et al. In vivo AAV-SB-CRISPR screens of tumor-infiltrating primary NK cells identify genetic checkpoints of CAR-NK therapy. *Nat Biotechnol.* 2025;43(5):752-61.
20. Gao Y, Souza-Fonseca-Guimaraes F, Bald T, Ng SS, Young A, Ngiow SF, et al. Tumor immunoevasion by the conversion of effector NK cells into type 1 innate lymphoid cells. *Nat Immunol.* 2017;18(9):1004-15.
21. Cortez VS, Cervantes-Barragan L, Robinette ML, Bando JK, Wang Y, Geiger TL, et al. Transforming Growth Factor-beta Signaling Guides the Differentiation of Innate Lymphoid Cells in Salivary Glands. *Immunity.* 2016;44(5):1127-39.
22. Jacquilot N, Seillet C, Vivier E, Belz GT. Innate lymphoid cells and cancer. *Nat Immunol.* 2022;23(3):371-9.

3rd Nov 2025

Dear Prof. Wajapeyee,

Thank you for submitting your revised study. We have now received the reports from referees #2 and #3. As you will see below, they are satisfied with the revisions, and I will therefore be able to accept your manuscript once the following editorial concerns are addressed:

1/ Manuscript text:

- Please indicate in track changes mode any new modification in the text.
- An email bounced for groping.cai@yale.edu, please check and correct if needed.
- Please provide up to 5 keywords.
- Materials and methods should be renamed Methods:
 - o Antibodies: please provide dilutions/concentrations
 - o Animals: for all, please indicate age and gender at time of experiment. Please provide reference number for approval.
 - o Statistics: please provide a statement on sample size, exclusion/inclusion criteria, blinding and randomization.
- Data availability section: please note that the dataset must be public before acceptance of the manuscript. Please provide a URL for the deposited dataset.

2/ Figures:

- Please provide individual data points in the graphs where possible.
- You currently have 8 main figures and 16 EV figures: please either merge some EV figures together, or include some EV figures in an Appendix file (figures bundled together with their legends in a single PDF file called *Appendix*, which should start with a short Table of Content. Appendix figures should be referred to in the main text as: "Appendix Figure S1, Appendix Figure S2" etc.)
- Please make sure that all figures and figure panels are referenced in the text. Currently, callouts are missing for panels A and B in Figure EV16.
- In the reagent table, please update the citation for Table EV3.
- Dataset EV legends: please correct the legend for Suppl. Table 1 to "Table EV1" and Suppl. Table 2 to "Table EV2". Suppl. Table 2 should be renamed "Dataset EV1". Please remove the table legends from the manuscript text and add them to the EV tables and to the dataset - for the dataset, please place them on a separate tab/worksheet labeled "Legend".
- Please address the queries from our data editors in the figure legends:
 1. Please note that the exact p values are not provided in the legends of figures 1C, E, G; 2C, D, F, G, J, K; 3E, G, I, J, K, L; 5D, E, F, G, H, I; 6A, C, F, G, I, J, L; 7A, C, D, E, G; 8A, C, E, G, H, J; EV1 A-E; EV2 C; EV3 A-E; EV4 B, C, F; EV5 A, B, C, D, E, F; EV8, EV9 A-C; EV10 A-D; EV11 A, F, G; EV12 A, F, G, I; EV3 B, C, D; EV14 A, B, D, F, H.
 2. Please indicate the statistical test used for data analysis in the legends of figures 1C, E, G; 2C, D, F, G, J, K; 3E, G, I, J, K, L; 5D, E, F, G, H, I; 6A, C, F, G, I, J, L, M; 7A, C, D, E, G; 8A, C, E, G, H, J; EV1 A-E; EV2 C; EV3 A-E; EV4 B, C, F; EV5 A, B, C, D, E, F; EV8, EV9 A-C; EV10 A-D; EV11 A, C, D, F, G; EV12 A, C, D, F, G, H, I; EV3 B, C, D; EV14 A, B, D, F, H.
 3. Please note that information related to n is missing in the legends of figures 1C, E, G; 2C, D, F, G, J, K; 3E, G, I, J, K, L; 4F, G, H; 5C, D, E, F, G, H, I; 6A, C, F, G, I, J, L, M; 7A, C, D, E, G; 8A, C, E, G, H, J; EV1 A-E; EV2 C, EV3 A-E; EV4 AB, C, F; EV5 A, B, C, D, E, F; EV8, EV9 A-C; EV10 A-D; EV11 A, C, D, F, G; EV12 A, C, D, F, G, H, I; EV13 B, C, D; EV 14 A, B, D, F, H

3/ Source Data:

- Please provide uncropped and annotated Western blots images (molecular markers).
- Please check the raw data provided for Fig. 6M, weeks 4 and 5.

4/ Checklist:

Please fill all subsections of "Experimental study design and statistics"

5/ Thank you for providing The Paper Explained. I have added minor modifications in the "Results" paragraph as follows. Please let me know if you agree or amend as you see fit, and add it to the manuscript text file:

"Results

This study identifies the histone methyltransferase EHMT2 as a key suppressor of NK cell-mediated anti-tumor immunity. Inhibiting EHMT2 in cancer cells, using either pharmacological inhibitors or genetic approaches, enhanced NK cell-mediated killing in several cancer types, including uveal melanoma, triple-negative breast cancer, and pancreatic cancer. EHMT2 loss led to increased expression of AZGP1, a protein that inhibits the immunosuppressive cytokine TGF- β 1. This in turn elevated the levels of activating ligands MICB and ULBP3 on tumor cells, which are recognized by NK cells. Additionally, EHMT2 loss increased the expression of chemokines such as CXCL10 and CCL27 in cancer cells, promoting NK cell migration. In immunocompetent mouse models, inhibition of EHMT2 suppressed tumor growth in an NK cell-dependent manner. These effects were lost in immune-deficient mice lacking NK cells, confirming that the anti-tumor activity of EHMT2 inhibition is mediated predominantly through NK cells."

6/ Synopsis: thank you for providing a nice visual abstract. I have cropped a small portion to serve as thumbnail for the table of content on our webpage (attached), please let me know if you agree or provide an alternative image at the same dimensions (115px x 70px).

7/ As part of the EMBO Publications transparent editorial process initiative (see our Editorial at <http://embomolmed.embopress.org/content/2/9/329>), EMBO Molecular Medicine will publish online a Review Process File (RPF) to accompany accepted manuscripts.

This file will be published in conjunction with your paper and will include the anonymous referee reports, your point-by-point response and all pertinent correspondence relating to the manuscript. Let us know whether you agree with the publication of the RPF and as here, if you want to remove or not any figures from it prior to publication.

I look forward to receiving your revised manuscript.

Yours sincerely,

Lise Roth

***** Reviewer's comments *****

Referee #2 (Comments on Novelty/Model System for Author):

The authors have addressed my concerns and improved their model.

Referee #2 (Remarks for Author):

The authors have substantially improved the manuscript and included many additional experiments which align with the hypothesis and strengthen the results.

Minor point: As the NK cell-depletion does not strongly increase the tumor size when the mice are injected with EMT6 NTshRNA it would have been more suitable to perform the adoptive transfer experiment with EMT6 with the Ehmt2 KD. However, this is not a necessary experiment anymore as the authors could provide enough evidence for NK-cell dependence of their model.

Referee #3 (Comments on Novelty/Model System for Author):

The medical impact was rated medium because it is difficult to estimate the potential side effects of systemic EHMT2 inhibition in human patients. However the authors have made a great effort to experimentally address potential adverse effects in the preclinical setting.

Referee #3 (Remarks for Author):

The authors have sufficiently addressed the concerns that were initially raised. Congratulations!

***** Reviewer's comments *****

Referee #2 (Comments on Novelty/Model System for Author):

The authors have addressed my concerns and improved their model.

Referee #2 (Remarks for Author):

The authors have substantially improved the manuscript and included many additional experiments which align with the hypothesis and strengthen the results. Minor point: As the NK cell-depletion does not strongly increase the tumor size when the mice are injected with EMT6 NTshRNA it would have been more suitable to perform the adoptive transfer experiment with EMT6 with the Ehmt2 KD. However, this is not a necessary experiment anymore as the authors could provide enough evidence for NK-cell dependence of their model.

We thank the reviewer for noting our effort in revising the manuscript and stating that “the authors have addressed my concerns and improved their model”. The reviewer has no additional comments related to our revised manuscript.

Referee #3 (Comments on Novelty/Model System for Author):

The medical impact was rated medium because it is difficult to estimate the potential side effects of systemic EHMT2 inhibition in human patients. However the authors have made a great effort to experimentally address potential adverse effects in the preclinical setting.

Referee #3 (Remarks for Author):

The authors have sufficiently addressed the concerns that were initially raised. Congratulations!

We thank the reviewer for acknowledging our effort in revising the manuscript and stating that “the authors have sufficiently addressed the concerns that were initially raised”. The reviewer has no additional comments related to our revised manuscript.

24th Nov 2025

Dear Prof. Wajapeyee,

Thank you for submitting your revised files. I am pleased to inform you that your manuscript is accepted for publication and is now being sent to our publisher to be included in the next available issue of EMBO Molecular Medicine!

Please note that while we appreciate you providing a new point-by-point rebuttal letter with minor changes to the figure legends, this was not seen by the referees when they performed their evaluation. As the RPF is intended to make the review process transparent, the original point-by-point rebuttal will be included in the RPF instead.

Yours sincerely,

Lise Roth
